# A Spectral-Grassmann Wasserstein metric for operator representations of dynamical systems

**Thibaut Germain**
CMAP, Ecole Polytechnique
`thibaut.germain@polytechnique.edu`

**Rémi Flamary**
CMAP, Ecole Polytechnique
`remi.flamary@polytechnique.edu`

**Vladimir R. Kostic**
Istituto Italiano di Tecnologia
& University of Novi Sad
`vladimir.kostic@iit.it`

**Karim Lounici**
CMAP, Ecole Polytechnique
`karim.lounici@polytechnique.edu`

## Abstract

The geometry of dynamical systems estimated from trajectory data is a major challenge for machine learning applications. Koopman and transfer operators provide a linear representation of nonlinear dynamics through their spectral decomposition, offering a natural framework for comparison. We propose a novel approach that represents each system as a distribution over its joint operator eigenvalues and spectral projectors and defines a metric between systems leveraging optimal transport. The proposed metric is invariant to the sampling frequency of trajectories. It is also computationally efficient, supported by finite-sample convergence guarantees, and enables the computation of Fréchet means, providing interpolation between dynamical systems. Experiments on simulated and real-world datasets show that our approach consistently outperforms standard operator-based distances in machine learning applications, including dimensionality reduction and classification, and provides meaningful interpolation between dynamical systems.

## 1 Introduction

Dynamical systems are widely used across scientific and engineering disciplines to model state variables' evolution over time (Lasota & Mackey, 2013). Nonlinear ordinary or partial differential equations typically govern these systems and may incorporate stochastic components (Meyn & Tweedie, 2012). However, in many practical situations, analytical models are unavailable or intractable, motivating the use of data-driven approaches to infer the underlying dynamics from sampled trajectories. In this context, Koopman and transfer operator regressions have emerged as a powerful framework for learning and interpreting dynamical systems from data (Brunton et al., 2022). Rather than directly modeling the evolution of state variables, these operators advance observables (scalar functions defined on the state space) by mapping each to its expected future value conditioned on the current state. Crucially, these operators are linear even when the underlying systems are not linear. Under suitable conditions, they admit a spectral decomposition that provides insight into the system's long-term behavior, stability, and modal structure (Mauroy et al., 2020). These properties have made the operator-centric framework particularly appealing for both theoretical analysis and practical applications across various domains, including chemistry for molecular kinetics explainability (Wu et al., 2017), robotics for control (Bruder et al., 2020), and fluid dynamics for prediction (Lange et al., 2021).

**Koopman and transfer operators for dynamical systems.** From a learning standpoint, Koopman and transfer operators provide a compact and structured representation of dynamical systems, making them well-suited for machine learning applications requiring system comparison, such as time series classification (Surana, 2020) and dynamical graph clustering (Klus & Djurdjevac Conrad, 2023). However, in order to leverage these representations in standard statistical and machine

learning pipelines, one must first define a meaningful metric between them. Unfortunately, despite recent advances in operator estimation (Colbrook et al., 2023; Kostic et al., 2023; 2024a; Bevanda et al., 2023), the development of similarity measures between operator representations of dynamical systems remains relatively underexplored despite the growing need for interpretable metrics on dynamical systems in machine learning applications (Ishikawa et al., 2018).

**Comparing dynamical systems.** We succinctly review existing similarity measures on dynamical systems; a detailed account is given in Appendix A. The case of (stochastic) linear dynamical systems (LDSs) and linear state-space models was first addressed in the literature (Afsari & Vidal, 2014). While early metrics are theoretically sound and leverage the manifold structure of LDS spaces, they suffer from high computational cost, making them impractical in most machine learning settings (Hanzon & Marcus, 1982; Gray, 2009). Originally designed for ARMA models, the Martin pseudo-metric (Martin, 2002) offers a practical alternative and has later been extended to general LDS spaces and inspired kernel-based variants (Chaudhry & Vidal, 2013). These measures have been generalized to nonlinear systems through the Koopman/transfer operator framework (Fujii et al., 2017; Ishikawa et al., 2018). More recent work considers topological conjugacy, where similarities can be defined via alignment methods (Ostrow et al., 2023; Glaz, 2025) or Optimal Transport (OT) between operator spectra (Redman et al., 2024; Zhang et al., 2025). A related line of research studies Wasserstein-type metrics on functional spaces such as Antonini & Cavalletti (2021), introducing OT between measures derived from the eigenvalues of normal operators.
The above approaches face key limitations. Norm-based measures and the Martin pseudo-metric are noise-sensitive and lack interpretability. OT-based similarities improve interpretability by comparing spectral geometry, but they are restricted to self-adjoint operators and define pseudo-metrics rather than metrics. As a result, no existing method combines theoretical soundness, robustness, and computational efficiency, and defining a principled metric for dynamical systems remains an open challenge.

**Contributions.** In Section 3, we introduce a novel representation of transfer operators as joint distributions over eigenvalues and eigenspaces. Building on tools from optimal transport and Grassmann geometry, we propose a new Wasserstein metric, named Spectral–Grassmann Optimal Transport (SGOT), that compares transfer operators through their joint spectral distribution. We show that SGOT is theoretically well-founded, computationally efficient, and broadly applicable, as it is compatible with any operator estimation method. Assuming operator estimation via reduced-rank regression, we further strengthen the theory of non-parametric spectral learning by establishing learning bounds for Koopman eigenvalues and eigenfunctions under weaker regularity assumptions, which in turn yield finite-sample convergence guarantees for SGOT. We then exploit SGOT to design a scalable algorithm for computing Fréchet barycenters of dynamical systems, enabling new forms of system averaging and interpolation. Finally, in Section 4, we empirically validate the advantages of our approach over existing metrics and demonstrate its utility in system interpolation and machine learning tasks, using operator estimators derived from kernel methods and deep learning.

## 2 BACKGROUND

**Linear evolution operators.** Let $(X_t)_{t \in \mathbb{T}}$ be the flow in some state space $\mathcal{X}$ whose governing laws are temporally invariant, where the time index $t$ can be either discrete ($\mathbb{T} = \mathbb{N}_0$) or continuous ($\mathbb{T} = [0, +\infty)$). While the flows of many important dynamical systems are nonlinear and possibly stochastic, under quite general assumptions they admit *linear operator representations* on a suitably chosen space of real-valued functions $\mathcal{F} \subset \mathbb{R}^{\mathcal{X}}$, henceforth referred to as observable space. Namely, letting $t \in \mathbb{T}$, the *transfer operator*, also known as *Koopman operator* for deterministic systems, $A_t \colon \mathcal{F} \to \mathcal{F}$ evolves an observable $f \colon \mathcal{X} \to \mathbb{R}$ for time $t$ via conditional expectation

$$[A_t(f)](x) := \mathbb{E}[f(X_t) | X_0 = x], \ x \in \mathcal{X}. \tag{1}$$

Clearly, since $A_t A_s = A_{t+s}$, in the discrete-time setting the process can be studied only through the transfer operator $A := A_1$ of one unit of time, typically a second. On the other hand, when time is continuous, the process is characterized by the infinitesimal generator of the semigroup $(A_t)_{t \geq 0}$, defined as $L := \lim_{t \to 0^+} (A_t - \mathrm{Id})/t$ that is a differential operator with domain in $\mathcal{F}$ that encodes the equations of motion and generate dynamics as $A_t = e^{Lt}$, see Lasota & Mackey (1994); Ross (1995).

**Spectral decomposition.** The utility of transfer operator representations stems from its linearity on a suitably chosen $\mathcal{F}$ that is *invariant* space under the action of $A_t$, that is $A_t[\mathcal{F}] \subseteq \mathcal{F}$ for all $t$ (a property that we tacitly assumed above), and *rich enough* to represent the flow of the process, i.e., it contains observables from which we can reconstruct all the relevant information of the state (e.g. in the case of a stochastic system distribution $\mu_t$ at any time $t$ ). Namely, using the spectral theory of linear operators (Kato, 2013), under suitable assumptions, one can spectrally decompose generator $L = \sum_{j \in J} (\lambda_j P_j + N_j) + P_c L$ into distinct complex scalars $\lambda_j \in \mathbb{C}$, called eigenvalues, forming point-spectrum and mutually commuting Riesz spectral projectors $P_j$ that satisfy satisfy equations $LP_j = \lambda_j P_j$ and $P_c = I - \sum_j P_j$, $P_J$ being of finite rank $m_j$ (geometric multiplicity), $N_j$ being nilpotent, and $j \in J$ being countably many. Assuming for simplicity that $\mathcal{F}$ is a separable Hilbert space and $L$ is a non-defective operator with purely discrete spectrum, e.g. stable diffusion processes, see Ross (1995), we have that $L = \sum_{j \in \mathbb{N}} \lambda_j g_j \otimes_{\mathcal{F}} f_j$, with $Lf_j = \lambda_j f_j$, $L^* g_j = \overline{\lambda_j} g_j$, and $\langle f_j, g_j \rangle_{\mathcal{F}} = \delta_{i,j}$, where $(\lambda_j, f_j, g_j)_{j \in \mathbb{N}}$ are eigen-triplets consisting of an eigenvalue, left and right eigenfunction, respectively. This, in turn, allows one to decouple the evolution of an arbitrary observable $f \in \mathcal{F}$

$$\mathbb{E}[f(X_t) \mid X_0 = x_0] = [A_t f](x) = \sum_{j \in \mathbb{N}} e^{\lambda_j t} \langle f, g_j \rangle_{\mathcal{F}} f_j(x_0) = \sum_{j \in \mathbb{N}} e^{\tau_j t} e^{i2\pi\omega_j t} m_j^f(x_0), \quad (2)$$

into modes $m_j^f = \langle f, g_j \rangle_{\mathcal{F}} f_j \colon \mathcal{X} \to \mathbb{R}$ that evolve as scalar oscillators at timescales given by reciprocals of $\tau_j = \Re(\lambda_j)$ and frequencies $\omega_j = \Im(\lambda_j) / 2\pi$ in Hz (assuming time in seconds).

**Learning transfer operators.** In machine learning applications, dynamical systems are only observed, and neither $A$ nor its domain, such as the space of square integrable functions w.r.t. the equilibrium measure, is known, providing a key challenge to learn them from data. The most popular algorithms (Brunton et al., 2022) aim to learn the action of $A \colon \mathcal{F} \to \mathcal{F}$ on a predefined, possibly infinite dimensional, Reproducing Kernel Hilbert Space (RKHS), resulting in estimating the *restriction* of $A$ on $\mathcal{H} \subseteq \mathcal{F}$ by projection, that is $P_{\mathcal{H}} A_{|_{\mathcal{H}}} \colon \mathcal{H} \to \mathcal{H}$, typically via empirical risk minimization (Kawahara, 2016; Kostic et al., 2022). When $\mathcal{H}$ is given by a universal reproducing kernel (Steinwart & Christmann, 2008), meaning it is dense in $\mathcal{F}$, such techniques have strong spectral estimation guarantees (Kostic et al., 2023), can forecast well the states (Bevanda et al., 2023; Alexander & Giannakis, 2020), and evolve distributions of stochastic processes via kernel mean embeddings (Kostic et al., 2024c). As an alternative, finite-dimensional $\mathcal{H}$ spaces can be used with these methods (Kutz et al., 2016) or be learned from data in the form of rich neural representations (Liu et al., 2024), that can be also trained to minimize the projection error $\|P_{\mathcal{H}}^{\perp} A_{|_{\mathcal{H}}}\|_{\mathcal{F} \to \mathcal{F}}$ (Kostic et al., 2024b). In these settings, a major limitation of the existing statistical learning guarantees is assuming well-specifiedness, i.e., the existence of an exact RKHS representation of $A$. A more realistic learning scenario requires that only the most relevant spectral part of $A$ lives in a suitable universal RKHS space.

**Discrete optimal transport.** Optimal Transport (OT) is a well-defined framework to compare probability distributions, with many applications in machine learning (Peyré et al., 2019). In discrete OT, one seeks a transport plan mapping samples from a source distribution to those of a target distribution while minimizing a transportation cost. Formally, consider $\mathcal{Z}_S = \{z_i \in \mathcal{Z} \mid i \in [k_S]\}$ and $\mathcal{Z}_T = \{z_i' \in \mathcal{Z} \mid i \in [k_T]\}$ as the sets of source and target samples in a space $\mathcal{Z}$. We associate to these sets the probability distributions $\mu_S = \sum_{i \in [k_S]} a_i \delta_{z_i}$ and $\mu_T = \sum_{i \in [k_T]} b_i \delta_{z_i'}$ with $(\mathbf{a}, \mathbf{b}) \in \Delta^{k_S} \times \Delta^{k_T}$ and $\Delta^n = \{\mathbf{p} \in \mathbb{R}_+^n \mid \sum_{i \in [n]} p_i = 1\}$ the $n$-simplex. Let $\mathbf{C} \in \mathbb{R}_+^{k_S \times k_T}$ be the cost matrix with $C_{ij} = c(z_i, z_j')$ being the transport cost between $z_i$ and $z_j'$ given by the cost function $c$. The Monge-Kantorovich problem aims at identifying a coupling matrix, also denoted as OT plan $\mathbf{P}^* \in \mathbb{R}_+^{k_S \times k_T}$, that is a solution of the constrained linear problem:

$$\min_{\mathbf{P} \in \Pi(\mu_S, \mu_T)} \langle \mathbf{C}, \mathbf{P} \rangle_F \quad \text{s.t} \quad \Pi(\mu_S, \mu_T) = \{\mathbf{P} \in \mathbb{R}_+^{k_S \times k_T} \mid \mathbf{P}\mathbf{1} = \mathbf{a}, \mathbf{P}^\intercal \mathbf{1} = \mathbf{b}\}, \quad (3)$$

where $\Pi(\mathbf{a}, \mathbf{b})$ is the set of joint-distributions over $\mathcal{Z}_S \times \mathcal{Z}_T$ with marginals $\mathbf{a}$ and $\mathbf{b}$. In what follows, we denote $L_c(\mu_S, \mu_T)$ the application returning the optimal value of problem (3) where $c$ indicates the cost function. A fundamental property of OT is that, under suitable conditions on the cost function, the Wasserstein distance defined as $W_p(\mu, \nu) \triangleq (L_{d^p}(\mu, \nu))^{\frac{1}{p}}$ is a metric on the space of probability measures, see Villani et al. (2008, Theorem 6.18).

## 3 SPECTRAL-GRASSMANN OPTIMAL TRANSPORT (SGOT)

**Problem setting and assumptions.** Machine learning tasks on (stochastic) dynamical systems, such as comparing trajectories, identifying regimes, or clustering dynamics, require a discriminative and computationally efficient notion of distance between observed processes. We address this by representing each system through its associated Koopman/transfer operator and then introducing the SGOT metric to compare them. To that end, let us formalize the problem setting (A1) and make the main learnability assumptions (A2)-(A3) necessary to obtain strong statistical learning guarantees.

**(A1, Dynamical system sampling)** Consider $N \in \mathbb{N}^*$ time homogeneous, Markovian dynamical systems defined on a common state space $\mathcal{X}$ and characterized by their generators $L_k \colon \text{dom}(L_k) \subset \mathcal{F}_k \to \mathcal{F}_k$ defined on the respective spaces $\mathcal{F}_k$ of observables $\mathcal{X} \to \mathbb{R}$, $k \in [N]$. For every $k \in [N]$, let $\mathcal{D}_k = \{(x_i^k, y_i^k)\}_{i \in [n_k]}$ be a dataset of observations of the of the $k$-th system, consisting of consecutive states separated by time-lag $\Delta t_k$. Notably, in the case of a single trajectory $y_i^k = x_{i+1}^k$.

Since data-driven methods can distinguish between systems only up to the temporal resolution at which the observations are made (Zayed, 2018), recalling equation 2, the systems differing in spectral components beyond the observable range of timescales $1/\tau_j$ and frequencies $\omega_j$ are undistinguishable from measurements. Therefore, we focus below on spectral projections of dynamics that can be learned from finite data.

**(A2, Low rank)** For every $k \in [N]$ there exists $r_k \in \mathbb{N}$ such that $r_k$ eigenvalues of $L_k$ closest to the origin are separated from the rest of the spectrum, and let $P_{\leq r_k} \colon \mathcal{F}_k \to \mathcal{F}_k$ denote the corresponding spectral projector.

Recalling the case of dynamical systems sampled at equilibrium, i.e. $\mathcal{F}_k = \mathcal{L}^2_{\pi_k}(\mathcal{X})$ with $\pi_k$ being the invariant measure of the $k$-th system, a central conceptual difficulty in introducing distance between systems is that transfer operators for different systems naturally act on different spaces, and therefore cannot be compared directly. To resolve this, we restrict each operator to a common reproducing kernel Hilbert space $\mathcal{H}$ that is included in the domain of all the transfer operators.

**(A3, Common functional space)** Let $\mathcal{H}$ be a separable RKHS associated with kernel $\kappa$, such that for all $k \in [N]$ it holds $\text{Im}(P_{\leq r_k} L_k) \subset \mathcal{H} \subset \mathcal{F}_k$. Hence, there exists representation $T_k = e^{(P_{\leq r_k} L_k)_{|\mathcal{H}}} \colon \mathcal{H} \to \mathcal{H}$ with spectral decomposition $T_k = \sum_{j \in [\ell_k]} e^{\lambda_j^k} Q_j^k$ where $\ell_k$ is the number of distinct eigenvalues.

First, we remark that this assumptions is significantly relaxed compare to the most related prior work (Kostic et al., 2023). In fact, for typical Langevin dynamics, under suitable assumptions on the potential, a universal Gaussian RBF RKHS with properly chosen landscale parameter contains a finite number of leading eigenfunctions of generators $L_k$ defined on $\mathcal{L}^2_{\pi_i}(\mathcal{X})$ spaces weighted by Boltzmann distributions $\pi_k$. Thus, (A3) holds true, while aforementioned is violated. Moreover, one can formally build a finite-dimensional space $\mathcal{H}$ by choosing exactly the basis of such a generator's eigenfunctions; the complexity of the problem is then transferred to learning $\mathcal{H}$. Beyond this case, one can similarly work in other domains, see e.g. Colbrook et al. (2025); Alexander & Giannakis (2020); Bevanda et al. (2023).

Finally, if Assumptions (A2) and (A3) hold, we obtain an exact spectral representation in $\mathcal{H}$, which, as we will show, enables unbiased operator comparison. When these assumptions are violated, the comparison incurs a bias that can be assessed through the metric distortion between the chosen common subspace and the true operator domain. This phenomenon is analyzed in (Kostic et al., 2023), which also provides an efficient empirical estimator of the distortion.

**Spectral Grassmanian Wasserstein metric.** Since the spectral decomposition of a non-defective operator $T_k$ into its eigenvalues and spectral projectors is uniquely defined up to a permutation, any meaningful comparison approach based on operators' spectral decomposition must be invariant to permutations and change of basis in which spectral projectors are expressed. While discrete optimal transport naturally provides invariance to permutations through the minimizing coupling matrix eq. (3), we need to design a ground metric that takes into account both spectral and subspace aspects to obtain a true OT metric. This is done below, where we define a Wasserten metric on the set of non-defective operators (complete proof in Appendix C).

**Theorem 1.** *Let $\mathcal{H}$ be a separable $\mathbb{C}$-Hilbert space and $\mathcal{S}_r(\mathcal{H})$ the set of non-defective operators with rank at most $r \in \mathcal{D}$. Let $(\mathcal{G}, d_{\mathcal{G}})$ be the Grassmanian manifold of the space of Hilbert-Schmidt operators on $\mathcal{H}$. Given $p \in \mathbb{N}^*$ and $\eta \in (0,1)$, let $\mu: S_r(\mathcal{H}) \to \mathcal{P}_p(\mathbb{C} \times \mathcal{G})$ and $d_\eta: (\mathbb{C} \times \mathcal{G})^2 \to \mathbb{R}_+$ be given by*

$$\mu(T) \triangleq \sum_{j \in [\ell]} \frac{m_j}{m_{tot}} \delta_{(\lambda_j, \mathcal{V}_j)} \quad and \quad d_\eta[(\lambda', \mathcal{V}'), (\lambda', \mathcal{V}')] \triangleq \eta|\lambda - \lambda'| + (1-\eta) d_{\mathcal{G}}(\mathcal{V}, \mathcal{V}'), \quad (4)$$

*with $|\cdot|$ applied on polar coordinates $\lambda, \lambda'$, $m_{tot} = \sum_{i \in [\ell]} m_i$, $\mathcal{V}_j$ the $m_j$-dimensional vector space in $\mathrm{HS}(\mathcal{H}, \mathcal{H})$ spanned by the rank one operators of the right/left eigenfunctions associated with the eigenvalue $e^{\lambda_j}$ of $T$ (same notation for $T'$). Then, $(\mathcal{S}_r(\mathcal{H}), d_{\mathcal{S}})$ is a metric space, where $d_{\mathcal{S}}: \mathcal{S}_r(\mathcal{H}) \to \mathbb{R}_+$ is given by*

$$d_{\mathcal{S}}(T, T') = W_{d_\eta, p}(\mu(T), \mu(T')). \quad (5)$$

First, recalling equation 2 and (A1), note that while typically in data-driven methods datasets are sampled at some frequency $\omega_k^{ref} = 1/\Delta t_k$ to estimate eigenvalues $e^{\lambda_i^k \Delta t_k}$ of transfer operators $A_k^{\Delta t_k}$, we build a metric using the difference in the generator eigenvalues. This is to compare Koopman modes' eigenvalues as physical quantities, since for the $k$-th system the observed time-scales are $\tau_j^k / \omega_k^{ref}$ and the oscillating frequencies $\omega_j^k / \omega_k^{ref}$. So, by re-normalizing eigenvalues, we can compare systems observed at different time-scales in the universal time units. Further, we remark that assuming non-defective operators is not a major bottleneck, since Theorem 1 can be extended to the space of general linear operators with rank at most $r$ by leveraging the Dunford-Jordan decomposition (Dunford & Schwartz, 1988). In this case, the cost metric in $d_\eta$ compares the spectrum and subspaces of Jordan blocks. As well, depending on the prior geometry one wishes to emphasize on $\mathcal{G}$, one can consider other metrics as depicted in Appendix C.3.

**Metric computation.** In order to evaluate the SGOT metric, one needs to compute the cost matrix (see section 2), i.e., $d_\eta$ for each pair of spectrals. Following (A1)-(A3), let $\widehat{T}$ be an operator estimated from samples $\{(x_i, y_i)\}_{i \in [n]}$ with a kernel based method. Suppose that $\widehat{T}$ admits $l$ eigenvalues, each with multiplicity $m_i$. Let $\boldsymbol{\beta}_i, \boldsymbol{\alpha}_i \in (\mathbb{C}^{n \times l_i})^2$ be the control parameters of the left/right eigenfunctions related to the $i^{\text{th}}$ eigenvalue and preprocessed to form an orthonormal basis. Let $\widehat{T}'$ be another estimated operator, and $\mathbf{M}_\epsilon \triangleq \{k(\epsilon_i, \epsilon_j')\}_{(i,j) \in [n] \times [n']}$ with $\epsilon \in \{\mathbf{x}, \mathbf{y}\}$, be the cross-kernel matrices. For $p = 1$, the cost matrix $\mathbf{C} \in \mathbb{R}_+^{l \times l'}$ is given by:

$$C_{i,j} = \eta|\lambda_i - \lambda_j'| + (1-\eta)(m_i + m_j - 2\operatorname{Tr}((\boldsymbol{\beta}_i^* \mathbf{M}_y \boldsymbol{\beta}_j)(\boldsymbol{\alpha}_i^* \mathbf{M}_x \boldsymbol{\alpha}_j)))^{\frac{1}{2}}. \quad (6)$$

With the rank $r \geq \max(l, l')$, the time complexity of $d_{\mathcal{S}}$ is in $O(n^2 r^2 + r^3 log(r))$ respectively due to the cost matrix computation and the OT solver (that is negligible for small $r$). Consequently, $d_{\mathcal{S}}$ and the kernel metric computation are asymptotically equivalent, overcoming the usual computational drawbacks of OT-based methods relative to kernel ones. If needed, both metrics can further benefit from standard kernel scaling techniques (Meanti et al., 2023).

**Statistical guarantees.** In the following, we show how using RRR estimators yields unbiased estimation of the SGOT. To that end, consider $\widehat{T}_k = (\widehat{C}_x^k + \gamma I)^{-\frac{1}{2}} [\![ (\widehat{C}_x^k + \gamma I)^{-\frac{1}{2}} \widehat{C}_{xy}^k ]\!]_{r_k}$, where $\widehat{C}_x^k = \frac{1}{n_k} \sum_{i \in [n_k]} \kappa_{x_i^k} \otimes \kappa_{x_i^k}$, $\widehat{C}_{xy}^k = \frac{1}{n_k} \sum_{i \in [n_k]} \kappa_{x_i^k} \otimes \kappa_{y_i^k}$, $\gamma > 0$ and $[\![\cdot]\!]_r$ denoting best rank-$r$ approximation. As discussed above, one can efficiently compute $d_{\mathcal{S}}(\widehat{T}_1, \widehat{T}_2)$ so that the following holds.

**Theorem 2.** *Let (A1)-(A3) hold with $k \in [2]$, $\mathcal{F}_k = \mathcal{L}_{\pi_k}^2(\mathcal{X})$ and $\kappa(x, x) < \infty$ a.s. for $x \sim \pi_k$. Let $\mathbb{E}[\widehat{C}_x^k] = C_x^k$ and assume that for some $\alpha \in [1,2]$ and $\beta \in [0,1]$ it holds that $\|\|[(C_x^k)^\dagger]^{\frac{\alpha-1}{2}} T_k \|_{\mathcal{H} \to \mathcal{H}} < \infty$ and $\lambda_i(C_x^k) \leq \lesssim i^{-1/\beta}$ for $i \in \mathbb{N}$. Given $\delta \in (0,1)$, if $n$ is large enough and $\lambda_{r_k} \lesssim - \frac{\alpha \log n}{2(\alpha+\beta)}$, then w.p.a.l. $1-\delta$ in the i.i.d. draw of samples $\mathcal{D}_1$ and $\mathcal{D}_2$ it holds $|d_{\mathcal{S}}(\widehat{T}_1, \widehat{T}_2) - d_{\mathcal{S}}(T_1, T_2)| \lesssim n^{-\frac{\alpha-1}{2(\alpha+\beta)}} \ln(2\delta^{-1})$.*

*Sketch of Proof.* To obtain this result, we needed to overcome the overly strong assumption of well-specifiedness of $\mathcal{H}$ made in Kostic et al. (2023), which significantly reduces the applicability of those bounds to estimate the distance between true generators $L_k$ with high probability. By carefully

treating the approximation errors originating from rank reductions, $\widetilde{T}_k$ being population version of $\widehat{T}_k$, we obtain $\|\widetilde{T}_k - T_k\| \lesssim \gamma^{\frac{\alpha-1}{2}} + e^{\lambda_{r_k}}$ under realistic assumption (A3). Furthermore, we derive an upper bound on the operator norm $\|\widetilde{T}_k - \widehat{T}_k\|_{\mathcal{H}\to\mathcal{H}} \lesssim \sqrt{\gamma^{-\beta-1}n^{-1}\log(\delta^{-1})}$ w.p.a.l. $1 - \delta$. Balancing the two terms gives the bound on $\|\widetilde{T}_k - T_k\|$. Next, we apply standard polar analysis and Davis-Kahan perturbation analysis to derive the bound on $d_\eta(\widetilde{T}_k, T_k)$. Finally, the stability property of the Wasserstein distance gives the final bound. Full proof is available in appendix E. $\qquad\square$

**Spectral Grassmann OT barycenter, parametric model and optimization.** Computation of barycenters is fundamental for many unsupervised methods; it is known as the *Fréchet mean problem* in metric spaces. It consists in identifying an element that minimizes a weighted sum of distances to the observations. Formally, given the importance weights $\boldsymbol{\gamma} \in \Delta^N$, assuming (A1)-(A3), and $p=2$ in Theorem 1, we aim to solve:

$$\arg\min_{T \in \mathcal{S}_r(\mathcal{H})} \sum_{k \in [N]} \gamma_i d_\mathcal{S}(T, T_k)^2, \tag{7}$$

By construction of $d_\mathcal{S}$, problem 7 corresponds to a *free-support Wasserstein barycenter* estimation problem which aims at optimizing the support of the atoms parametrizing the barycenter, in our case, its spectral decomposition. State-of-the-art algorithms typically rely on a coordinate descent scheme, alternating between transport plan computation and measure optimization (Cuturi & Doucet, 2014; Claici et al., 2018).

Whenever the RKHS $\mathcal{H}$ is infinite dimensional, the Fréchet mean problem (eq. (7)) is intractable. So we restrain the optimization over a set of parametrized operators defined such that for any $\boldsymbol{\theta} \triangleq (\boldsymbol{\lambda}, \boldsymbol{\alpha}, \boldsymbol{\beta}, \mathbf{x})$:

$$T_{\boldsymbol{\theta}} : h \in \mathcal{H} \mapsto \sum_{i \in [r]} \lambda_i \langle \kappa_\mathbf{x} \boldsymbol{\alpha}_i, h \rangle_\mathcal{H} \kappa_\mathbf{x} \boldsymbol{\beta}_i \in \mathcal{H} \tag{8}$$

where $\boldsymbol{\lambda} \in \mathbb{C}^r$, $\mathbf{x} \in \mathcal{X}^n$ are state space control points, and $\boldsymbol{\alpha}, \boldsymbol{\beta} \in \mathbb{C}^{n \times r}$ control parameters acting on the representer functions $\kappa_\mathbf{x} = \{\kappa(., x_j)\}_{j \in [n]}$ with $\kappa$ the kernel of $\mathcal{H}$, i.e. $\kappa_\mathbf{x} \boldsymbol{\alpha}_i \triangleq \sum_{j \in [n]} \kappa_{x_j} \alpha_{ji}$. While these operators are compact with rank at most $r$, further constraints on the control points and parameters are required to ensure a spectral decomposition (see eq. (2)). Together with the definition of discrete optimal transport (see Section 2), it leads to the constrained optimization problem:

$$\arg\min_{\boldsymbol{\theta}, \mathbf{P}} \sum_{i \in [N]} \gamma_i \langle \mathbf{C}_i(\boldsymbol{\theta}), \mathbf{P}_i \rangle_F \quad \text{s.t.} \quad \begin{cases} \boldsymbol{\alpha}^* \mathbf{K} \boldsymbol{\beta} = \mathbf{I} & \mathbf{K} = \{\kappa(x_i, x_j)\}_{(i,j) \in [n]^2} \\ \boldsymbol{\beta}_j^* \mathbf{K} \boldsymbol{\beta}_j = 1, \ \forall j \in [r] & \mathbf{P}_i \in \Pi(\mu(T_{\boldsymbol{\theta}}), \mu(\widehat{T}_i)), \ \forall i \in [N] \end{cases} \tag{9}$$

where $\mathbf{P} = \{\mathbf{P}_i\}_{i \in [N]}$, $\widehat{\boldsymbol{T}} = \{\widehat{T}_i\}_{i \in [N]}$, such that $(\mathbf{C}_i(\boldsymbol{\theta}), \mathbf{P}_i)$ are the cost and transport matrices associated to the Wasserstein metric $d_\mathcal{S}$ defined in Theorem 1, between the parametric operator $T_{\boldsymbol{\theta}}$ and $\widehat{T}_i$.

Following Cuturi & Doucet (2014) and considering a differentiable kernel w.r.t the control points, we propose an inexact coordinate descent scheme with a cyclic update rule for optimizing problem 9. Each cycle begins with the computation of the optimal transport plans, then the subsequent coordinate updates are performed with a few gradient descent steps and a closed-form projection scheme to enforce the constraints. In Appendix D we provide more detail about the computational and theoretical aspects of the barycenters.

## 4 NUMERICAL EXPERIMENTS

We now illustrate the benefits of our metric and barycenter through numerical experiments on dynamical systems. We first study the behavior of different similarity measures under various shifts and compare them on unsupervised and supervised machine learning tasks. Finally, we demonstrate the properties of operator barycenters using our proposed algorithm on two simulated examples.

**Compared similarity measures.** In addition to our proposed metric SGOT, we compare other OT-based similarities that focus solely on the eigenvalues (SOT) (Redman et al., 2024) or solely on the eigenspaces using a Grassmannian metric (GOT) (Antonini & Cavalletti, 2021). We also include metrics induced by the Hilbert–Schmidt and operator norms, as well as the Martin similarity (Martin, 2002).

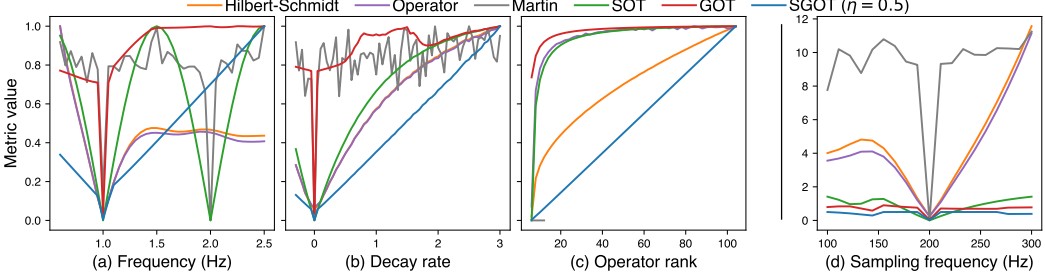

Figure 1: Similarity measures' behaviors under four scenarios of shifts of a linear oscillatory system: (a) frequency shift, (b) decay rate shift, (c) operator rank/subspace shift, (d) sampling frequency variation. In scenarios (a,b,c), metric values are normalized by their maximum.

### 4.1 COMPARISON WITH OTHER SIMILARITY MEASURES

**Simulated system and shifts.** First, we illustrate the behavior of different similarity measures between dynamical systems with regard to variations of the spectral decompositions of their Koopman operators. We consider a referent linear oscillatory system that is the sum of two simple harmonic oscillators with frequencies 0.5Hz and 1.0Hz, respectively, with a trajectory sampled at 200Hz. Considering the linear kernel, we compare the Koopman operator of the referent system with those of shifted systems according to four scenarios: **(a) Frequency shift**, changes the 1Hz harmonic frequency. **(b) Decay rate shift**, changes the 1Hz harmonic decay rate. **(c) Subspace shift (rank)** gradually transforms the 1Hz sine wave into a 1Hz square wave signal using a Fourier decomposition of a square wave signal with increasing order. **(d) Sampling frequency shift** where the system is sampled at different sampling frequencies instead of the reference 200Hz. In each scenario, Koopman operators are estimated from sampled trajectories with the RRR method (Kostic et al., 2022) with rank fixed to twice the number of harmonic oscillators.

**Results & interpretation.** Values of the different metrics as a function of the shifts are shown in Figure 1. In scenarios (a,b,c), our metric SGOT increases continuously (near linearly) with the shifts almost everywhere. In contrast, other similarities tend to saturate quickly, and some even oscillate as shifts increase. In particular, OT-based competitors exhibit extreme behaviors: the pseudo-metric SOT oscillates in the frequency scenario, while GOT saturates fastest overall. Likewise, the Hilbert-Schmidt and operator metrics present both a saturating and an oscillating behavior, introducing many local minima. . When changing the sampling frequency in scenario (d), only GOT and our metric SGOT are robust and remain low and almost constant. Appendix F and G.5 provides details and a sensitivity analysis of the $\eta$ parameter in SGOT.

### 4.2 MACHINE LEARNING OF DYNAMICAL SYSTEMS

**Experimental setup.** We now illustrate and study the usability of our metric SGOT in machine learning applications, both unsupervised (dimensionality reduction) and supervised (classification), when sequential data are embedded by estimated operators governing their dynamics. In both experiments, we considered 14 multivariate time series datasets from the UEA database (Ruiz et al., 2021), and time series are represented with Koopman operators estimated with the RRR method (Kostic et al., 2022).

**Dimensionality reduction.** We first explore the dimensionality reduction capabilities of the different similarity measures. Considering a linear kernel, 5 datasets and all similarities, the estimated operators are embedded as 2D vector with the T-distributed Stochastic Neighbor Embedding (T-SNE) (Maaten & Hinton, 2008) method fitted on the cross-distance matrix estimated with the similarity.

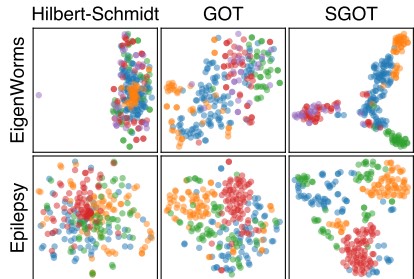

Figure 2: T-SNE embeddings. Datasets on rows, metrics on columns, classes in colors.

Table 1: Classification rank per kernel type. Deep: kernel based on learned deep features. **Best** and second best performers are highlighted (lower is better). Ranks are denoted: $\langle$mean$\rangle \pm \langle$std$\rangle$.

|  | Hilbert-Schmidt | Operator | Martin | SOT | GOT | SGOT |
|---|---|---|---|---|---|---|
| Linear | $3.29 \pm 1.02$ | $3.92 \pm 1.1$ | $5.30 \pm 1.31$ | $4.49 \pm 1.15$ | $2.66 \pm 1.18$ | **$1.34 \pm 0.79$** |
| RBF | $3.74 \pm 1.27$ | NA | $4.02 \pm 0.98$ | $3.28 \pm 1.15$ | $2.48 \pm 1.19$ | **$1.48 \pm 0.70$** |
| Deep | $3.33 \pm 1.56$ | $4.14 \pm 1.27$ | $5.06 \pm 1.48$ | $3.84 \pm 1.34$ | $2.94 \pm 1.33$ | **$1.71 \pm 0.77$** |

Table 2: Classification accuracy for operators estimated with RBF kernels. Datasets on rows and similarities on columns. **Best** and second best performers are highlighted. Accuracy scores are denoted: $\langle$mean$\rangle \pm \langle$std$\rangle$.

|  | Hilbert-Schmidt | Martin | SOT | GOT | SGOT |
|---|---|---|---|---|---|
| BasicMotions | $0.26 \pm 0.17$ | $0.77 \pm 0.06$ | $0.87 \pm 0.05$ | $0.69 \pm 0.14$ | **$0.95 \pm 0.02$** |
| ERing | $0.74 \pm 0.07$ | $0.22 \pm 0.05$ | $0.38 \pm 0.05$ | $0.96 \pm 0.01$ | **$0.98 \pm 0.02$** |
| Epilepsy | $0.31 \pm 0.02$ | $0.80 \pm 0.01$ | $0.77 \pm 0.02$ | $0.93 \pm 0.02$ | **$0.95 \pm 0.02$** |
| FingerMovements | $0.53 \pm 0.06$ | $0.50 \pm 0.03$ | **$0.53 \pm 0.05$** | $0.50 \pm 0.06$ | $0.53 \pm 0.01$ |
| NATOPS | $0.59 \pm 0.06$ | $0.25 \pm 0.02$ | $0.35 \pm 0.02$ | $0.78 \pm 0.03$ | **$0.80 \pm 0.05$** |

Figure 2 illustrates the embeddings for the most discrim-
inative similarities on datasets *EigenWorms* (motion) and *Epilepsy* (biomedical). TSNE embedding for all 5 datasets and metrics are available in Appendix G.2, Figure 8. The Hilbert-Schmidt distance is too conservative, and no clusters or classes can be identified. For OT-based metrics, GOT better identifies classes; however, they do not form distinct clusters as is obtained with our metric SGOT.

**Classification setup.** We now quantify similarities' performances on a classification task. We consider three types of kernels for operator estimation via RRR Kostic et al. (2022): linear, RBF, and kernels based on learned deep features. For each kernel type and dataset, we run a Monte-Carlo nested cross-validation procedure with a (0.7, 0.3) train/test split and no data preprocessing. We perform 10 iterations for the linear case and 5 iterations for the RBF and deep-feature cases. For every similarity measure, we train a $k$-NN classifier and tune the hyperparameters—$k$ (and $\eta$ for SGOT) with a 5-fold inner cross-validation. The RBF experiments are restricted to the five smallest datasets, and the operator metric is excluded due to computational constraints. For the deep-feature experiments, an additional preprocessing step is required: features are learned from the training data using a Multi-Layer Perceptron (MLP), following the strategy of Kostic et al. (2024b) for learning invariant representations of time-homogeneous stochastic dynamical systems. Further details on the experimental protocol are provided in Appendix G.

**Classification results.** Table 1 presents the average rank per metric/kernel combination, while Table 2 provides the accuracy scores in the RBF setting and Figure 3 compares the accuracy between our metric SGOT and other metrics in the linear case. Overall, SGOT consistently outperforms other metrics across most datasets and for any kernel type. These results indicate that SGOT is a robust and well-behaved metric for operator comparison, independent of the underlying estimation method. In particular, SGOT surpasses both SOT and GOT by jointly leveraging information from eigenvalues and eigensubspaces. Detailed results for each kernel type, including comparison plots, full performance tables, critical-difference diagrams, and execution times, are provided in Appendix G.2, G.3, and G.4.

### 4.3 BARYCENTERS AND INTERPOLATION OF DYNAMICAL SYSTEMS

**Interpolation between 1D DS.** In this experiment, we compare the interpolation between dynamical systems through the weighted Fréchet barycenters of their Koopman operators, estimated with a linear kernel, for different metrics. The two systems are linear oscillatory systems, each being the sum of two simple harmonic oscillators with different frequencies and decay rates, and additive Gaussian noise. The interpolation is controlled by a ratio parameter $\gamma \in [0, 1]$ with weights $(1 - \gamma, \gamma)$ in the Fréchet mean problem equation 7. We compare (a) the Hilbert-Schmidt metric without spectral decomposition constraints given by $\mathbf{T}_{bar} = (1 - \gamma)\mathbf{T}^{(0)} + \gamma\mathbf{T}^{(1)}$, (b) the Hilbert-Schmidt metric with spectral decomposition constraints, and (c) our proposed metric SGOT. For (b) and

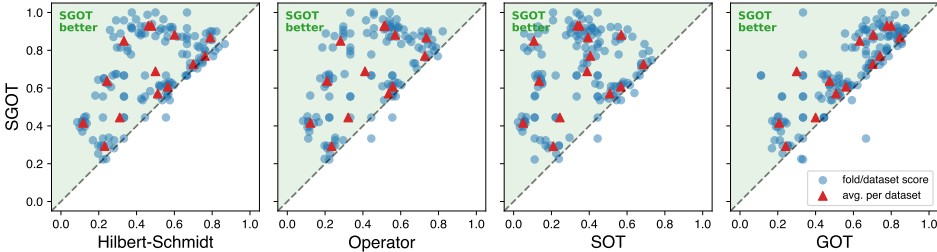

Figure 3: Classification performance (accuracy) comparison between SGOT and competitive metrics. Each point represents a dataset accuracy, with SGOT on the y-axis and the competing metrics on the x-axis.

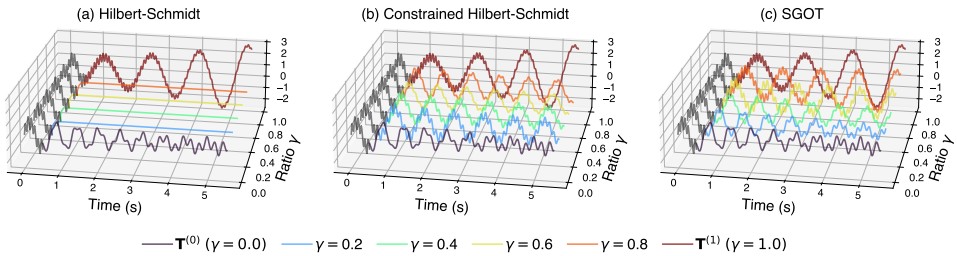

Figure 4: Predictions of interpolated systems between two linear oscillating systems from the same initialization. Interpolated systems correspond to weighted Fréchet barycenter for three different metrics: (a) Hilbert-Schmidt, (b) Hilbert-Schmidt with spectral decomposition constraints, and (c) our metric SGOT. The interpolation is controlled by a ratio parameter $\gamma \in [0, 1]$ which sets operators' weights.

(c), barycentric operators are estimated with the proposed optimization scheme, and experimental settings are detailed in Appendix H.

The interpolated predictions, starting from an identical initialization signal (in gray) containing all four frequencies, are illustrated in Figure 4 for all three metrics. In the Hilbert-Schmidt case (fig. 4.a) leads to overdamped systems $\forall \gamma \in (0, 1)$. Adding spectral decomposition constraints on the Hilbert-Schmidt barycenter (fig. 4. (b) mitigates the damping effects; however, the oscillatory frequencies and decay rate converge to a local minimum close to initialization, as expected by the saturating behavior of the Hilbert-Schmidt metric (see fig. 1). Only SGOT barycenters naturally interpolate between the two systems, notably by retrieving the frequencies and the decay rates.

**Interpolating fluid dynamics.** We aim to compute the barycenter of two fluid dynamics systems. To that end, we consider the *Flow past a bluff object* dataset (Tali et al., 2025), which gathers trajectories of time-varying 2D velocity and pressure fields of incompressible Navier-Stokes fluids flowing around static objects. We select two trajectories, one with a cylinder object and the other with a triangular object. We only kept the velocity field along the flowing direction for each trajectory, leading to trajectories containing 242 samples of 1024x256 grids, which we down-sampled to grids with a 256x64 resolution. We estimate a Koopman operator with linear kernel using the RRR method from each trajectory: $\mathbf{T}^{(0)}$ for the cylinder and $\mathbf{T}^{(1)}$ for the triangle. The operators are restricted to the fourth leading eigenvalues and eigenfunctions. We compute the SGOT barycenter with the optimization scheme described in Appendix D with an initialization being the average of eigenvalues and eigenfunctions. In Appendix H we detail the experimental settings.

Figure 5 illustrates the non-conjugated right eigenfunctions of all three Koopman operators (cylinder, barycenter, triangle). By symmetry of boundary conditions and the cylinder, the eigenfunctions in the cylinder case have an axial symmetry that is lost with the triangle. SGOT by interpolating between both introduces the asymmetry in the eigenfunctions of the barycenter.

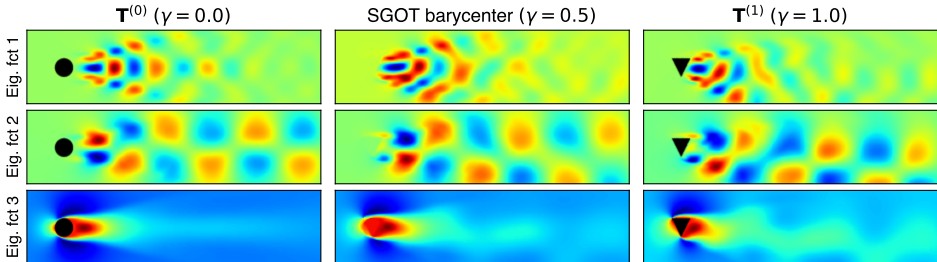

Figure 5: SGOT barycenter of Koopman operators of flows past static objects: a cylinder $\mathbf{T}^{(0)}$ and a triangle $\mathbf{T}^{(1)}$. Each operator's three leading right eigenfunctions are displayed and can be associated with the vortex-shedding phenomenon of the fluids flowing from left to right.

## 5 CONCLUSION

In this paper, we proposed SGOT, a novel optimal transport metric between distributional representations of transfer operators in the joint spectral–Grassmann space. The metric has strong theoretical properties, induces a meaningful geometry for barycenters and interpolation, and can be computed efficiently. Numerical experiments demonstrate the superiority of the proposed metric for machine learning tasks and system interpolation. Our method opens the door to machine learning applications on dynamical systems, with future work including dictionary learning and conditional prediction to accelerate numerical simulations.

## 6 ACKNOWLEDGMENTS

This project received funding from the European Union's Horizon Europe research and innovation program under grant agreement 101120237 (ELIAS), Fondation de l'Ecole Polytechnique, Hi! PARIS, the French National Research Agency (ANR) through France 2030 program (ANR-23-IACL-0005 and ANR-25-PEIA-0005), NextGenerationEU and MUR PNRR project PE0000013 CUP J53C22003010006 "Future Artificial Intelligence Research (FAIR)"

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

## A    RELATED WORK

**Metric for linear dynamical systems.**    A substantial body of research addresses the comparison of (stochastic) linear dynamical systems (LDSs) and linear state-space models (Afsari & Vidal, 2014). Early methods exploit the Riemannian manifold structure of LDS spaces to define meaningful metrics (Hanzon & Marcus, 1982), with related developments in power spectral density spaces (Georgiou, 2007), including approaches based on Wasserstein metrics (Gray, 2009). However, these methods suffer from high computational cost. The Martin distance (Martin, 2002) offers a practical alternative, comparing ARMA models via their cepstrum. It has been generalized to state-space models and shown equivalent to metrics based on angles between observability subspaces (De Cock & De Moor, 2002; Sinha et al., 2024). Other approaches include kernel-based metrics derived from the Binet-Cauchy theorem (Vishwanathan et al., 2007), Kullback–Leibler divergence (Chan & Vasconcelos, 2005), and moment matching (Bissacco et al., 2007). However, compared to the Martin distance, these metrics are sensitive to trajectory initial conditions, so extensions have been proposed to address this issue (Chaudhry & Vidal, 2013).

**Extension to nonlinear dynamical systems.** For nonlinear dynamical systems, most work leverages the Koopman framework to linearize dynamics. The Binet-Cauchy kernel has been extended to nonlinear systems (Fujii et al., 2017) within this context. Another kernel leverages Koopman representation to compare observability subspaces (Ishikawa et al., 2018). The latest has been used alongside a deep learning method for estimating Koopman operators (ResDMD Colbrook et al. (2023)) in the case of continuous spectrum (Sakata & Kawahara, 2024). However, both kernels are sensitive to trajectory initial conditions like the linear case. In Mezić & Banaszuk (2004), the authors propose metrics to compare the asymptotic dynamics of measure-preserving systems via Koopman representations, later extended to dissipative systems over finite time (Mezic, 2016).

**Metric for topologically conjugated dynamical systems.** Recently, interest has grown in comparing neural network dynamics in neuroscience and deep learning (Klabunde et al., 2025). Such comparisons often consider topological conjugacy, leading to metrics on quotient spaces. Redman et al. (2022; 2024) show that topologically conjugate systems share identical Koopman spectra and propose a pseudo-metric based on optimal transport. Ostrow et al. (2023) extends Procrustes analysis to compare Koopman representations up to orthogonal transformations, extending earlier work in the LDS setting (Afsari & Vidal, 2013). Glaz (2025) further generalizes these metrics to accommodate broader transformation classes.

**Optimal transport on functional spaces.** A related direction studies measures on functional spaces. Some works have studied measures on Gaussian processes (Masarotto et al., 2019; Mallasto & Feragen, 2017), for which there exists a closed-form of the metric. In Antonini & Cavalletti (2021), the authors propose a theoretical Wasserstein metric between measures derived from the spectral decomposition of normal operators. More recently, Zhu et al. (2024) introduced a computable approximation of the Wasserstein metric between measures on infinite-dimensional Hilbert spaces, obtained by restriction to linear mappings.

## B    LEARNING KOOPMAN TRANSFER OPERATORS WITH KERNEL METHODS

In many practical scenarios, $A_\pi$ is unknown, but data from system trajectories are available. For such cases, Koopman operator regression in reproducing kernel Hilbert spaces (RKHS) provides a learning framework to estimate $A_\pi$ on $\mathcal{L}^2_\pi(\mathcal{X})$ (Kostic et al., 2022). Let $\mathcal{H}$ be a RKHS with a bounded kernel $k$ and feature map $\phi$ such that $k(x,y) = \langle \phi(x), \phi(y) \rangle$. We recall that the injection operator $S_\pi : \mathcal{H} \to \mathcal{L}^2_\pi(\mathcal{X})$ is Hilbert-Schmidt (Caponnetto & De Vito, 2007; Steinwart & Christmann, 2008), and thus so is the restricted Koopman operator $Z_\pi := A_\pi S_\pi : \mathcal{H} \to \mathcal{L}^2_\pi(\mathcal{X})$.

The goal is to approximate $Z_\pi = A_\pi S_\pi$ by minimizing the risk $\mathcal{R}(G) = \mathbb{E}_{x\sim\pi} \sum_{i\in\mathbb{N}} \mathbb{E}\big[(h_i(X_{t+1}) - (Gh_i)(X_t))^2 | X_t = x\big]$ over Hilbert-Schmidt operators $G \in \mathrm{HS}(\mathcal{H})$, where $(h_i)_{i\in\mathbb{N}}$ is an orthonormal basis of $\mathcal{H}$. This risk admits a decomposition $\mathcal{R}(G) = \mathcal{R}_0 + \mathcal{E}_{\mathrm{HS}}(G)$, where

$$\mathcal{R}_0 = \|S_\pi\|^2_{\mathrm{HS}} - \|Z_\pi\|^2_{\mathrm{HS}} \geq 0 \quad \text{and} \quad \mathcal{E}_{\mathrm{HS}}(G) = \|A_\pi S_\pi - S_\pi G\|^2_{\mathrm{HS}} = \|A_\pi S_\pi - S_\pi G\|^2_{\mathrm{HS}(\mathcal{H}, \mathcal{L}^2_\pi(\mathcal{X}))} \tag{10}$$

are the irreducible risk and the excess risk, respectively. Using universal kernels, the excess risk can be made arbitrarily small: $\inf_{G\in\mathrm{HS}(\mathcal{H})} \mathcal{E}_{\mathrm{HS}}(G) = 0$.

A common approach is to solve the Tikhonov-regularized problem

$$\min_{G\in\mathrm{HS}(\mathcal{H})} \mathcal{R}^\gamma(G) := \mathcal{R}(G) + \gamma\|G\|^2_{\mathrm{HS}}, \tag{11}$$

with $\gamma > 0$. Defining the covariance operator $C_x := S^*_\pi S_\pi = \mathbb{E}_{x\sim\pi}\phi(x) \otimes \phi(x)$ and the cross-covariance operator $C_{xy} := S^*_\pi Z_\pi = \mathbb{E}_{(x,y)\sim\rho}\phi(x) \otimes \phi(y)$ (where $\rho$ is the joint measure of consecutive states), the unique solution to equation 11 is the Kernel Ridge Regression (KRR) estimator $G_\gamma := C^{-1}_\gamma C_{xy}$, where $C_\gamma := C_x + \gamma \mathrm{Id}_{\mathcal{H}}$.

To approximate the leading eigenvalues of $A_\pi$, low-rank estimators are used. The Reduced Rank Regression (RRR) estimator (Kostic et al., 2022) is the solution to equation 11 under a rank-$r$ constraint:

$$C^{-1/2}_\gamma [\![C^{-1/2}_\gamma C_{xy}]\!]_r = \underset{G\in\mathrm{B}_r(\mathcal{H})}{\arg\min} \mathcal{R}^\gamma(G), \tag{12}$$

where $\mathrm{B}_r(\mathcal{H})$ denotes the set of rank-$r$ HS operators and $[\![\cdot]\!]_r$ is the $r$-truncated SVD.

Given data $\mathcal{D} = \{(x_i, y_i)\}_{i \in [n]}$, empirical estimators are derived by minimizing the regularized empirical risk $\widehat{\mathcal{R}}^\gamma(G) := \frac{1}{n} \sum_{i \in [n]} \|\phi(y_i) - G^* \phi(x_i)\|_2^2 + \gamma \|G\|_{\mathrm{HS}}^2$. Introducing the sampling operators for data $\mathcal{D}$ and RKHS $\mathcal{H}$ by

$$\widehat{S} \colon \mathcal{H} \to \mathbb{R}^n \quad \text{s.t. } f \mapsto \tfrac{1}{\sqrt{n}} [f(x_i)]_{i \in [n]} \quad \text{and} \quad \widehat{Z} \colon \mathcal{H} \to \mathbb{R}^n \quad \text{s.t. } f \mapsto \tfrac{1}{\sqrt{n}} [f(y_i)]_{i \in [n]},$$

and their adjoints by

$$\widehat{S}^* \colon \mathbb{R}^n \to \mathcal{H} \quad \text{s.t. } w \mapsto \tfrac{1}{\sqrt{n}} \sum_{i \in [n]} w_i \phi(x_i) \quad \text{and} \quad \widehat{Z}^* \colon \mathbb{R}^n \to \mathcal{H} \quad \text{s.t. } w \mapsto \tfrac{1}{\sqrt{n}} \sum_{i \in [n]} w_i \psi(y_i),$$

we obtain $\widehat{\mathcal{R}}^\gamma(G) = \|\widehat{Z} - \widehat{S}G\|_{\mathrm{HS}}^2 + \gamma \|G\|_{\mathrm{HS}}^2$.

The empirical covariance and cross-covariance operators are:

$$\widehat{C}_x := \widehat{S}^* \widehat{S}, \quad \widehat{D} := \widehat{Z}^* \widehat{Z}, \quad \widehat{C}_{xy} := \widehat{S}^* \widehat{Z}. \tag{13}$$

The corresponding regularized empirical covariance is $\widehat{C}_\gamma := \widehat{C}_x + \gamma \operatorname{Id}_{\mathcal{H}}$. The kernel Gram matrices are:

$$K := \widehat{S}\widehat{S}^*, \quad L := \widehat{Z}\widehat{Z}^*. \tag{14}$$

The empirical RRR estimator is then $\widehat{C}_\gamma^{-1/2} [\![\widehat{C}_\gamma^{-1/2} \widehat{C}_{xy}]\!]_r$. These empirical estimators can be expressed in the form $\widehat{G} = \widehat{S} U_r V_r^\top \widehat{Z}$ for matrices $U_r, V_r \in \mathbb{R}^{n \times r}$ (Kostic et al., 2022), enabling the computation of spectral decompositions in infinite-dimensional RKHS.

**Theorem 1** ((Kostic et al., 2022)). *Let $1 \leq r \leq n$ and $\widehat{G} = \widehat{S} U_r V_r^\top \widehat{Z}$, where $U_r, V_r \in \mathbb{R}^{n \times r}$. If $V_r^\top M U_r \in \mathbb{R}^{r \times r}$, for $M = n^{-1}[k(y_i, x_j)_{i,j \in [n]}]$, is full rank and non-defective, the spectral decomposition $(\widehat{\lambda}_i, \widehat{\xi}_i, \widehat{\psi}_i)_{i \in [r]}$ of $\widehat{G}$ can be expressed in terms of the spectral decomposition $(\widehat{\lambda}_i, \widehat{u}_i, \widehat{v}_i)_{i \in [r]}$ of $V_r^\top M U_r$ as $\widehat{\xi}_i = \widehat{\lambda}_i \widehat{Z}^* V_r \widehat{u}_i / |\widehat{\lambda}_i|$ and $\widehat{\psi}_i = \widehat{S}^* U_r \widehat{v}_i$, for all $i \in [r]$.*

**RKHS embedddings into $\mathcal{L}_\pi^2(\mathcal{X})$.** We recall some facts on the injection operator $S_\pi$. Note first that $S_\pi \in \mathrm{HS}\left(\mathcal{H}, \mathcal{L}_\pi^2(\mathcal{X})\right)$. Then according to the spectral theorem for positive self-adjoint operators, $S_\pi$ has an SVD, i.e. there exists at most countable positive sequence $(\sigma_j)_{j \in J}$, where $J := \{1, 2, \ldots, \} \subseteq \mathbb{N}$, and ortho-normal systems $(\ell_j)_{j \in J}$ and $(h_j)_{j \in J}$ of $\mathrm{cl}(\mathrm{Im}(S_\pi))$ and $\mathrm{Ker}(S_\pi)^\perp$, respectively, such that $S_\pi h_j = \sigma_j \ell_j$ and $S_\pi^* \ell_j = \sigma_j h_j$, $j \in J$.

Now, given $\alpha \geq 0$, let us define scaled injection operator $S_\alpha \colon \mathcal{H} \to \mathcal{L}_\pi^2(\mathcal{X})$ as

$$S_\alpha := \sum_{j \in J} \sigma_j^\alpha \ell_j \otimes h_j. \tag{15}$$

Clearly, we have that $S_\pi = S_1$, while $\mathrm{Im}\, S_0 = \mathrm{cl}(\mathrm{Im}(S_\pi))$. Next, we equip $\mathrm{Im}(S_\alpha)$ with a norm $\|\cdot\|_\alpha$ to build an interpolation space:

$$[\mathcal{H}]_\alpha := \left\{ f \in \mathrm{Im}(S_\alpha) \mid \|f\|_\alpha^2 := \sum_{j \in J} \sigma_j^{-2\alpha} \langle f, \ell_j \rangle^2 < \infty \right\}.$$

## C SPECTRAL-GRASSMANN WASSERSTEIN METRIC (SGOT) PROOF

### C.1 MAIN PROOF

In this section we prove that $\mathcal{S}_r(\mathcal{H})$ can be endowed with a Wasserstein metric based on operator spectral decomposition as summarized by the following theorem:

**Theorem 3.** *Let $\mathcal{H}$ be a separable $\mathbb{C}$-Hilbert space and $\mathcal{S}_r(\mathcal{H})$ the set of non-defective operators with rank at most $r \in \mathcal{D}$. Let $(\mathcal{G}, d_{\mathcal{G}})$ be Grassmanian manifold of the space of Hilbert-Schmidt operators on $\mathcal{H}$. Given $p \in \mathbb{N}^*$ and $\eta \in (0, 1)$, let $\mu \colon \mathcal{S}_r(\mathcal{H}) \to \mathcal{P}_p(\mathbb{C} \times \mathcal{G})$ and $d_\eta \colon (\mathbb{C} \times \mathcal{G})^2 \to \mathbb{R}_+$ be given by*

$$\mu(T) \triangleq \sum_{j \in [\ell]} \tfrac{m_j}{m_{tot}} \delta_{(\lambda_j, \mathcal{V}_j)} \quad \text{and} \quad d_\eta[(\lambda', \mathcal{V}'), (\lambda', \mathcal{V}')] \triangleq \eta |\lambda - \lambda'| + (1 - \eta) d_{\mathcal{G}}(\mathcal{V}, \mathcal{V}'), \tag{16}$$

with $| \cdot |$ applied on polar coordinates $\lambda, \lambda'$, $m_{tot} = \sum_{i \in [\ell]} m_i$, $\mathcal{V}_j$ the $m_j$-dimensional vector space in $\mathrm{HS}(\mathcal{H}, \mathcal{H})$ spanned by the rank one operators of the right/left eigenfunctions associated with the eigenvalue $e^{\lambda_j}$ of $T$ (same notation for $T'$). Then, $(\mathcal{S}_r(\mathcal{H}), d_\mathcal{S})$ is a metric space, where $d_\mathcal{S} \colon \mathcal{S}_r(\mathcal{H}) \to \mathbb{R}_+$ is given by

$$d_\mathcal{S}(T, T') = W_{d_\eta, p}(\mu(T), \mu(T')). \tag{17}$$

**Discrete Optimal transport.** For conciseness, we first recall discrete OT where one seeks a transport plan mapping samples from a source distribution to those of a target distribution while minimizing a transportation cost. Formally, consider $\mathcal{Z}_S = \{z_i \in \mathcal{Z} \mid i \in [k_S]\}$ and $\mathcal{Z}_T = \{z_i' \in \mathcal{Z} \mid i \in [k_T]\}$ as the sets of source and target samples in a space $\mathcal{Z}$. We associate with these sets the probability distributions $\mu_S = \sum_{i \in [k_S]} a_i \delta_{z_i}$ and $\mu_T = \sum_{i \in [k_T]} b_i \delta_{z_i'}$ with $(\mathbf{a}, \mathbf{b}) \in \Delta^{k_S} \times \Delta^{k_T}$ and $\Delta^n = \{\mathbf{p} \in \mathbb{R}_+^n \mid \sum_{i \in [n]} p_i = 1\}$ the $n$-simplex. Let $\mathbf{C} \in \mathbb{R}_+^{k_S \times k_T}$ be the cost matrix with $C_{ij} = c(z_i, z_j')$ being the transport cost between $z_i$ and $z_j'$ given by the cost function $c$. The Monge-Kantorovich problem aims at identifying a coupling matrix, also denoted as OT plan $\mathbf{P}^* \in \mathbb{R}_+^{k_S \times k_T}$, that is solution of the constrained linear problem:

$$\min_{\mathbf{P} \in \Pi(\mu_S, \mu_T)} \langle \mathbf{C}, \mathbf{P} \rangle_F \quad \text{s.t} \quad \Pi(\mu_S, \mu_T) = \{\mathbf{P} \in \mathbb{R}_+^{k_S \times k_T} \mid \mathbf{P}\mathbf{1} = \mathbf{a}, \ \mathbf{P}^\intercal \mathbf{1} = \mathbf{b}\}, \tag{18}$$

where $\Pi(\mathbf{a}, \mathbf{b})$ is the set of joint-distributions over $\mathcal{Z}_S \times \mathcal{Z}_T$ with marginals $\mathbf{a}$ and $\mathbf{b}$. In what follows, we denote $L_c(\mu_S, \mu_T)$ the application returning the optimal value of problem (18) where $c$ indicates the cost function. A fundamental property of OT is that, under suitable conditions on the cost function, the Wasserstein distance is a metric on the space of probability measures:

**Theorem 4** (Theorem 6.18 in Villani et al. (2008)). *Let $(\mathcal{Z}, d)$ be a separable complete metric space endowed with its Borel set. Let $p \in \mathbb{N}^*$, and $\mathcal{P}_p(\mathcal{Z})$ the set of probability distributions on $\mathcal{Z}$ admitting moments of order $p$. Consider the application:*

$$W_p : (\mu, \nu) \in \mathcal{P}(\mathcal{Z}) \times \mathcal{P}(\mathcal{Z}) \mapsto (L_{d^p}(\mu, \nu))^{\frac{1}{p}} \in \mathbb{R}_+ . \tag{19}$$

*Then, $(\mathcal{P}_p(\mathcal{Z}), W_p)$ defines a separable complete metric space, known as a Wasserstein space.*

**Main proof.** For proof correctness, we restrict the Grassmann manifold on Hilbert-Schmidt operators $\mathcal{G}$ to the set of operators with rank at most $r$, denoted by $\mathcal{G}_r$. This restricted space endowed with the Hilbert-Schmidt norm is a complete metric space as detailed in Appendix C.2. The next two propositions detail the essential building blocks to derive a Wasserstein metric on $\mathcal{S}_r(\mathcal{H})$. Proposition 1 specifies an inclusion map from $\mathcal{S}_r(\mathcal{H})$ to a space of probability measures while proposition 2 defines a metric on the measures' support space with sufficient topological properties to derive a Wassertein metric on $\mathcal{S}_r(\mathcal{H})$.

**Proposition 1.** *Consider $p \in \mathbb{N}^*$ and the embedding map:*

$$\mu : T \in S_r(\mathcal{H}) \mapsto \sum_{i \in [l]} \frac{l_i}{l_{tot}} \delta_{(\lambda_i, \mathcal{V}_i)} \in \mathcal{P}_p(\mathbb{C} \times \mathcal{G}_r) , \tag{20}$$

*where $\mathcal{V}_j$ the $m_j$-dimensional vector space in $\mathrm{HS}(\mathcal{H}, \mathcal{H})$ spanned by the rank one operators of the right/left eigenfunctions associated with the eigenvalue $e^{\lambda_j}$ of $T$, and $m_{tot} = \sum_{i \in [l]} m_i$. Then, $L$ is a one-to-one inclusion map.*

*Proof.* Let $T \neq T' \in \mathcal{S}_r(\mathcal{H})$, both operators differ by at least one pair $(\lambda_i, \mathcal{V}_i) \in \mathcal{G}_r$, by symmetry the pair is associated to $T$. Since $(\mathbb{C}, |.|)$ and $(\mathcal{G}_r, d_\mathcal{G})$ are metric spaces, the singleton $\{(\lambda_i, \mathcal{V}_i)\}$ belongs to the Borel set. Therefore, $\mu_T((\lambda_i, \mathcal{V}_i)) = m_i/m_{tot}$ while $\mu_{T'}((\lambda_i, \mathcal{V}_i)) = 0$, i.e $\mu_T \neq \mu_{T'}$. $\qquad\square$

**Proposition 2.** *Consider $\eta \in (0, 1)$, $\omega_{ref} \in \mathbb{R}_+^*$, and the application:*

$$d_\eta : ((\lambda, \mathcal{V}), (\lambda', \mathcal{V}')) \in (\mathbb{C} \times \mathcal{G}_r)^2 \mapsto \eta|\lambda - \lambda'| + (1 - \eta)d_\mathcal{G}(\mathcal{V}, \mathcal{V}') \in \mathbb{R}_+ . \tag{21}$$

*Then, $(\mathbb{C} \times \mathcal{G}_r, d_\eta)$ is a separable complete metric space.*

*Proof.* By proposition 4, $(\mathcal{G}_r, d_\mathcal{G})$ is a separable complete metric space. Hence, for any $\eta \in (0, 1)$, $(\mathbb{C} \times \mathcal{G}_r, d_\eta)$ is a separable complete metric space as $(\mathbb{C}, d_{val})$ is homeomorphic to $(\mathbb{C}, |.|)$. $\qquad\square$

Note that we introduce a metric, $d_{val}$, that compares Koopman modes' eigenvalues from physics-informed quantities, namely the time-scales $\rho$ and the ocsillating frequencies $\omega$. The previous two propositions lead to our main contribution, a Wasserstein metric on the space of non-defective finite rank operators $\mathcal{S}_r(H)$:

**Proposition 3.** *Consider $\eta \in (0, 1)$, $p \in \mathbb{N}^*$, and the application:*

$$d_{\mathcal{S}} : (T, T') \in \mathcal{S}_r(\mathcal{H}) \times \mathcal{S}_r(\mathcal{H}) \mapsto W_{d_\eta, p}(\mu(T), \mu(T')) \in \mathbb{R}_+ . \tag{22}$$

*Then, $(\mathcal{S}_r(\mathcal{H}), d_{\mathcal{S}})$ is a metric space.*

*Proof.* Application of theorem 4 with Propositions 1 and 2. $\qquad\square$

## C.2 GRASSMAN METRIC

A Grassmann manifold is a collection of vector subspaces of a given vector space. Such manifolds appear in a handful of applications whenever subspaces must be compared. The particular case of Grassman manifolds gathering all equidimensional subspaces of a finite-dimensional real vector space has been extensively studied, see Bendokat et al. (2024) for a thorough review. In our context, this particular setting is limiting as we must consider a manifold including subspaces of various dimensions over a possibly infinite-dimensional complex vector space. On such manifolds, a classical metric compares subspaces through the associated orthogonal projectors with the operator norm (Andruchow, 2014). Unfortunately, this metric is computationally expensive, and the topology it induces does not provide the necessary conditions to derive Wasserstein metrics, namely, the separability. In the following proposition, we define a Grassmann manifold with the necessary conditions to derive Wasserstein metrics.

**Proposition 4.** *Let $r \in \mathbb{N}^*$ be fixed, and $\mathcal{G}_r(\mathcal{H})$ denote the set of all closed vector subspaces of a (possibly infinite-dimensional) separable Hilbert space $\mathcal{H}$ having dimension at most $r$. Endow $\mathcal{G}_r(\mathcal{H})$ with the well-defined metric:*

$$d_{\mathcal{G}} : (\mathcal{U}, \mathcal{V}) \in \mathcal{G}_r(\mathcal{H}) \times \mathcal{G}_r(\mathcal{H}) \mapsto \|P_{\mathcal{U}} - P_{\mathcal{V}}\|_{\mathcal{HS}} \in \mathbb{R}_+ , \tag{23}$$

*where $P_{\mathcal{U}}$ is the orthogonal projector onto $\mathcal{U}$, and $\|.\|_{\mathcal{HS}}$ is the Hilbert-Schmidt norm. Then $(\mathcal{G}_r(\mathcal{H}), d_{\mathcal{G}})$ is a separable complete metric space.*

*Proof.* Before the main proof, we investigate the properties of an inclusion map, which is useful for determining the metric and completeness properties.

**Lemma 1.** *The map $i : \mathcal{V} \in \mathcal{G}_r(\mathcal{H}) \mapsto P_{\mathcal{V}} \in \mathcal{HS}(\mathcal{H})$, which associates to any subspace the orthogonal projector onto itself, is well defined and a one-to-one inclusion.*

*Proof.* Since any $\mathcal{V} \in \mathcal{G}_r(\mathcal{H})$ is finite dimensional, it is closed, and the orthogonal projector $P_{\mathcal{V}}$ is a well-defined bounded linear operator by consequence of the Hilbert projection theorem. Furthermore, since $\mathcal{H}$ is separable, it admits an orthogonal basis, respecting the orthogonal decomposition $\mathcal{H} = \mathcal{V} \oplus \mathcal{V}^\perp$. Since $\dim(\mathcal{V}) \leq r$ and by invariance of the Hilbert-Schmidt norm to change of basis, $\|P_{\mathcal{V}}\|_{\mathcal{HS}}$ is finite, more precisely: $\|P_{\mathcal{V}}\|_{\mathcal{HS}}^2 = \dim(\mathcal{V}) < r$. Furthermore, for any $\mathcal{V} \neq \mathcal{V}' \in \mathcal{G}_r(\mathcal{H})$, $P_{\mathcal{V}} \neq P_{\mathcal{V}'}$ due to the orthogonal decomposition $\mathcal{H} = \mathcal{V} \cap \mathcal{V}' \oplus \mathcal{V}/(\mathcal{V} \cap \mathcal{V}') \oplus \mathcal{V}'/(\mathcal{V} \cap \mathcal{V}') \oplus (\mathcal{V} \cup \mathcal{V}')^\perp$. $\qquad\square$

**Lemma 2.** *$\mathcal{P}_r = \{P_{\mathcal{V}} \mid \mathcal{V} \in \mathcal{G}_r(\mathcal{H})\}$ is a closed subspace of $\mathcal{HS}(\mathcal{H})$ for the topology induced by the Hilbert-Schmidt norm.*

*Proof.* First notice that, $\mathcal{P}_r \subset \mathcal{HS}(\mathcal{H})$ and $\mathcal{HS}(\mathcal{H})$ is a Hilbert space, thus complete. Consider a sequence $(P_n)_{n \in \mathbb{N}} \in \mathcal{P}_r$ converging to an element $P \in \mathcal{HS}(\mathcal{H})$ (i.e. $\|Pn - p\|_{\mathcal{HS}} \to 0$), let's prove that $P \in \mathcal{P}_r$.

Since $P \in \mathcal{HS}(\mathcal{H})$, it follows that the adjoint operator $P^* \in \mathcal{HS}(\mathcal{H})$ exists and since $\|P^* - P_n^*\|_{\mathcal{HS}} = \|P - P_n\|_{\mathcal{HS}} \to 0$, the operator $P$ is self-adjoint $P = P^*$. Furthermore by composition $P^2 \in \mathcal{HS}(\mathcal{H})$, and:

$$\|P^2 - P\|_{\mathcal{HS}} \leq \|P^2 - P_n^2\|_{\mathcal{HS}} + \|P_n^2 - P_n\|_{\mathcal{HS}} + \|P_n - P\|_{\mathcal{HS}} \tag{24}$$

$$\leq \|P^2 - P_n^2\|_{\mathcal{HS}} + \|P_n - P\|_{\mathcal{HS}} \tag{25}$$

$$\leq \|P_n - P\|_{\mathcal{HS}}(1 + \|P\|_{\mathcal{HS}} + \|P_n\|_{\mathcal{HS}}) \tag{26}$$

Since $\|P_n - P\|_{\mathcal{HS}} \to 0$, it follows that $P^2 = P$, meaning that $P$ is an orthogonal projector. Let $\mathcal{V}$ denote the closed vector subspace associated to $P$. Since $P$ is an orthogonal projector with a finite Hilbert-Schmidt norm, $\mathcal{V}$ is finite dimentional, and $\dim(\mathcal{V}) = \|P\|_{\mathcal{HS}}^2 \leq r$, as $\|P_n\|_{\mathcal{HS}}^2 \leq r$ for any $n \in \mathbb{N}$. Thus $P \in \mathcal{P}_r$, indicating that $\mathcal{P}_r$ is a closed subset of $\mathcal{HS}(\mathcal{H})$. $\qquad\square$

**Main proof.** Since the map $i$, defined in Lemma 1, is a one-to-one inclusion into the space $\mathcal{HS}(\mathcal{H})$, the metric derived from the Hilbert-Schmidt norm ($\|.\|_{\mathcal{HS}}$) induces a metric onto the space $\mathcal{G}_r(\mathcal{H})$. Furthermore, since $\mathcal{P}_r = \{P_\mathcal{V} \mid \mathcal{V} \in \mathcal{G}_r(\mathcal{H})\}$ is a closed subset of a complete space by Lemma 2, it is complete. Hence, the metric space $(\mathcal{G}_r(\mathcal{H}), d_\mathcal{G})$ is complete. Lastly, the space $\mathcal{HS}(\mathcal{H})$ is separable as it is homeomorphic to $\mathcal{H} \otimes \mathcal{H}$, which is a separable space as the tensor product of the separable space $\mathcal{H}$. Hence $\mathcal{P}_r \subset \mathcal{HS}(\mathcal{H})$ is also separable by inclusion. Finally, the metric space $(\mathcal{G}_r(\mathcal{H}), d_\mathcal{G})$ is separable and complete, which concludes the proof.

$\qquad\square$

### C.3 Alternative metrics on Grassmanm manifold

Table 3: Metrics on Grassmann manifold $\mathcal{G}(k, n)$: angle-based and matrix-based formulations. Here $\mathbf{M} = \mathbf{U}^\top \mathbf{V}$, $\mathbf{P} = \mathbf{U}\mathbf{U}^\top$, $\mathbf{Q} = \mathbf{V}\mathbf{V}^\top$, and $S = \sqrt{\mathbf{M}\mathbf{M}^\top}$ where $\mathbf{U}, \mathbf{V}$ are orthonormal bases.

| Metric | Angle formulation | Matrix formulation |
|---|---|---|
| Geodesic (canonical) | $d_{\text{geo}} = \left( \sum_{i=1}^{k} \theta_i^2 \right)^{1/2}$ | $d_{\text{geo}} = \left\| \arccos(\mathbf{S}) \right\|_F$ |
| Chordal | $d_{\text{chord}} = \left( \sum_{i=1}^{k} \sin^2 \theta_i \right)^{1/2}$ | $d_{\text{chord}} = \frac{1}{\sqrt{2}} \|\mathbf{P} - \mathbf{Q}\|_F$ |
| Procrustes | $d_{\text{proc}} = \left( 2k - 2 \sum_{i=1}^{k} \cos \theta_i \right)^{1/2}$ | $d_{\text{proc}} = \sqrt{2k - 2\operatorname{tr}(\mathbf{S})}$ |
| Binet–Cauchy | $d_{\text{BC}} = \sqrt{1 - \prod_{i=1}^{k} \cos^2 \theta_i}$ | $d_{\text{BC}} = \sqrt{1 - \det(\mathbf{M})^2}$ |
| Martin | $d_{\text{Martin}} = \sqrt{-\sum_{i=1}^{k} \log(\cos^2 \theta_i)}$ | $d_{\text{Martin}} = \sqrt{-\log\det(\mathbf{M}\mathbf{M}^\top)}$ |
| Fubini–Study | $d_{\text{FS}} = \arccos\left( \prod_{i=1}^{k} \cos \theta_i \right)$ | $d_{\text{FS}} = \arccos(|\det(\mathbf{M})|)$ |
| Spectral (max) | $d_{\max} = \max_i \theta_i$ | $d_{\max} = \|\arccos(\mathbf{S})\|_2$ |
| Nuclear (sum) | $d_{\text{nuc}} = \sum_{i=1}^{k} \theta_i$ | $d_{\text{nuc}} = \operatorname{tr}(\arccos(\mathbf{S}))$ |

**Principal angle definition.** Considering the real vector space $(\mathbb{R}^n, \langle ., . \rangle)$, let $\mathcal{G}(k, n)$ denotes the Grassmann manifold of all vector subspace of dimension $k$. Many metrics on the Grassmann manifold are based on the principal angles between subspaces. Formally, let $\mathcal{U}, \mathcal{V} \subset \mathbb{R}^n$ be two $k$-dimensional subspaces, the *principal angles*, $0 \leq \theta_1 \leq \theta_2 \leq \cdots \leq \theta_k \leq \frac{\pi}{2}$, between $\mathcal{U}$ and $\mathcal{V}$ are defined recursively by

$$\cos \theta_i = \max_{\substack{u \in \mathcal{U},\, v \in \mathcal{V} \\ \|u\|=\|v\|=1}} u^\top v, \quad \text{s.t} \left\{ \begin{array}{l} u^\top u_j = 0 \\ v^\top v_j = 0 \end{array} \right., \quad \forall j \in [i-1],$$

where $u_j$ and $v_j$ are the previously chosen principal vectors.

**Computation.** Let $\mathbf{U}, \mathbf{V} \in \mathbb{R}^{n \times k}$ be orthonormal basis matrices of $\mathcal{U}, \mathcal{V}$. Let $\mathbf{M} = \mathbf{U}^\top \mathbf{V}$ and $\mathbf{W} \boldsymbol{\Sigma} \mathbf{Z}^\top$ be the singular decomposition of $\mathbf{M}$, the the principal angles are given by

$$\cos \theta_i = \sigma_i(\mathbf{M}) = \boldsymbol{\Sigma}_{ii}, \quad \forall i \in [K],$$

and the corresponding principal vectors are

$$\mathbf{u}_i = \mathbf{U}\mathbf{w}_i, \quad \mathbf{v}_i = \mathbf{V}\mathbf{z}_i,$$

where $\mathbf{w}_i$ and $\mathbf{z}_i$ are the $i$-th left and right singular vectors of $\mathbf{M}$.

**Metric based on principal angles.** Table 3 provides a non-exhaustive list of metrics to compare subspaces of identical dimensions from principal angles. For each metric, angle-based and matrix-based formulations are provided. These metrics can be extended to compare subspaces of different dimensions following the methods proposed in Ye & Lim (2016).

## D SPECTRAL GRASSMAN BARYCENTER

### D.1 PROBLEM FORMULATION

Computing barycenters is a fundamental problem for many unsupervised methods. When data lie in a metric space, it is known as the *Fréchet mean problem*. It involves identifying an element that minimizes a weighted sum of distances to the observations. Formally, given the importance weights $\boldsymbol{\gamma} \in \Delta^N$, assuming (A1)-(A3), for $p=2$ in Theorem 1 we aim to solve:

$$\arg\min_{T \in \mathcal{S}_r(\mathcal{H})} \sum_{k \in [N]} \gamma_i d_{\mathcal{S}}(T, T_k)^2, \tag{27}$$

By construction of $d_{\mathcal{S}}$, problem 7 corresponds to the estimation of Wasserstein barycenter over a set of finite measures with support on a manifold embedded in a (possibly infinite-dimensional) Hilbert space. From a theoretical standpoint, the existence (and uniqueness) of Wasserstein barycenters has been established in several settings, including continuous measures (Agueh & Carlier, 2011), and discrete measures on finite-dimensional Euclidean spaces (Anderes et al., 2016), and measures on geodesic spaces (Le Gouic & Loubes, 2017). In Han et al. (2024), the authors address the case of continuous measures on infinite-dimensional metric spaces. In our settings, we assume the existence of a barycenter in the closure of $\mathcal{S}_r(\mathcal{H})$, see discussion on the extension to general operators in section 3. A formal proof would require extending previous works to finite measures on manifolds in infinite-dimensional Hilbert spaces.

From a computational standpoint, problem 27 is closely related to the *free-support Wassertein barycenter* estimation, which aims at optimizing the support and, optionally, the mass of the atoms parametrizing the barycenter. State-of-the-art algorithms typically rely on a coordinate descent scheme (Wright, 2015), alternating between transport plan computation and measure optimization with strategies including gradient descent (Cuturi & Doucet, 2014), fixed point iteration (Álvarez-Esteban et al., 2016; Lindheim, 2023), stochastic optimization (Claici et al., 2018; Li et al., 2020), or proximal operators (Qian & Pan, 2021). In our context, on the measure's support, i.e., the barycenter's spectral decomposition, must be optimized as the eigensubspaces' dimensions condition the masses according to the embedding map in eq. (16).

**A parametric problem formulation.** Whenever the RKHS $\mathcal{H}$ is infinite dimensional (the finite case is discussed in appendix D.3), the Fréchet mean problem (eq. (27)) is intractable in its original form. We restrained the optimization over a set of parametrized operators defined such that for any $\boldsymbol{\theta} \triangleq (\boldsymbol{\lambda}, \boldsymbol{\alpha}, \boldsymbol{\beta}, \mathbf{x})$:

$$T_{\boldsymbol{\theta}} : h \in \mathcal{H} \mapsto \sum_{i \in [r]} \lambda_i \langle \kappa \boldsymbol{\alpha}_i, h \rangle_{\mathcal{H}} \kappa \boldsymbol{\beta}_i \in \mathcal{H} \tag{28}$$

where $\boldsymbol{\lambda} \in \mathbb{C}^r$, $\mathbf{x} \in \mathcal{X}^n$ are state space control points, and $\boldsymbol{\alpha}, \boldsymbol{\beta} \in \mathbb{C}^{n \times r}$ control parameters acting on the representer functions $\kappa_{\mathbf{x}} = \{\kappa(., x_j)\}_{j \in [n]}$ with $\kappa$ the kernel of $\mathcal{H}$, i.e. $\kappa_{\mathbf{x}} \boldsymbol{\alpha}_i \triangleq \sum_{j \in [n]} \kappa_{x_j} \alpha_{ji}$. While these operators are compact with rank at most $r$, further constraints on the control points and parameters are required to ensure a spectral decomposition (see Equation (2)). Together with the definition of discrete optimal transport (see Section 2), it leads to the constrained optimization problem:

$$\arg\min_{\boldsymbol{\theta}, \mathbf{P}} \sum_{i \in [N]} \gamma_i \langle \mathbf{C}_i(\boldsymbol{\theta}), \mathbf{P}_i \rangle_F \quad \text{s.t.} \quad \left\{ \begin{array}{ll} \boldsymbol{\alpha}^* \mathbf{K} \boldsymbol{\beta} = \mathbf{I} & \mathbf{K} = \{\kappa(x_i, x_j)\}_{(i,j) \in [n]^2} \\ \boldsymbol{\beta}_j^* \mathbf{K} \boldsymbol{\beta}_j = 1, \ \forall j \in [r] & \mathbf{P}_i \in \Pi(\mu_{T_{\boldsymbol{\theta}}}, \mu_{T_i}), \ \forall i \in [N] \end{array} \right. \tag{29}$$

where $\mathbf{P} = \{\mathbf{P}_i\}_{i \in [N]}$, $\widehat{\boldsymbol{T}} = \{\widehat{T}_i\}_{i \in [N]}$, such that $(\mathbf{C}_i(\boldsymbol{\theta}), \mathbf{P}_i)$ are the cost and transport matrices associated to the Wasserstein metric, $d_{\mathcal{S}}$ defined in proposition 3, between the parametric operator $T_{\boldsymbol{\theta}}$ and $\widehat{T}_i$.

## D.2 BARYCENTER ESTIMATION METHOD

**An inexact coordinate descent scheme.** In what follows, let $\mathcal{X}$ be a bounded open set of $\mathbb{R}^d$ with $d \in \mathbb{N}^*$ and $k : \mathcal{X} \times \mathcal{X} \to \mathbb{R}$ be a differentiable kernel. Following Cuturi & Doucet (2014), we propose an inexact coordinate descent scheme with a cyclic update rule designed to converge to a stationary point of problem 29. Each cycle begins with the computation of the exact optimal transport plans $\mathbf{P}$ to enforce sparsity. This step is carried out with the algorithm of Bonneel et al. (2011), whose complexity depends on the number of eigenvalues, typically small in practice (Brunton et al., 2022). The subsequent coordinate updates are performed using a few gradient descent steps with a first-order optimizer such as ADAM (Kingma & Ba, 2014). It starts with the eigenvalues $\boldsymbol{\lambda}$, optionally followed by the state spaces control points $\mathbf{x}$, for which no optimization constraints exist. Next, the right eigenfunctions, $\boldsymbol{\beta}$, are updated only considering the normalization constraints: $\boldsymbol{\beta}_j^* \mathbf{K} \boldsymbol{\beta}_j = 1$, $j \in [r]$. Finally, the left eigenfunctions, $\boldsymbol{\alpha}$, are updated considering the affine constraints: $\boldsymbol{\alpha}^* \mathbf{K} \boldsymbol{\beta} = \mathbf{I}$, leading to an iterated closed-form projection scheme detailed in Equation (33). Algorithm 1 summarizes the full procedure, and further implementation details are provided in the next paragraphs. We usually repeat 10 gradient descent steps in experiments when updating $\boldsymbol{\lambda}, \boldsymbol{\alpha}, \boldsymbol{\beta}$ and $\mathbf{x}$ at each cycle.

---

**Algorithm 1** Spectral Barycenter

---

**Require:** $\widehat{\boldsymbol{T}} \triangleq \{\widehat{T}_i\}_{i \in [N]} \in \mathcal{S}_r(\mathcal{H})^N$,
1: $\boldsymbol{\theta} \triangleq (\boldsymbol{\lambda}, \boldsymbol{\alpha}, \boldsymbol{\beta}, \mathbf{x}) \leftarrow Initialization(\widehat{\boldsymbol{T}})$         $\triangleright$ *Operator parameters, see eq.* (8)
2: **while** not converged **do**
3:   $\mathbf{P} \leftarrow ComputeTransportPlans(T_{\boldsymbol{\theta}}, \widehat{\boldsymbol{T}})$      $\triangleright$ *See Theorem 1, and Section 2*
4:   $\boldsymbol{\lambda} \leftarrow UpdateEigenValues(\boldsymbol{\theta}, \mathbf{P}, \widehat{\boldsymbol{T}})$
5:   **if** optimize control points **then**
6:    $\mathbf{x} \leftarrow UpdateControlPoints(\boldsymbol{\theta}, \mathbf{P}, \widehat{\boldsymbol{T}})$
7:   $\boldsymbol{\beta} \leftarrow UpdateRightEigenFunctions(\boldsymbol{\theta}, \mathbf{P}, \widehat{\boldsymbol{T}})$     $\triangleright$ *Detailed in Algorithm 2*
8:   $\boldsymbol{\alpha} \leftarrow UpdateLeftEigenFunctions(\boldsymbol{\theta}, \mathbf{P}, \widehat{\boldsymbol{T}})$     $\triangleright$ *Detailed in Algorithm 3*
9: $\mathbf{P} \leftarrow ComputeTransportPlans(T_{\boldsymbol{\theta}}, \widehat{\boldsymbol{T}})$
  **return** $\boldsymbol{\theta}, \mathbf{P}$

---

**Update right eigenfunctions.** We detail the *UpdateRightEigenFunctions* step of Algorithm 1. Let $\boldsymbol{\lambda}, \mathbf{x}, \boldsymbol{\alpha}$ and $\mathbf{P}$ be fixed; we aim to perform minimization steps of problem 29 with regard to $\boldsymbol{\beta}$, the parameters controlling the right eigenfunctions. Each optimization step consists of a first-order gradient descent step followed by a projection of each eigenfunction on the RKHS unit sphere as described in Algorithm 2.

---

**Algorithm 2** UpdateRightEigenFunctions

---

**Require:** $\boldsymbol{\theta} = (\boldsymbol{\lambda}, \mathbf{x}, \boldsymbol{\alpha}, \boldsymbol{\beta}), \mathbf{P}, \widehat{\boldsymbol{T}}$
1: **while** stopping criteria not met **do**
2:   $\widehat{\boldsymbol{\beta}} \leftarrow$ Gradient descent step of $J(\boldsymbol{\theta}, \mathbf{P}; \widehat{\boldsymbol{T}})$ w.r.t $\boldsymbol{\beta}$.
3:   **for** $i \in [r]$ **do**          $\triangleright$ *r being the number of eigenfunctions*
4:    $\boldsymbol{\beta}_i \leftarrow \widehat{\boldsymbol{\beta}}_i / \sqrt{\widehat{\boldsymbol{\beta}}_i \mathbf{K} \widehat{\boldsymbol{\beta}}_i}$      $\triangleright$ *Projection on the RKHS unit sphere*
  **return** $\boldsymbol{\beta}$

---

**Update left eigenfunctions.** We detail the *UpdateLeftEigenFunctions* step of Algorithm 1. Let $\boldsymbol{\lambda}, \mathbf{x}, \boldsymbol{\beta}$ and $\mathbf{P}$ be fixed; we aim to perform minimization steps of problem 29 with regard to $\boldsymbol{\alpha}$, the parameters controlling the left eigenfunctions. Each optimization step consists of a first-order gradient descent step followed by a projection onto the manifold induced by the spectral decomposition

constraint: $\boldsymbol{\alpha}^* \mathbf{K} \boldsymbol{\beta} = \mathbf{I}$. Algorithm 3 describes the optimization procedure, and the next paragraph discusses the projection.

---

**Algorithm 3** UpdateLeftEigenFunctions

---

**Require:** $\boldsymbol{\theta} = (\boldsymbol{\lambda}, \mathbf{x}, \boldsymbol{\alpha}, \boldsymbol{\beta}), \mathbf{P}, \widehat{\boldsymbol{T}}$
1: **while** stopping criteria not met **do**
2:     $\widehat{\boldsymbol{\alpha}} \leftarrow$ Gradient descent step of $J(\boldsymbol{\theta}, \mathbf{P}; \widehat{\boldsymbol{T}})$ w.r.t $\boldsymbol{\alpha}$.
3:     $\boldsymbol{\alpha} \leftarrow \widehat{\boldsymbol{\alpha}} - \boldsymbol{\beta} \left( (\widehat{\boldsymbol{\alpha}}^* \mathbf{K} \boldsymbol{\beta} - \mathbf{I}) (\boldsymbol{\beta}^* \mathbf{K} \boldsymbol{\beta})^{-1} \right)^*$           ▷ *Manifold Projection, see below.*
    **return** $\boldsymbol{\alpha}$

---

**Projection step.** Let $\widehat{\boldsymbol{\alpha}} \in \mathbb{C}^{n \times r}$ be the estimated parameters controlling the left eigenfunctions after a gradient descent step without constraints. Hence, these parameters might not verify the spectral decomposition constraint. We aim to identify the closest parameters, $proj(\widehat{\boldsymbol{\alpha}}) \in \mathbb{C}^{n \times r}$, for the RKHS metric and lying on the manifold induced by the spectral decomposition constraint. It leads to a constrained optimization problem:

$$Proj(\widehat{\boldsymbol{\alpha}}) \triangleq \underset{\boldsymbol{\alpha}}{\arg\min} \quad \text{Tr}\left((\boldsymbol{\alpha} - \widehat{\boldsymbol{\alpha}})^* \mathbf{K} (\boldsymbol{\alpha} - \widehat{\boldsymbol{\alpha}})\right) \\ \text{s.t.} \quad \boldsymbol{\alpha}^* \mathbf{K} \boldsymbol{\beta} = \mathbf{I} \tag{30}$$

Considering the real representation of the problem, it becomes a convex problem, and since $r \ll n$, it is strictly feasible. Strong duality holds by Slater's constraint qualification. As the optimization function $J$ and constraints are differentiable with respect to $\alpha$, the KKT conditions are necessary and sufficient conditions to characterize the optimum. The Lagrangian can be expressed as:

$$L(\boldsymbol{\alpha}, \boldsymbol{\mu}, \boldsymbol{\nu}) \triangleq \text{Tr}\left((\boldsymbol{\alpha} - \widehat{\boldsymbol{\alpha}})^* \mathbf{K} (\boldsymbol{\alpha} - \widehat{\boldsymbol{\alpha}})\right) + \boldsymbol{\mu}^\mathsf{T} (\text{Re}(\boldsymbol{\alpha}^* \mathbf{K} \boldsymbol{\beta}) - \mathbf{I}) + \boldsymbol{\nu}^\mathsf{T} (\text{Im}(\boldsymbol{\alpha}^* \mathbf{K} \boldsymbol{\beta})) . \tag{31}$$

Taking Wirtinger derivative notation, the optimal primal-dual variables verify:

$$\begin{cases} \nabla_{\overline{\boldsymbol{\alpha}}} L(\boldsymbol{\alpha}, \boldsymbol{\mu}, \boldsymbol{\nu}) = \mathbf{K}(\boldsymbol{\alpha} - \widehat{\boldsymbol{\alpha}} + \boldsymbol{\beta}(\boldsymbol{\mu} - i\boldsymbol{\nu})^\mathsf{T}) = \mathbf{0} \\ \nabla_{\boldsymbol{\mu}} L(\boldsymbol{\alpha}, \boldsymbol{\mu}, \boldsymbol{\nu}) = \text{Re}(\boldsymbol{\alpha}^* \mathbf{K} \boldsymbol{\beta}) - \mathbf{I} = \mathbf{0} \\ \nabla_{\boldsymbol{\nu}} L(\boldsymbol{\alpha}, \boldsymbol{\mu}, \boldsymbol{\nu}) = \text{Im}(\boldsymbol{\alpha}^* \mathbf{K} \boldsymbol{\beta}) = \mathbf{0} \end{cases} \tag{32}$$

Regarless of the rank of $\mathbf{K}$, $\boldsymbol{\alpha} - \widehat{\boldsymbol{\alpha}} + \boldsymbol{\beta}(\boldsymbol{\mu} - i\boldsymbol{\nu})^\mathsf{T} = \mathbf{0}$ always verifies the first optimality equation. It leads to the projector:

$$Proj(\widehat{\boldsymbol{\alpha}}) = \widehat{\boldsymbol{\alpha}} - \boldsymbol{\beta} \left( (\widehat{\boldsymbol{\alpha}}^* \mathbf{K} \boldsymbol{\beta} - \mathbf{I}) (\boldsymbol{\beta}^* \mathbf{K} \boldsymbol{\beta})^{-1} \right)^* . \tag{33}$$

### D.3 CASE OF FINITE-DIMENSIONAL RKHS

Consider $\mathcal{H}$ be finite $d$-dimensional RKHS with the orthonormal basis $\mathbf{f} = \{f_i\}_{i \in [d]}$. For instance, $\mathcal{H}$ is based on a functional dictionary as used in extended DMD (Kutz et al., 2016). Let $\mathcal{S}_r(\mathcal{H})$ be the set of non-defective compact operators acting on $\mathcal{H}$ with rank at most $r \le d$. We aim to solve:

$$\underset{T \in \mathcal{Z}_r(\mathcal{H})}{\arg\min} \sum_{i \in [N]} \gamma_i d_{\mathcal{S}}(T, \widehat{T}_i)^2 , \tag{34}$$

with $\boldsymbol{\gamma} \in \Delta^N$ the importance weights, $\{\widehat{T}_i \in \mathcal{Z}_r(\mathcal{H}) \mid i \in [N]\}$ estimated operators, and $d_{\mathcal{S}}$ defined in Theorem 1 given $\eta \in (0, 1)$ and $p = 2$. For any compact operator acting on $\mathcal{H}$ with rank at most $r$, there exists coefficients $\boldsymbol{\lambda} \in \mathbb{C}^r$, and control parameters of the functional basis $\boldsymbol{\alpha}, \boldsymbol{\beta} \in \mathbb{C}^{d \times r}$ such that:

$$T_{\theta} : h \in \mathcal{H} \mapsto \sum_{i \in [r]} \lambda_i \langle f_{\boldsymbol{\alpha}_i}, h \rangle_{\mathcal{H}} f_{\boldsymbol{\beta}_i} \in \mathcal{H} , \tag{35}$$

where $f_{\boldsymbol{\alpha}_i} \triangleq \sum_{j \in d} \alpha_{ji} f_j$, $f_{\boldsymbol{\beta}_i} \triangleq \sum_{j \in d} \beta_{ji} f_j$, and $\theta \triangleq (\boldsymbol{\lambda}, \boldsymbol{\alpha}, \boldsymbol{\beta})$. To ensure non-defectiveness of $T_{\theta}$ further constraints are imposed on the control parameters, which leads to the constrained optimization problem:

$$\underset{\boldsymbol{\lambda}, \boldsymbol{\alpha}, \boldsymbol{\beta}}{\arg\min} \quad \sum_{i \in [N]} \gamma_i d_{\mathcal{S}}(T_{\theta}, \widehat{T}_i)^2 \\ \text{s.t.} \quad \boldsymbol{\alpha}^* \boldsymbol{\beta} = \mathbf{I} \\ \boldsymbol{\beta}_i^* \boldsymbol{\beta}_i = 1, \quad \forall i \in [r] \tag{36}$$

Note that this optimization problem is related to the infinite-dimensional problem defined eq. (9) by assuming the control points to be fixed such that the kernel matrix is the identity matrix, i.e. $\mathbf{K} = \mathbf{I}$. It follows that the optimization procedure described in the case of infinite-dimensional RKHS in Appendix D.2 also handles the finite-dimensional case.

## E    PROOFS OF STATISTIC RESULTS

We now prove the main statistical results in this section.

**Theorem 2.** *Let (A1)-(A3) hold with $k \in [2]$, $\mathcal{F}_k = \mathcal{L}^2_{\pi_k}(\mathcal{X})$ and $\kappa(x,x) < \infty$ a.s. for $x \sim \pi_k$. Let $\mathbb{E}[\widehat{C}^k_x] = C^k_x$ and assume that for some $\alpha \in [1,2]$ and $\beta \in [0,1]$ it holds that $\|\|[(C^k_x)^\dagger]^{\frac{\alpha-1}{2}} T_k\|_{\mathcal{H} \to \mathcal{H}} < \infty$ and $\lambda_i(C^k_x) \leq \lesssim i^{-1/\beta}$ for $i \in \mathbb{N}$. Given $\delta \in (0,1)$, if $n$ is large enough and $\lambda_{r_k} \lesssim -\frac{\alpha \log n}{2(\alpha+\beta)}$, then w.p.a.l. $1-\delta$ in the i.i.d. draw of samples $\mathcal{D}_1$ and $\mathcal{D}_2$ it holds $|d_{\mathcal{S}}(\widehat{T}_1, \widehat{T}_2) - d_{\mathcal{S}}(T_1, T_2)| \lesssim n^{-\frac{\alpha-1}{2(\alpha+\beta)}} \ln(2\delta^{-1})$.*

*Proof of Theorem 2.* Without loss of generality, we can assume that the operators eigenvalues are of multiplicity 1. Then the discrete distribution representation of the operator $T_k$ provided in equation 4 becomes

$$\mu(T_k) \triangleq \frac{1}{r_k} \sum_{j \in [r_k]} \delta_{(\lambda_j(k), \mathcal{V}_j(k))}. \tag{37}$$

Similarly

$$\mu(\widehat{T}_k) \triangleq \frac{1}{r_k} \sum_{j \in [r_k]} \delta_{(\widehat{\lambda}_j(k), \widehat{\mathcal{V}}_j(k))}. \tag{38}$$

**Stability of the $d_{\mathcal{S}}$ metric.**    Next by definition of $d_{\mathcal{S}}$ in equation 5 and the triangular inequality applied to the Wasserstein metric:

$$\left| d_{\mathcal{S}}(\widehat{T}_1, \widehat{T}_2) - d_{\mathcal{S}}(T_1, T_2) \right| \leq W_p\big(\mu(T_1), \mu(\widehat{T}_1)\big) + W_p\big(\mu(T_2), \mu(\widehat{T}_2)\big). \tag{39}$$

We recall that $W_p\big(\mu(T_k), \mu(\widehat{T}_k)\big)$ is defined as:

$$W_p\big(\mu(T_k), \mu(\widehat{T}_k)\big) := \left( \min_{\mathbf{P} \in \Pi_{\text{uniform}}(r_k)} \sum_{i=1}^{r_k} \sum_{j=1}^{r_k} c^p_{i,j} \, P_{i,j} \right)^{1/p},$$

where the cost matrix $C_k = (c_{i,j})_{i,j \in [r_k]}$ is defined as

$$c_{i,j} := d_\eta((\lambda_i(k), \mathcal{V}_i(k)), (\widehat{\lambda}_j(k), \widehat{\mathcal{V}}_j(k))) \geq 0, \quad \forall i,j \in [r_k],$$

and the set of uniform transport plans is:

$$\Pi_{\text{uniform}}(r_k) := \left\{ \mathbf{P} \in \mathbb{R}^{r_k \times r_k}_+ \; \Big| \; \mathbf{P1} = \frac{1}{r_k}\mathbf{1}, \; \mathbf{P}^\intercal \mathbf{1} = \frac{1}{r_k}\mathbf{1} \right\}.$$

Then we note that the transport plan $\pi_{i,i} = 1/r_k$ for any $i \in [r_k]$ and $\pi_{i,j} = 0$ for any $i \neq j$ belongs to the set $\Pi_{\text{uniform}}(r_k)$. Consequently

$$W^p_p\big(\mu(T_k), \mu(\widehat{T}_k)\big) \leq \sum_{i=1}^{r_k} \sum_{j=1}^{r_k} c^p_{i,j} \, \pi_{i,j} = \frac{1}{r_k} \sum_{i=1}^{r_k} c^p_{i,i}, \tag{40}$$

In view of equation 49 below, we prove w.p.a.l. $1 - \delta$ that

$$\max_{i \in [r_k]} \{c_{i,i}\} \lesssim \varepsilon_n(\delta) := n^{-\frac{\alpha-1}{2(\alpha+\beta)}} \ln(2\delta^{-1}), \quad \forall k \in [2].$$

Then we get with the same probability

$$W_p\big(\mu(T_k), \mu(\widehat{T}_k)\big) \lesssim n^{-\frac{\alpha-1}{2(\alpha+\beta)}} \ln(2\delta^{-1}), \quad \forall k \in [2],$$

and consequently we obtain the final bound on $\left| d_{\mathcal{S}}(\widehat{T}_1, \widehat{T}_2) - d_{\mathcal{S}}(T_1, T_2) \right|$ in view of equation 39.

**Bounding the learning error $\|T - \widehat{T}\|$.** For brevity we set $\| \cdot \|$ for the operator norm $\| \cdot \|_{\mathcal{H} \to \mathcal{H}}$ and $\| \cdot \|_{\mathcal{H}}$ for the Hilbert-Schmidt norm $\| \cdot \|_{\mathrm{HS}(\mathcal{H}, \mathcal{H})}$. For any $k$, we introduce the population RRR and Ridge operators as

$$T_{k,\gamma} := (C_x^k + \gamma I)^{-\frac{1}{2}} [\![ (C_x^k + \gamma I)^{-\frac{1}{2}} C_{xy}^k ]\!]_{r_k}, \quad T_{k,\gamma}^R = (C_x^k + \gamma)^{-1} C_{xy}^k.$$

Then we have the following Bias-Variance decomposition in operator norm:

$$\|T_k - \widehat{T}_k\| \leq \underbrace{\|T_k - T_{k,\gamma}\|}_{=: \, a_1 \text{ "Bias"}} + \underbrace{\|T_{k,\gamma} - \widehat{T}_k\|}_{=: \, a_2 \text{ "Variance"}}. \tag{41}$$

**Bias term $a_1$.** We have

$$a_1 \leq \|T_k - T_{k,\gamma}^R\| + \|T_{k,\gamma}^R - T_{k,\gamma}\|. \tag{42}$$

Next since $P_{\leq r_k} A_k S_{\pi_k} = S_{\pi_k} T_k$, we get

$$S_{\pi_k}^* P_{\leq r_k} A_k S_{\pi_k} = C_x^k T_k,$$

and

$$\begin{aligned} T_{k,\gamma}^R &= (C_x^k + \gamma I)^{-1} C_x^k T_k + (C_x^k + \gamma I)^{-1} S_{\pi_k}^* P_{\leq r_k}^\perp A_{\pi_k} S_{\pi_k} \\ &= T_k - \gamma (C_x^k + \gamma I)^{-1} (C_x^k)^{(\alpha-1)/2} (C_x^k)^{\dagger(\alpha-1)/2} T_k + (C_x^k + \gamma I)^{-1} S_{\pi_k}^* P_{\leq r_k}^\perp A_{\pi_k} S_{\pi_k}. \end{aligned}$$

Therefore,

$$\begin{aligned} \|T_k - T_{k,\gamma}^R\| &\leq \gamma \|(C_x^k + \gamma I)^{-1} (C_x^k)^{(\alpha-1)/2}\| \, \|[(C_x^k)^\dagger]^{1-\alpha} T_k\| + \frac{1}{\sqrt{\gamma}} \|P_{\leq r_k}^\perp A_{\pi_k}\| \sqrt{c_\mathcal{H}} \\ &\leq \gamma^{(\alpha-1)/2} \|[(C_x^k)^\dagger]^{1-\alpha} T_k\| + \sqrt{\frac{c_\mathcal{H}}{\gamma}} \|P_{\leq r_k}^\perp A_{\pi_k}\| \end{aligned} \tag{43}$$

We tackle now the second term in the right-hand side of equation 42. We note first that

$$T_{k,\gamma}^R - T_{k,\gamma} = (C_x^k + \gamma I)^{-1} C_{xy}^k (I - \Pi_{r_k}).$$

Hence we get

$$\|T_{k,\gamma}^R - T_{k,\gamma}\| \leq \frac{1}{\sqrt{\gamma}} \sigma_{r_k+1}((C_x^k + \gamma I)^{-1/2} C_{xy}^k) \leq \frac{\sqrt{c_\mathcal{H}}}{\sqrt{\gamma}} \|P_{\leq r_k}^\perp A_{\pi_k}\|. \tag{44}$$

Combining equation 42, equation 43 and equation 44 with the assumption $\||[(C_x^k)^\dagger]^{\frac{\alpha-1}{2}} T_k\|_{\mathcal{H} \to \mathcal{H}} < \infty$, we obtain the following control on the bias:

$$\begin{aligned} a_1 = \|T_k - T_{k,\gamma}\| &\leq \gamma^{(\alpha-1)/2} \|[(C_x^k)^\dagger]^{1-\alpha} T_k\| + \frac{2\sqrt{c_\mathcal{H}}}{\sqrt{\gamma}} \|P_{\leq r_k}^\perp A_{\pi_k}\| \\ &\lesssim \gamma^{(\alpha-1)/2} + \frac{2\sqrt{c_\mathcal{H}}}{\sqrt{\gamma}} \|P_{\leq r_k}^\perp A_{\pi_k}\| \end{aligned} \tag{45}$$

**Variance term $a_2$.** We note first

$$T_{k,\gamma} - \widehat{T}_k = (C_x^k + \gamma I)^{-1/2} (C_x^k + \gamma I)^{1/2} (T_{k,\gamma} - \widehat{T}_k)$$

Taking the operator norm, we get

$$a_2 = \|T_{k,\gamma} - \widehat{T}_k\| \leq \|C_x^k + \gamma I\|^{-1/2} \|(C_x^k + \gamma I)^{1/2} (T_{k,\gamma} - \widehat{T}_k)\| \leq \frac{1}{\sqrt{\gamma}} \|(C_x^k + \gamma I)^{1/2} (T_{k,\gamma} - \widehat{T}_k)\|.$$

Define $B_k := (C_x^k + \gamma I)^{-1/2} C_{xy}^k$. An analysis of the variance of the RRR estimation (see Sections D.3.4. and D.4 and more specifically Lemma 1 and the proof of Proposition 18 in Kostic et al. (2023)) gives for any $\delta \in (0, 1)$ w.p.a.l. $1 - \delta$

$$\begin{aligned} a_2 &\leq \frac{1}{\sqrt{\gamma}} \|(C_x^k + \gamma I)^{1/2} (T_{k,\gamma} - \widehat{T}_k)\| \\ &\lesssim \frac{1}{n^{1/2} \gamma^{(\beta+1)/2}} \ln \delta^{-1} + \frac{1}{\sqrt{\gamma n}} \left( 1 + \frac{\sigma_1(B_k)}{\sigma_{r_k}^2(B_k) - \sigma_{r+1}^2(B_k)} \right) \ln \delta^{-1}. \end{aligned} \tag{46}$$

Combining the previous display with equation 45, we get w.p.a.l. $1 - \delta$

$$\|T_k - \widehat{T}_k\| \lesssim \gamma^{(\alpha-1)/2} + \frac{1}{n^{1/2}\gamma^{(\beta+1)/2}} \ln \delta^{-1}$$
$$+ \frac{2\sqrt{c_{\mathcal{H}}}}{\sqrt{\gamma}} \|P^{\perp}_{\leq r_k} A_k\| + \frac{1}{\sqrt{\gamma n}} \left(1 + \frac{\sigma_1(B_k)}{\sigma^2_{r_k}(B_k) - \sigma^2_{r+1}(B_k)}\right) \ln \delta^{-1}. \quad (47)$$

Since we assumed that the spectrum of $A_k$ decreases exponentially fast to 0, that is $\lambda_{r_k} \lesssim -\frac{\alpha \log n}{2(\alpha+\beta)}$, and assuming in addition that the gap $gap_{r_k}$ is bounded away from 0. Then, for $\gamma \in (0,1)$ small, the dominating terms in the above display are the first two terms and we propose to balance $\gamma$ using only those two. Hence we get for $\gamma \asymp n^{-\frac{1}{\alpha+\beta}}$ w.p.a.l. $1 - \delta$

$$\|T_k - \widehat{T}_k\| \lesssim n^{-\frac{\alpha-1}{2(\alpha+\beta)}} \ln \delta^{-1}. \quad (48)$$

**Perturbation bounds.** For simplicity we assume here that the all the eigenvalues admit multiplicity 1. By a standard Davis-Kahan perturbation argument, we get

$$|\nu_i - \widehat{\nu}_i| \leq \|\xi_i\| \|\psi_i\| \|T - \widehat{T}\|$$

$$\|\xi_i - \widehat{\xi}_i\| \leq \|\xi_i\| \|\psi_i\| \frac{\|T - \widehat{T}\|}{gap_i}$$

$$\|\psi_i - \widehat{\psi}_i\| \leq \|\xi_i\| \|\psi_i\| \frac{\|T - \widehat{T}\|}{gap_i}$$

**Final Bound on the metric**. An union combining equation 48 for any $k \in [N]$, we get w.p.a.l. $1 - \delta$ that the condition in equation 50 in Lemma 5 is satisfied with

$$\varepsilon_0 = \varepsilon_1 = n^{-\frac{\alpha-1}{2(\alpha+\beta)}} \ln(N\delta^{-1}) =: \varepsilon_n(\delta).$$

Proposition 5 guarantees w.p.a.l. $1 - \delta$ that for any $k \in [N]$, the operators $T_k, \widehat{T}_k$ with corresponding spectral decomposition $(\nu_i^{(k)}, P_i^{(k)})$ and $(\widehat{\nu}_i^{(k)}, \widehat{P}_i^{(k)})$: $\forall i \in [r_k]$,

$$|d_\eta\big((\nu_i^{(k)}, P_i^{(k)}), (\widehat{\nu}_i^{(k)}, \widehat{P}_i^{(k)})\big)| \leq 2\sqrt{2} \frac{\|\xi_i^{(k)}\| \|\psi_i^{(k)}\|}{gap_i^{(k)} \wedge |\lambda_i^{(k)}|} \varepsilon_n(\delta), \quad \forall i \in [r_k], \ \forall k \in [N]. \quad (49)$$

$\square$

### E.1 AUXILIARY RESULTS

We propose a control on the metric $d_\eta(\cdot, \cdot)$. For simplicity, we assume that all the eigenvalues of the Koopman transfer operators are of multiplicity 1.

**Proposition 5.** *Let $\varepsilon_0, \varepsilon_1 \in (0, 1/2)$ be an absolute constant such that, for any $i \in [r]$,*

$$\frac{|\nu_i - \hat{\nu}_i|}{|\nu_i|} \leq \frac{\|\xi_i\| \|\psi_i\|}{|\nu_i|} \varepsilon_0 \quad and \quad \|P_i - \widehat{P}_i\| \leq \frac{\|\xi_i\| \|\psi_i\|}{gap_i} \varepsilon_1. \quad (50)$$

*Then we have for any $i \neq j \in [r]$*

$$|d_\eta(\nu_i, P_i), (\nu_j, P_j)) - d_\eta(\widehat{\nu}_i, \widehat{P}_i), (\widehat{\nu}_j, \widehat{P}_j))|$$
$$\leq 2\sqrt{2} \left(\left(\frac{\|\xi_i\| \|\psi_i\|}{|\nu_i|} \vee \frac{\|\xi_j\| \|\psi_j\|}{|\nu_j|}\right) \varepsilon_0 + \left(\frac{\|\xi_i\| \|\psi_i\|}{gap_i} \vee \frac{\|\xi_j\| \|\psi_j\|}{gap_j}\right) \varepsilon_1\right).$$

*Similarly for any $i \in [r]$*

$$d_\eta(\nu_i, P_i), (\widehat{\nu}_i, \widehat{P}_i)) \leq 2\sqrt{2} \left(\frac{\|\xi_i\| \|\psi_i\|}{|\nu_i|} \varepsilon_0 + \frac{\|\xi_i\| \|\psi_i\|}{gap_i} \varepsilon_1\right).$$

*Proof of Proposition 5.* The metric $d_\eta$ is a convex combination of two parts.

We focus on the distance between generator eigenvalues, which is the same as the polar distance $d_{val}$ between transfer operator eigenvalues. Similarly as above, we have by the triangular inequality

$$\left| d_{val}(\nu, \nu') - d_{val}(\widehat{\nu}, \widehat{\nu}') \right| \leq \|(\tau, \omega) - (\hat{\tau}, \hat{\omega})\|_2 + \|(\tau', \omega') - (\hat{\tau}', \hat{\omega}')\|_2$$

Using Lemma 1, we get that

$$\|(\tau, \omega) - (\hat{\tau}, \hat{\omega})\|_2^2 \leq |\nu - \hat{\nu}|^2 + \arcsin\left(\frac{|\nu - \hat{\nu}|^2}{4|\nu||\hat{\nu}|}\right). \tag{51}$$

Assume the relative eigenvalue error is small:

$$\frac{|\nu - \hat{\nu}|}{|\nu|} \vee \frac{|\nu' - \hat{\nu}'|}{|\nu'|} \leq \varepsilon < \frac{1}{2}. \tag{52}$$

Under equation 52 we have $|\hat{\nu}| \geq (1 - \varepsilon)|\nu|$ and therefore

$$u := \frac{|\nu - \hat{\nu}|^2}{4|\nu||\hat{\nu}|} \leq \frac{\varepsilon^2}{4(1 - \varepsilon)} < 1,$$

so the argument of $\arcsin(\cdot)$ lies in $(0, 1)$ as required. Moreover, since $\arcsin(x)$ is Lipschitz near 0 and $\arcsin(x) \leq (\pi/2)x$ for all $x \in [0, u]$, we get

$$\arcsin\left(\frac{|\nu - \hat{\nu}|^2}{4|\nu||\hat{\nu}|}\right) \leq \frac{\pi}{2}\frac{|\nu - \hat{\nu}|^2}{4|\nu||\hat{\nu}|}.$$

Hence

$$\|(\tau, \omega) - (\hat{\tau}, \hat{\omega})\|_2^2 \leq |\nu - \hat{\nu}|^2\left(1 + \frac{\pi}{8|\nu||\hat{\nu}|}\right) \leq |\nu - \hat{\nu}|^2\left(1 + \frac{\pi}{8(1 - \varepsilon)|\nu|^2}\right),$$

and therefore

$$\|(\tau, \omega) - (\hat{\tau}, \hat{\omega})\|_2 \leq |\nu - \hat{\nu}|\sqrt{1 + \frac{\pi}{8(1 - \varepsilon)|\nu|^2}} \leq \sqrt{2}\frac{|\nu - \hat{\nu}|}{|\nu|}, \tag{53}$$

since $|\nu| \leq 1$ for all transfer operator eigenvalues.

Apply the same bound to $(\nu', \hat{\nu}')$ and combine with the first display to obtain, for each matched pair of eigenvalues,

$$\left| d_{val}(\nu, \nu') - d_{val}(\widehat{\nu}, \widehat{\nu}') \right| \lesssim \left(\frac{|\nu - \hat{\nu}|}{|\nu|} + \frac{|\nu' - \hat{\nu}'|}{|\nu'|}\right).$$

Now apply this inequality to every eigenvalue pairs $i \neq j \in [r]$. In view of equation 50, we get

$$\left| d_{val}(\nu_i, \nu_j) - d_{val}(\widehat{\nu}_i, \widehat{\nu}_j) \right| \lesssim \left(\frac{\|\xi_i\|\|\psi_i\|}{|\nu_i|} + \frac{\|\xi_j\|\|\psi_j\|}{|\nu_j|}\right)\varepsilon_0, \quad \forall i \neq j \in [r], \tag{54}$$

For the Grassmanian part, we have for any $i \neq j \in [r]$

$$\left| \|P_i - P_j\| - \|\widehat{P}_i - \widehat{P}_j\| \right| \leq \|P_i - \widehat{P}_i - (P_j - \widehat{P}_j)\| \leq 2\sqrt{2}\left(\frac{\|\xi_i\|\|\psi_i\|}{\text{gap}_i} \vee \frac{\|\xi_j\|\|\psi_j\|}{\text{gap}_j}\right)\varepsilon_1.$$

Combining the last two displays gives the first result. The second result follows from a similar and actually simpler argument. □

**Lemma 1.** *Let $z_1 = r_1 e^{i\theta_1}$ and $z_2 = r_2 e^{i\theta_2}$ be complex numbers in polar form with $r_1, r_2 \geq 0$ and $\theta_1, \theta_2 \in [0, 2\pi)$. Then*

$$|z_1 - z_2|^2 = (r_1 - r_2)^2 + 2r_1 r_2\left(1 - \cos(\theta_1 - \theta_2)\right) = (r_1 - r_2)^2 + 4r_1 r_2 \sin^2\left(\frac{\theta_1 - \theta_2}{2}\right).$$

*Proof.* Write the difference and compute its squared modulus:

$$|z_1 - z_2|^2 = |r_1 e^{i\theta_1} - r_2 e^{i\theta_2}|^2 = (r_1 e^{i\theta_1} - r_2 e^{i\theta_2})(r_1 e^{-i\theta_1} - r_2 e^{-i\theta_2}).$$

Expanding yields

$$|z_1 - z_2|^2 = r_1^2 + r_2^2 - r_1 r_2 \big( e^{i(\theta_1 - \theta_2)} + e^{-i(\theta_1 - \theta_2)} \big).$$

Using $e^{i\phi} + e^{-i\phi} = 2\cos\phi$ with $\phi = \theta_1 - \theta_2$ gives

$$|z_1 - z_2|^2 = r_1^2 + r_2^2 - 2 r_1 r_2 \cos(\theta_1 - \theta_2).$$

Rearrange the first two terms as a perfect square plus a correction:

$$r_1^2 + r_2^2 - 2 r_1 r_2 \cos\phi = (r_1^2 + r_2^2 - 2 r_1 r_2) + 2 r_1 r_2 (1 - \cos\phi) = (r_1 - r_2)^2 + 2 r_1 r_2 (1 - \cos\phi).$$

Finally, apply the trigonometric identity $1 - \cos x = 2\sin^2(x/2)$ to obtain

$$2 r_1 r_2 (1 - \cos\phi) = 4 r_1 r_2 \sin^2\!\Big(\frac{\phi}{2}\Big),$$

which yields the claimed expression. $\qquad\square$

## F  COMPARISON WITH OTHER SIMILARITY MEASURES

### F.1  EXPERIMENT PROTOCOL

**Simulated system and shifts.**  We consider a referent linear oscillatory system that is the sum of two simple harmonic oscillators with frequencies 0.5Hz and 1.0Hz, respectively, with a noisy trajectory of length 4001 samples sampled at 200Hz, which is an additive Gaussian noise with standard deviation of $1e-2$. We compare the Koopman operator of the referent system with those of shifted systems according to four scenarios:

(a) **Frequency shift**, changes the 1Hz harmonic frequency from 0.6Hz to 2.5Hz in 39 evenly spaced frequencies.

(b) **Decay rate shift**, changes the 1Hz harmonic decay rate from -0.3 (diverging) to 3.0 (converging) in 67 evenly spaced rates.

(c) **Subspace shift (rank)** gradually transforms the 1Hz sine wave into a 1Hz square wave signal using a Fourier Decomposition of a square wave signal with increasing order up to 50. Series formulation of a square wave signal: $s(t) = \frac{4}{\pi} \sum_{n=0}^{\infty} \frac{1}{2n+1} \sin\big((2n+1)t\big)$.

(d) **Sampling frequency shift** where the system is sampled at different sampling frequencies ranging from 100Hz to 300Hz instead of the reference 200Hz. Performed in 19 evenly spaced sampling frequencies.

Koopman operators are estimated from sampled trajectories in each scenario with the RRR method (Kostic et al., 2022). We consider the linear kernel, the context (sliding window) is set to one second, the operators' rank is always fixed to twice the number of harmonic oscillators, and the Tikhonov regularization is set to $1e-8$.

**Compared similarity measures.**  We consider our proposed metric SGOT set with $\eta = 0.5$. SOT, an OT-based similarity comparing eigenvalues (Redman et al., 2024). GOT, an OT-based similarity comparing eigensubspaces with a Grassmannian metric and weighted by the normalized eigenvalues (Antonini & Cavalletti, 2021). Note that compared to its theoretical definition, we extend the similarity to non-normal operators by taking the absolute value of eigenvalues. We also included the metrics induced by the Hilbert-Schmidt and Operator norms, and the Martin similarity (Martin, 2002), which compares poles of LDS transfer functions.

### F.2  ABLATION STUDY FOR PARAMETER $\eta$ OF SGOT

Following the same protocol presented in the previous paragraph, we compare our proposed metric SGOT with the parameter controlling the balance between eigenvalues and eigensubspaces $\eta$ ranging in $[0.1, 0.2, \ldots, 0.9]$. Results are presented in Figure 6. In scenarios (a,b,c) for any $\eta$, SGOT behaves piecewise linearly, where the ascent gets steeper for scenario (a,b) as $\eta$ decreases (eigensubspaces have more weights in the cost function). For scenario (c), SGOT behaves similarly for all $\eta$. Finally, SGOT becomes slightly more sensitive to the sampling frequency as $\eta$ decreases. In (d), the metric scale is not normalized, and the metric values remain relatively small.

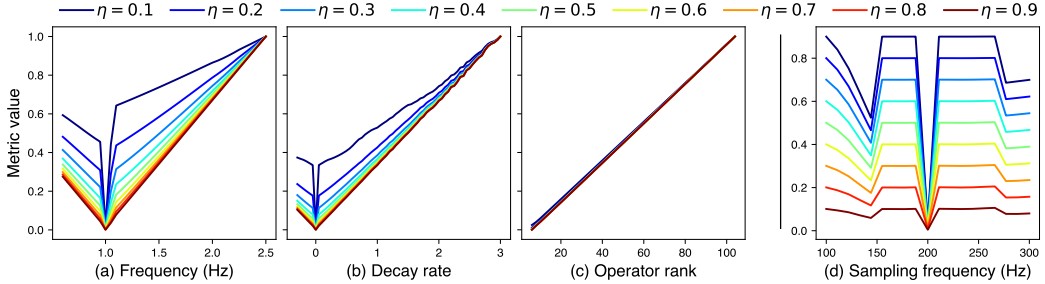

Figure 6: Influence of the $\eta$ parameter in SGOT under four scenarios of shifts of a linear oscillatory system: (a) frequency shift, (b) decay rate shift, (c) operator rank/subspace shift, (d) sampling frequency variation. In scenarios (a,b,c), metric values are normalized by their maximum.

## G    MACHINE LEARNING ON DYNAMICAL SYSTEMS

### G.1    EXPERIMENTAL PROTOCOL

We evaluate similarity performances on a time series classification task. We selected 14 multivariate datasets from the UEA database (Ruiz et al., 2021) whose main characteristics are described in Table 4. For each dataset, we estimate operators for individual time series of $n$ samples with the RRR method (Kostic et al., 2022), with the linear kernel, a Tikhonov regularization of $1e-2$, an arbitrary sampling frequency $f_{samp} \triangleq \min(100, (n/2) * 0.2)$ and a context window $w_{len} \triangleq \min(50, n/2)$. Once all operators are estimated, we perform a 10-iteration Monte-Carlo cross-validation with a 0.7/0.3 train/test split without any preprocessing step. To perform classification, we consider K-Nearest Neighbors (K-NN) estimators defined with similarities: Hilbert-Schmidt, Operator, Martin (Martin, 2002), SOT (Redman et al., 2024), GOT (Antonini & Cavalletti, 2021), and our metric SGOT. Note that the initialization-invariant Binet-Cauchy similarity has been excluded from this experiment as it relates to the Martin distance. At each cross-validation iteration, the number of neighbors (K) and the parameter $\eta$ for SGOT metric are set by grid search with a 5-fold cross-validation on the train set. K peaked between 1 and 10 and $\eta \in [0.001, 0.01, 0.1, 0.5, 0.9, 0.99]$. Scores are evaluated in terms of accuracy, and a training time limit has been set to 5 hours per dataset/metric pair. The experiment has been seeded for reproducibility.

Table 4: Datasets main characteristics: *Size*: number of time series, *Channels*: number of dimensions per time series, *Length*: time series length, *Classes*: number of classes.

|  | #Size | #Channels | Length | #Classes |
|---|---|---|---|---|
| AtrialFibrillation | 30 | 2 | 640 | 3 |
| BasicMotions | 80 | 6 | 100 | 4 |
| Cricket | 180 | 6 | 1197 | 12 |
| EigenWorms | 259 | 6 | 17984 | 5 |
| Epilepsy | 275 | 3 | 206 | 4 |
| ERing | 300 | 4 | 65 | 6 |
| FingerMovements | 416 | 28 | 50 | 2 |
| HandMovementDirection | 234 | 10 | 400 | 4 |
| Handwriting | 1000 | 3 | 152 | 26 |
| Heartbeat | 409 | 61 | 405 | 2 |
| NATOPS | 360 | 24 | 51 | 6 |
| SelfRegulationSCP1 | 561 | 6 | 896 | 2 |
| StandWalkJump | 27 | 4 | 2500 | 3 |
| UWaveGestureLibrary | 440 | 3 | 315 | 8 |

### G.2    LINEAR KERNEL: ADDITIONAL RESULTS

**Classification accuracy table.**    In addition to scores comparison plots between our metric SGOT and competitive similarities in the main body (see Figure 3), Table 5 provides mean and standard deviation of accuracy scores per dataset and metric computed over the 10 iterations. Our metric SGOT

is the best performer on all datasets, followed by GOT, another OT-based metric that only refers to eigensubspaces in its cost function. Also, SOT, a third OT-based similarity comparing operator, only from eigenvalues, performs poorly. Incorporating eigenvalues and eigensubspaces within the cost function improves performance on numerous datasets. By being more conservative (see Figure 1), Hilbert-Schmidt and Operator underperform compared to SGOT. Note that the Operator norm times out on Heartbeat. Lastly, Martin distance performs poorly and fails on some datasets due to its ill-definedness in some settings.

Table 5: Classification accuracy scores. Transfer operators are estimated with the finite dimensional linear kernel. Datasets on rows and similarities on columns. **Best** and second best performers are highlighted. Accuracy scores are denoted: $< mean > \pm < std >$.

|  | Hilbert-Schmidt | Operator | Martin | SOT | GOT | SGOT |
|---|---|---|---|---|---|---|
| AtrialFibrillation | $0.31 \pm 0.07$ | $0.32 \pm 0.13$ | $0.27 \pm 0.09$ | $0.24 \pm 0.14$ | $0.4 \pm 0.12$ | $\mathbf{0.44 \pm 0.13}$ |
| BasicMotions | $0.48 \pm 0.15$ | $0.51 \pm 0.13$ | $0.3 \pm 0.06$ | $0.35 \pm 0.1$ | $0.8 \pm 0.07$ | $\mathbf{0.93 \pm 0.05}$ |
| Cricket | $0.33 \pm 0.05$ | $0.28 \pm 0.05$ | $0.07 \pm 0.03$ | $0.11 \pm 0.04$ | $0.63 \pm 0.04$ | $\mathbf{0.85 \pm 0.05}$ |
| ERing | $0.79 \pm 0.04$ | $0.74 \pm 0.05$ | $0.15 \pm 0.04$ | $0.39 \pm 0.04$ | $0.85 \pm 0.01$ | $\mathbf{0.87 \pm 0.03}$ |
| EigenWorms | $0.6 \pm 0.04$ | $0.57 \pm 0.04$ | $\emptyset$ | $0.57 \pm 0.06$ | $0.71 \pm 0.04$ | $\mathbf{0.88 \pm 0.03}$ |
| Epilepsy | $0.46 \pm 0.05$ | $0.52 \pm 0.06$ | $\emptyset$ | $0.34 \pm 0.04$ | $0.78 \pm 0.04$ | $\mathbf{0.93 \pm 0.03}$ |
| FingerMovements | $0.51 \pm 0.05$ | $0.54 \pm 0.03$ | $\emptyset$ | $0.51 \pm 0.05$ | $0.51 \pm 0.05$ | $\mathbf{0.57 \pm 0.03}$ |
| HandMovementDirection | $0.23 \pm 0.04$ | $0.23 \pm 0.03$ | $0.27 \pm 0.04$ | $0.21 \pm 0.05$ | $0.24 \pm 0.05$ | $\mathbf{0.29 \pm 0.03}$ |
| Handwriting | $0.12 \pm 0.02$ | $0.12 \pm 0.02$ | $0.05 \pm 0.01$ | $0.05 \pm 0.01$ | $0.21 \pm 0.02$ | $\mathbf{0.42 \pm 0.02}$ |
| Heartbeat | $0.7 \pm 0.04$ | $\emptyset$ | $0.71 \pm 0.04$ | $0.69 \pm 0.02$ | $0.7 \pm 0.04$ | $\mathbf{0.73 \pm 0.04}$ |
| NATOPS | $0.76 \pm 0.04$ | $0.73 \pm 0.05$ | $0.25 \pm 0.04$ | $0.41 \pm 0.04$ | $0.74 \pm 0.04$ | $\mathbf{0.77 \pm 0.04}$ |
| SelfRegulationSCP1 | $0.57 \pm 0.02$ | $0.56 \pm 0.03$ | $\emptyset$ | $0.57 \pm 0.05$ | $0.56 \pm 0.03$ | $\mathbf{0.61 \pm 0.02}$ |
| StandWalkJump | $0.5 \pm 0.15$ | $0.41 \pm 0.13$ | $\emptyset$ | $0.39 \pm 0.13$ | $0.3 \pm 0.13$ | $\mathbf{0.69 \pm 0.16}$ |
| UWaveGestureLibrary | $0.24 \pm 0.04$ | $0.21 \pm 0.05$ | $\emptyset$ | $0.13 \pm 0.02$ | $0.47 \pm 0.03$ | $\mathbf{0.64 \pm 0.04}$ |
| avg. rank (lower is better) | $3.29 \pm 1.02$ | $3.92 \pm 1.1$ | $5.3 \pm 1.31$ | $4.49 \pm 1.15$ | $2.66 \pm 1.18$ | $\mathbf{1.34 \pm 0.79}$ |

**Computation times.** During the classification experiment, we kept track of all metric computation time, which we average per metric in Table 6. Operator norm is the least efficient metric, followed by the Hilbert-Schmidt. The most efficient similarities are Martin and SOT; however, they performed poorly. SGOT and GOT are slightly less effective than Martin and SOT but much more efficient than Hilbert-Schmidt and Operator.

Table 6: Average computation time per similarity on all validation folds.

| Hilbert-Schmidt | Operator | Martin | SOT | GOT | SGOT |
|---|---|---|---|---|---|
| 4.96ms | 13.04ms | 0.02ms | 0.03ms | 0.14ms | 0.12ms |

**Critical diagram difference.** Considering results from all 10 iterations of the Monte Carlo cross-validation, we compute the critical diagram difference to statistically compare all metric performances based on rank. The diagram is depicted in Figure 7. The test significance level is set to 0.05. We use Friedman's test to reject the null hypothesis (All metrics' performances are similar) and compute the critical differences using the Nemenyi post-hoc test. Results show that SGOT is the best performer and statistically different from the second performer (SGOT).

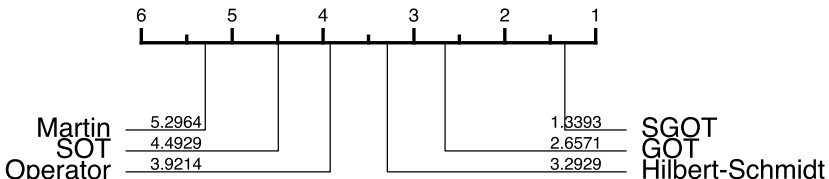

Figure 7: Critical diagram difference for comparing metrics' performances on a classification task. The classifiers are K-NN defined with the metrics: *Hilbert-Schmidt, Operator, Martin, SOT, GOT, and SGOT (ours)*. Computed from the performance of all 10 iterations of the Monte Carlo cross-validation. The test significance level is set to 0.05, the null hypothesis is rejected with Friedman's test, and the critical differences are computed using Nemenyi post-hoc test.

**2D T-SNE embeddings.** We illustrate the dimensionality reductions capabilities of the different similarity measures. We selected 5 datasets from fields including human activity recognition, motion recognition, and biomedical applications. For the 5 selected datasets and all similarities, dataset samples are embedded as a 2D vector with the T-distributed Stochastic Neighbor Embedding (T-SNE) Maaten & Hinton (2008) method fitted on the cross-distance matrix estimated with the similarities: Hilbert-Schmidt, Operator, Martin, SOT, GOT, and SGOT. Figure 8 displays the embeddings for all 5 datasets and metrics. On the Eigenworms and Epilepsy datasets, Martin is ill-defined, and the similarity values cannot be computed. No clusters or classes can be identified for Hilbert-Schmidt, Operator, Martin, and SOT. Regarding other OT-based metrics, GOT better identifies classes; however, they do not form distinct clusters as obtained with our metric SGOT.

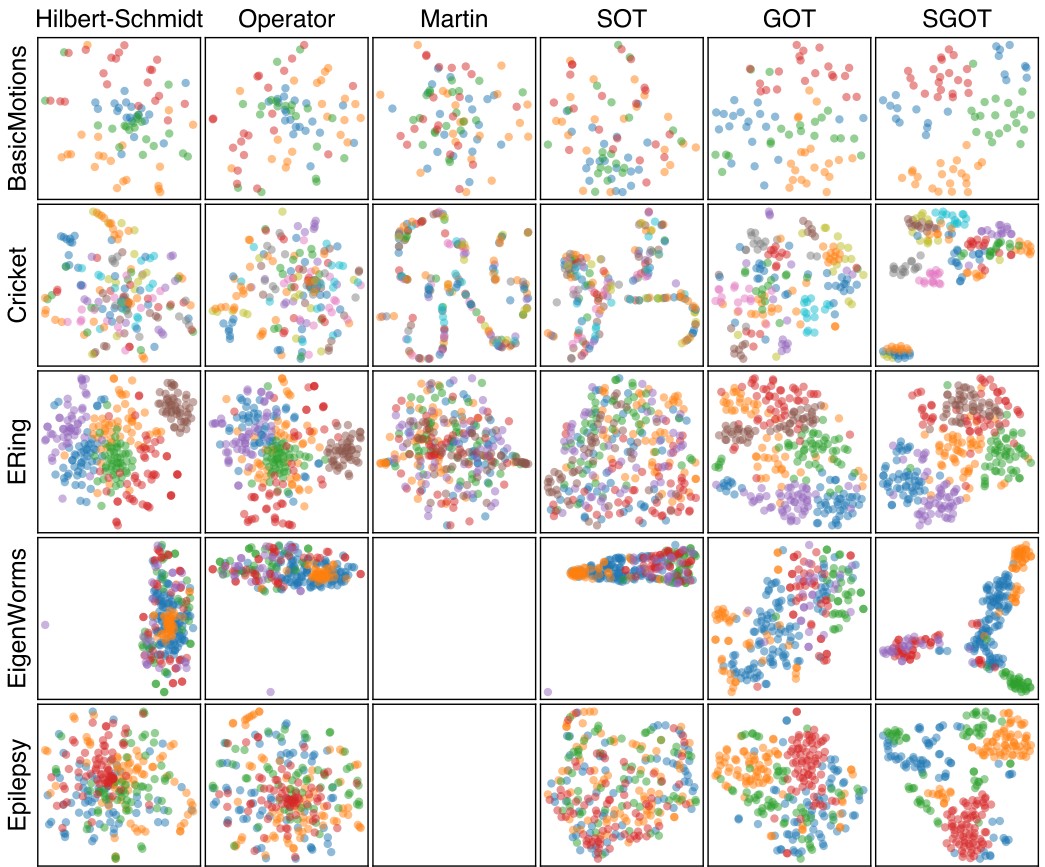

Figure 8: T-SNE 2D-embeddings of the classification datasets: *BasicMotions, Cricket, Ering, Eigen-Worms, Epilepsy* based on similarities: *Hilbert-Schmidt, Operator, Martin, SOT, GOT and SGOT (ours)*. Each point represents a dataset sample (a time series) whose color corresponds to its class. The Martin similarity is ill-defined on *EigenWorms* and *Epilepsy* datasets; thus, the corresponding T-SNEs are missing.

### G.3 GAUSSIAN KERNEL: RESULTS

**Experimental setup.** The experimental protocol follows the same procedure as for the linear kernel described in appendix G.1, with the following modifications:

1. **Estimating transfer operators with Gaussian kernels:** The linear kernel is replaced by the Gaussian kernel, $\kappa(x,y) \triangleq \exp(\|x-y\|^2/\sigma^2)$, such that for each datasets the kernel's scale parameter $\sigma$ is set according to the heuristic:

$$\sigma = \sqrt{(\text{number of dimension}) * (\text{context window length})}$$

2. **Experiment scalability.** Due to the computational cost of estimating transfer operators in infinite-dimensional kernel spaces, experiments are restricted to the five smallest datasets: BASICMOTIONS, ERING, EPILEPSY, FINGERMOVEMENTS, and NATOPS. Additionally, the nested Monte-Carlo cross-validation is limited to 5 iterations, and the Operator metric is omitted.

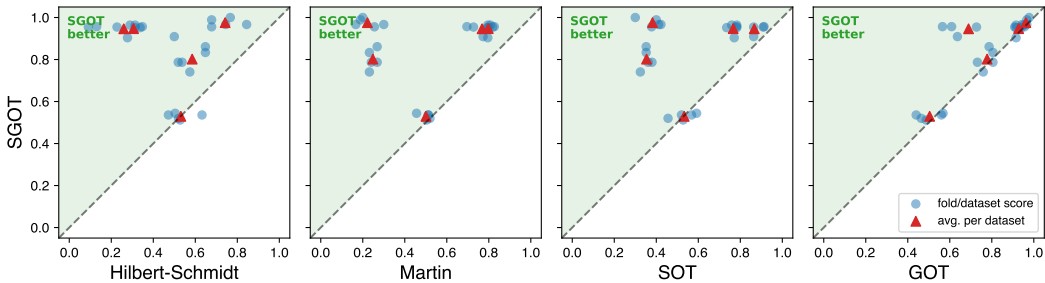

Figure 9: Classification performance (accuracy) comparison between SGOT and competitive metrics for transfer operators estimated with Gaussian kernels. Each point represents a dataset accuracy, with SGOT on the y-axis and the competing metrics on the x-axis.

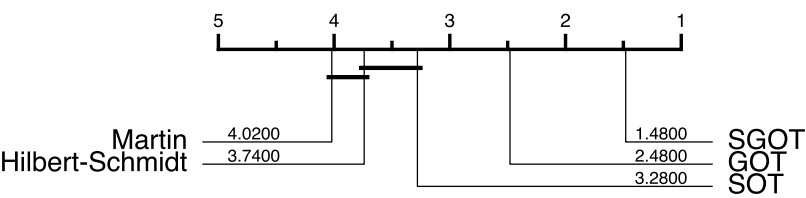

Figure 10: Critical diagram difference between metrics when transfer operators are estimated with the infinite dimensional Gaussian kernel. The classifiers are K-NN defined with the metrics: *Hilbert-Schmidt, Martin, SOT, GOT, and SGOT (ours)*. Computed from the performance of all 5 iterations of the Monte Carlo cross-validation. The test significance level is set to 0.05, the null hypothesis is rejected with Friedman's test, and the critical differences are computed using Nemenyi post-hoc test.

Table 7: Classification accuracy scores. The transfer operators are estimated with the infinite dimensional Gaussian kernel. Datasets on rows and similarities on columns. **Best** and second best performers are highlighted. Accuracy scores are denoted: $< mean > \pm < std >$.

|  | Hilbert-Schmidt | Martin | SOT | GOT | SGOT |
|---|---|---|---|---|---|
| BasicMotions | $0.26 \pm 0.17$ | $0.77 \pm 0.06$ | $0.87 \pm 0.05$ | $0.69 \pm 0.14$ | **$0.95 \pm 0.02$** |
| ERing | $0.74 \pm 0.07$ | $0.22 \pm 0.05$ | $0.38 \pm 0.05$ | $0.96 \pm 0.01$ | **$0.98 \pm 0.02$** |
| Epilepsy | $0.31 \pm 0.02$ | $0.8 \pm 0.01$ | $0.77 \pm 0.02$ | $0.93 \pm 0.02$ | **$0.95 \pm 0.02$** |
| FingerMovements | $0.53 \pm 0.06$ | $0.5 \pm 0.03$ | **$0.53 \pm 0.05$** | $0.5 \pm 0.06$ | $0.53 \pm 0.01$ |
| NATOPS | $0.59 \pm 0.06$ | $0.25 \pm 0.02$ | $0.35 \pm 0.02$ | $0.78 \pm 0.03$ | **$0.8 \pm 0.05$** |
| avg. rank (lower is better) | $3.74 \pm 1.27$ | $4.02 \pm 0.98$ | $3.28 \pm 1.15$ | $2.48 \pm 1.19$ | **$1.48 \pm 0.7$** |

**Results.** Table 7 reports the average accuracy over the 5-iterations of Monte-Carlo cross-validation for each dataset/metric pair. Figure 9 summarizes the comparative performance of our metric, SGOT, against several competitive alternatives: Hilbert–Schmidt, Martin, SOT, and GOT. Finally, Figure 10 presents the critical difference diagram obtained using a significance level of 0.05; the null hypothesis was rejected using Friedman's test, and pairwise comparisons were performed with the Nemenyi post-hoc test.

As in the linear-kernel setting, SGOT achieves the best performance across all five datasets, followed by GOT, which compares eigen-subspaces via optimal transport. This ranking is further supported by the critical difference diagram, which shows that SGOT statistically outperforms the other metrics. Moreover, in comparison to the linear kernel case, when transfer operators are estimated in an infinite-dimensional kernel space, classification accuracy increases with SGOT but decreases with

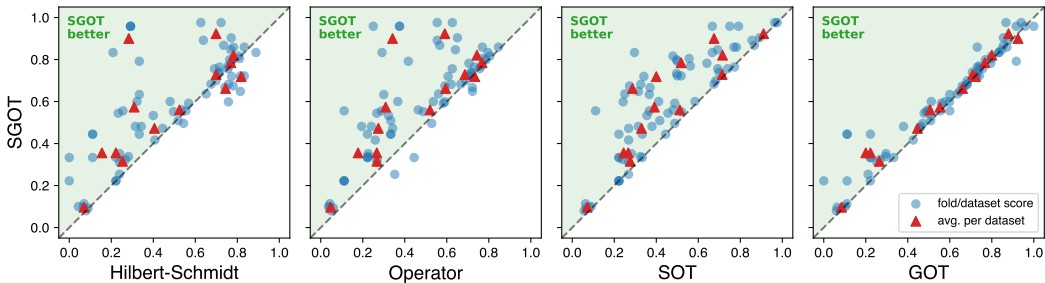

Figure 11: Classification performance (accuracy) comparison between SGOT and competitive metrics for transfer operators estimated with kernels defined by deep-functions. Each point represents a dataset accuracy, with SGOT on the y-axis and the competing metrics on the x-axis.

the Hilbert–Schmidt metric. This highlights the limitations of the Hilbert–Schmidt metric for high-dimensional representations, in contrast to the greater robustness exhibited by SGOT.

### G.4 DEEP-LEARNING EMBEDDINGS: RESULTS.

**Experimental setup.** The experimental protocol follows the same procedure as for the linear kernel described in appendix G.1, with the following modification:

- **Transfer operator estimation using kernels defined by learned deep features:** The linear kernel is replaced by a kernel of the form $\kappa(x, y) = \langle \phi_\theta(x), \phi_\theta(y) \rangle$ where $\phi_\theta : \mathcal{X} \mapsto \mathbb{R}^d$ is an embedding map parameterized by a neural network. The network architecture is a Multi-Layer Perceptron (MLP) with two hidden layers of dimension 128, a 32-dimensional output layer, and LeakyReLU activations with negative slope 0.01. For each dataset, after data augmentation using the context window, an embedding map is trained on the training set by following the strategy of (Kostic et al., 2024b) which is designed to learn invariant representations of time-homogeneous stochastic dynamical systems. Gradient descent is run for 4000 iterations; at each iteration, a window of length $\min(200, \text{n.samples}/2)$ is randomly extracted from a time series sampled uniformly from the training dataset.

Table 8: Classification accuracy scores. The transfer operators are estimated with a finite kernel defined with deep-functions. Datasets on rows and similarities on columns. **Best** and second best performers are highlighted. Accuracy scores are denoted: $< mean > \pm < std >$.

| | Hilbert-Schmidt | Operator | Martin | SOT | GOT | SGOT |
|---|---|---|---|---|---|---|
| AtrialFibrillation | $0.22 \pm 0.21$ | $0.18 \pm 0.1$ | $\underline{0.24 \pm 0.14}$ | $0.24 \pm 0.09$ | $0.2 \pm 0.21$ | $\mathbf{0.36 \pm 0.14}$ |
| BasicMotions | $0.28 \pm 0.05$ | $0.34 \pm 0.07$ | $0.81 \pm 0.06$ | $0.68 \pm 0.11$ | $\mathbf{0.92 \pm 0.06}$ | $\underline{0.9 \pm 0.08}$ |
| Cricket | $\mathbf{0.82 \pm 0.05}$ | $\underline{0.73 \pm 0.04}$ | $\emptyset$ | $0.4 \pm 0.1$ | $0.72 \pm 0.07$ | $0.72 \pm 0.07$ |
| ERing | $0.31 \pm 0.07$ | $0.31 \pm 0.04$ | $\emptyset$ | $0.39 \pm 0.04$ | $\underline{0.55 \pm 0.06}$ | $\mathbf{0.57 \pm 0.04}$ |
| EigenWorms | $0.78 \pm 0.03$ | $0.74 \pm 0.08$ | $\emptyset$ | $0.72 \pm 0.06$ | $\underline{0.8 \pm 0.06}$ | $\mathbf{0.82 \pm 0.05}$ |
| Epilepsy | $0.7 \pm 0.06$ | $0.59 \pm 0.05$ | $\emptyset$ | $\underline{0.91 \pm 0.06}$ | $0.88 \pm 0.04$ | $\mathbf{0.92 \pm 0.05}$ |
| FingerMovements | $\underline{0.52 \pm 0.04}$ | $0.52 \pm 0.05$ | $0.48 \pm 0.05$ | $0.51 \pm 0.04$ | $0.51 \pm 0.03$ | $\mathbf{0.56 \pm 0.04}$ |
| HandMovementDirection | $0.25 \pm 0.02$ | $0.27 \pm 0.05$ | $0.23 \pm 0.05$ | $\underline{0.28 \pm 0.03}$ | $0.26 \pm 0.04$ | $\mathbf{0.32 \pm 0.04}$ |
| Handwriting | $0.07 \pm 0.02$ | $0.05 \pm 0.01$ | $\emptyset$ | $0.07 \pm 0.01$ | $\underline{0.08 \pm 0.02}$ | $\mathbf{0.1 \pm 0.01}$ |
| Heartbeat | $0.7 \pm 0.02$ | $0.69 \pm 0.04$ | $\underline{0.72 \pm 0.04}$ | $0.71 \pm 0.04$ | $0.71 \pm 0.03$ | $\mathbf{0.73 \pm 0.04}$ |
| NATOPS | $0.41 \pm 0.08$ | $0.27 \pm 0.06$ | $0.3 \pm 0.06$ | $0.33 \pm 0.07$ | $\underline{0.45 \pm 0.05}$ | $\mathbf{0.47 \pm 0.03}$ |
| SelfRegulationSCP1 | $\underline{0.77 \pm 0.04}$ | $0.77 \pm 0.03$ | $0.51 \pm 0.03$ | $0.52 \pm 0.04$ | $0.77 \pm 0.03$ | $\mathbf{0.78 \pm 0.03}$ |
| StandWalkJump | $0.16 \pm 0.13$ | $0.27 \pm 0.13$ | $\emptyset$ | $\underline{0.27 \pm 0.1}$ | $0.22 \pm 0.08$ | $\mathbf{0.36 \pm 0.09}$ |
| UWaveGestureLibrary | $\mathbf{0.74 \pm 0.05}$ | $0.59 \pm 0.02$ | $\emptyset$ | $0.29 \pm 0.04$ | $0.66 \pm 0.05$ | $\underline{0.66 \pm 0.05}$ |
| avg. rank (lower is better) | $3.33 \pm 1.56$ | $4.14 \pm 1.27$ | $5.06 \pm 1.48$ | $3.84 \pm 1.34$ | $\underline{2.94 \pm 1.33}$ | $\mathbf{1.71 \pm 0.77}$ |

**Results.** Table 8 reports the average accuracy over the 5-iterations of Monte-Carlo cross-validation for each dataset/metric pair. Figure 11 summarizes the comparative performance of our metric, SGOT, against several competitive alternatives: Hilbert–Schmidt, Operator, SOT, and GOT. Finally,

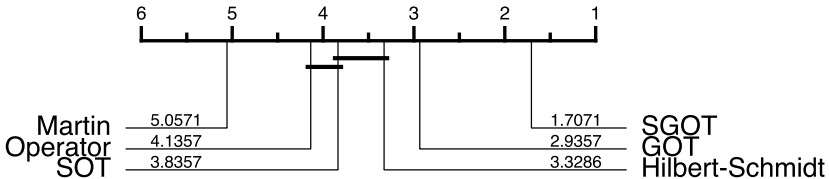

Figure 12: Critical diagram difference between metrics when transfer operators are estimated with a finite dimensional kernel defined with deep-functions. The classifiers are K-NN defined with the metrics: *Hilbert-Schmidt, Martin, SOT, GOT, and SGOT (ours)*. Computed from the performance of all 5 iterations of the Monte Carlo cross-validation. The test significance level is set to 0.05, the null hypothesis is rejected with Friedman's test, and the critical differences are computed using Nemenyi post-hoc test.

Figure 12 presents the critical difference diagram obtained using a significance level of 0.05; the null hypothesis was rejected using Friedman's test, and pairwise comparisons were performed with the Nemenyi post-hoc test.

When transfer operators are estimated using kernels built from learned deep features, the performance of all metrics decreases compared to the linear-kernel setting. This drop is attributable to the limited size of the datasets, which constrains the training of sufficiently representative deep features, a well-documented issue in operator embedding learning Lusch et al. (2018). Nevertheless, SGOT still achieves the highest performance on 11 out of the 14 datasets and ranks second or third on the remaining ones. This advantage is further confirmed by the critical-difference diagram, which indicates that SGOT statistically outperforms the competing metrics.

### G.5 SENSITIVITY ANALYSIS TO THE COST WEIGHTING PARAMETER

**Goal.**    We evaluate how the weighting parameter $\eta \in (0, 1)$ affects the classification performances. This parameter controls the trade-off between the eigenvalue term and the eigensubspace term in the cost of the Wasserstein SGOT metric defined in Theorem 1.

**Experimental setup.**    The experimental protocol follows the procedure described in appendix G.1. Transfer operators are estimated using linear kernels, and classification scores are computed for SGOT metrics with $\eta \in \{0.001, 0.01, 0.1, 0.2, 0.3, 0.4, 0.5, 0.6, 0.7, 0.8, 0.9, 0.99\}$. All results are obtained using a 5-iteration Monte-Carlo cross-validation.

**Results.**    Figure 13 reports the mean classification accuracy for each value of $\eta$. The vertical dashed line marks the heuristic value $\tilde{\eta} = (1 + f_c/\sqrt{2})^{-1} = (1 + f_{samp}/(2\sqrt{2}))^{-1}$, where $f_c = f_{samp}/2$ is the Nyquist frequency. This value corresponds to equal weighting of the eigenvalue and eigensubspace terms the cost of the Wasserstein SGOT metric (see Theorem 1).

Overall, Figure 13 shows that SGOT's performance varies smoothly with $\eta$, with optimal values typically favoring stronger emphasis on the eigensubspace cost. This trend suggests that the search range for $\eta$ can be substantially reduced in practice. In particular, the heuristic $\tilde{\eta}$ consistently lies near regions of high accuracy, and optimal values generally fall within $(0, \tilde{\eta})$ across datasets. Consequently, $\tilde{\eta}$ provides both a practical initial choice and a principled bound for restricting the grid-search budget.

## H    BARYCENTER OF DYNAMICAL SYSTEMS

### H.1    INTERPOLATION BETWEEN 1D DYNAMICAL SYSTEMS

**Experimental settings.**    In this experiment, we compare the interpolation between dynamical systems through weighted Fréchet barycenters of their Koopman operators, estimated with a linear kernel, for different metrics. The two systems are linear oscillatory systems, each being the sum of two

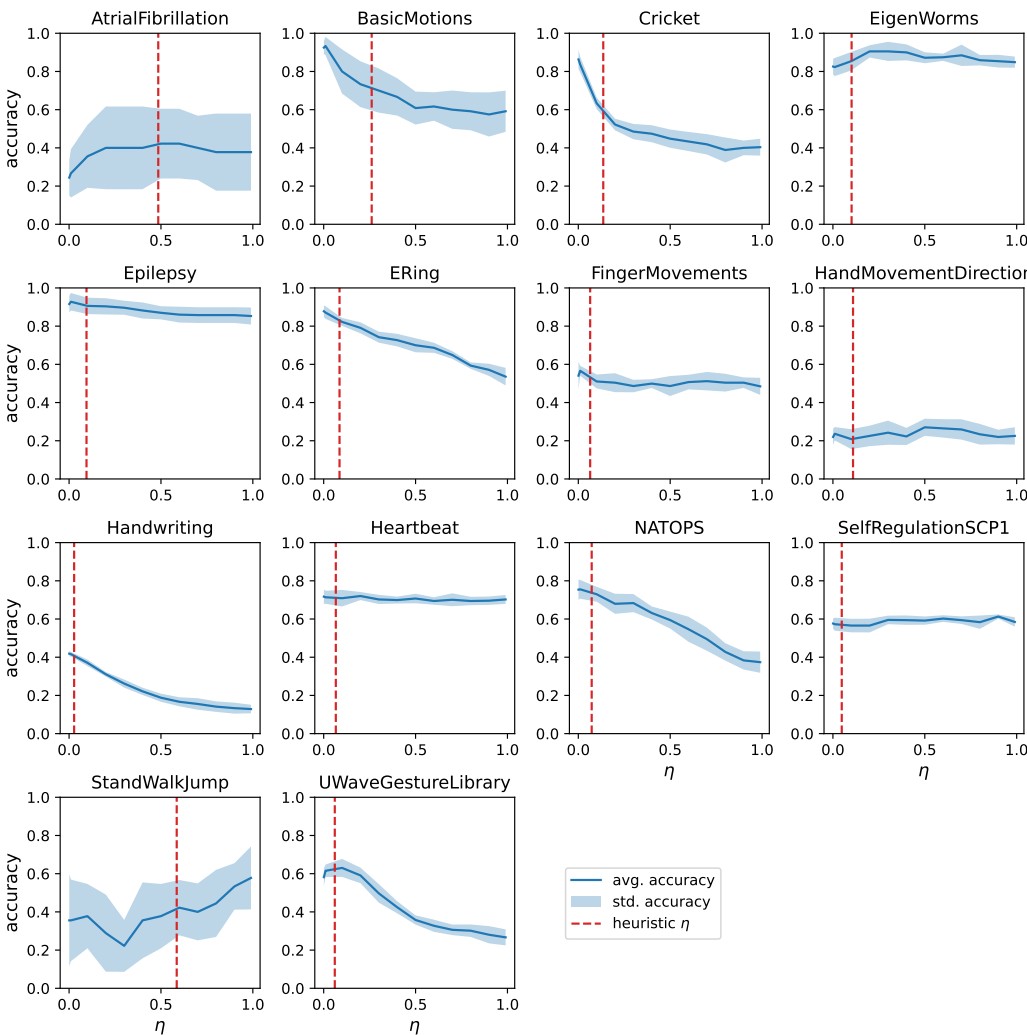

Figure 13: Sensitivity of SGOT's classification accuracy to the parameter $\eta$ across all datasets. The dark blue curve shows the mean accuracy over the 5-iteration Monte-Carlo cross-validation, and the light blue band indicates one standard deviation. The vertical dotted red lines mark the heuristic values $\tilde{\eta} = (1 + f_{samp}/(2\sqrt{2}))^{-1}$ which give equal weight to the eigenvalue and eigensubspace cost terms in the SGOT definition (see Theorem 1).

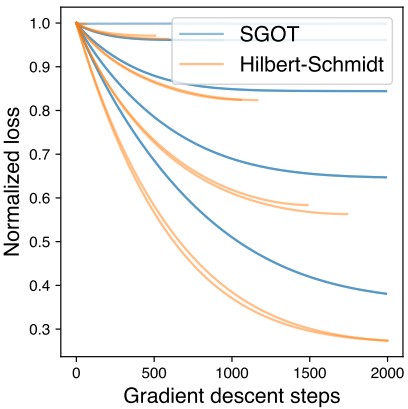

Figure 14: Normalized loss value per gradient descent step for the constrained Hilbert-Schmidt (b) and SGOT (c) barycenter for any interpolation step.

simple harmonic oscillators with different frequencies and decay rates, and additive Gaussian noise. The first system $\mathbf{T}^{(0)}$ combines a convergent low frequency oscillator ($\omega = 1.7$Hz, $\rho = -0.2$, amplitude=1.0) with a divergent high frequency oscillator ($\omega = 4.7$Hz, $\rho = 0.2$, amplitude=0.2) . The second system $\mathbf{T}^{(1)}$ is reversed; it combines a divergent low frequency oscillator ($\omega = 0.7$Hz, $\rho = 0.2$, amplitude=1) with a convergent high frequency oscillator ($\omega = 11.3$Hz, $\rho = -0.2$, amplitude=1). Both systems are noisy with a Gaussian noise with variance $\sigma^2 = 1e - 4$. The systems Koopman operators are estimated with the RRR methods (Kostic et al., 2022) from trajectories of length 5000 samples at 800Hz. RRR estimator is set to estimate a rank 4 operator with context window of 400 samples, a linear kernel, and Tikhonov regularization of $1e - 8$. The interpolation is controlled by a ratio parameter $\gamma$ going from 0 to 1 in 0.1 steps. At each interpolation step, the weights in the Fréchet mean (see equation 7) are $(1 - \gamma, \gamma)$. We compare (a) the Hilbert-Schmidt metric without spectral decomposition constraints given by $\mathbf{T}_{bar} = (1 - \gamma)\mathbf{T}^{(0)} + \gamma\mathbf{T}^{(1)}$, (b) the Hilbert-Schmidt metric with spectral decomposition constraints, and (c) our proposed metric SGOT. For (b) and (c), the barycentric operators are estimated with the proposed optimization scheme described in appendix D. In both cases, the initialization of the barycenter corresponds to the average of eigenvalues and eigenfunctions. For the Hilbert-Schmidt (b), the barycenter optimizer is set with a $3e - 5$ learning rate, a maximal number of iterations of 2000, with 1 gradient descent per coordinate at each iteration, the stopping criteria corresponds to a consecutive metric error lower than $1e - 6$. For the SGOT (c), $\eta = 0.9$, and the barycenter optimizer is set with a $1e - 2$ learning rate, a maximal number of iterations of 200, with 10 gradient descent per coordinate at each iteration, the stopping criteria corresponds to a consecutive metric error lower than $1e - 6$. Finally, for displaying the predicted signals from the interpolated barycenter in Figure 4, all predictions started with the same initialization set, being the first 400 samples of a linear system that is the sum of 4 harmonic oscillators of the systems to interpolate.

**Additional results.** For all constrained Hilbert-Schmidt (b) and SGOT (c) interpolated barycenters, we kept track of the decay rate and frequency of the two associated harmonic oscillators, the loss values, and the computation time. Figure 14 displays the normalized losses decrease per gradient descent step for each metric and interpolation step. The representation is in gradient descent step as the number of iterations and gradient descent step per cycle differ from metric to metric. In Figure 15 we display the decays and frequencies of the interpolated barycenters. In particular, Figure 14 shows that the barycenter algorithm has converged for any metric and interpolation step. However, Figure 15 shows that constrained Hilbert-barycenter (b) remains stuck in a local minima close to the initialization. In contrast, the SGOT barycenter perfectly (linearly) interpolates the decay and frequency between the source and target systems. Furthermore, the average computation time per gradient descent step is 13.11ms for the constrained Hilbert-Schmidt, while being 2.29ms for our metric SGOT, meaning that the barycenter algorithm is approximately 6x faster with the SGOT metric compared to the Hilbert-Schmidt.

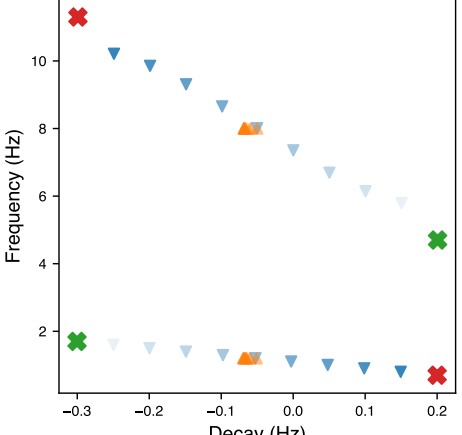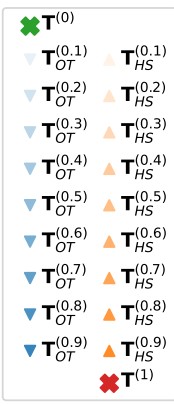

Figure 15: Decay rates and frequencies of the two harmonic oscillators associated with the interpolated barycenters for the constrained Hilbert-Schmidt (b) and the SGOT (c) metrics. The source system harmonic oscillator is in red, and the target system harmonic oscillator is in green.

## H.2 FLUID DYNAMIC INTERPOLATION

**Experimental settings.** We aim to compute the barycenter of two fluid dynamics systems. To that end, we consider the *Flow past a bluff object* dataset (Tali et al., 2025), which gathers trajectories of time-varying 2D velocity and pressure fields of incompressible Navier-Stokes fluids flowing around static objects. We select two trajectories, one with a cylinder object (Huggingface dataset file: harmonic/93) and the other with a triangular object (Huggingface dataset file: skeleton/48). For each trajectory, we only kept the velocity field along the flow direction, leading to trajectories containing 242 samples of 1024x256 grids, which we down-sampled to grids with a 256x64 resolution. We estimate a Koopman operator with a linear kernel using the RRR method from each trajectory sampled at 100Hz with a context window of 1, and a Tikhonov regularization of 1. The operators are restricted to the fourth leading eigenvalues and eigenfunctions. We compute the SGOT barycenter with the optimization scheme described in Appendix D with an initialization being the average of eigenvalues and eigenfunctions. For the SGOT (c), $\eta = 0.01$, and the barycenter optimizer is set with a $1e-4$ learning rate, a maximal number of iterations of 100, with 10 gradient descent per coordinate at each iteration, the stopping criteria corresponds to a consecutive metric error lower than $1e-6$.

