# OpenReview forum: "A Spectral-Grassmann Wasserstein metric for operator representations of dynamical systems"
_ICLR.cc/2026/Conference — ICLR 2026 Poster_

### Official Review · Reviewer_AGY8 · 2025-10-18

**Soundness:** 3
**Presentation:** 3
**Contribution:** 3
**Rating:** 6
**Confidence:** 3

**Summary:**

The authors proposed a representation of each dynamical system as a probability distribution over its joint operator eigenvalues and spectral projectors on the Grassmann manifold. They defined a metric between systems using optimal transport with a cost function that balances spectral information and geometric information, i.e., the eigenspace distances on the Grassmann manifold.

**Strengths:**

The paper introduces a new metric called Spectral-Grassmann Optimal Transport (SGOT) distance for comparing dynamical systems through their Koopman or transfer operator representations. Its main strength lies in integrating spectral and subspace information within an optimal transport framework, which is an original formulation that helps address several limitations of prior operator-based metrics. The theoretical analysis includes finite-sample guarantees, and the presentation is generally clear. The experiments provide evidence supporting the method’s potential usefulness in practical applications.

**Weaknesses:**

(1) All experiments use linear kernels. There is no exploration of RBF, polynomial, or learned kernels;
(2) Assumption (A3) requires a shared RKHS $\mathcal{H}$ that simultaneously contains the low-rank images of all Koopman operators. It is unclear how one could construct or validate such a space in real applications. Different dynamical systems may have very different scales and smoothness properties;

**Questions:**

(1) In Theorem 2, how would you verify the conditions $[(C_x^k)^{\dagger}]^{\frac{\alpha - 1}{2}}T_k$ and $\lambda_i(C_x^k)^{\dagger}$ by data?
(2) Theo theory replies on assumptions A1-A3, which is convenient to compare between the operator from different systems, but are they assured in practical cases?
(3) When $\lambda_j$ has multiplicity $m_j>1$, how is the space $V_j$ uniquely defined?
(4) Do we also need to show that $V_j$ is independent of basis choice?
(5) Is $d_{\mathcal{S}}(T, T')$ independent of the choice of bi-orthogonal basis used to represent spectral decompositions? It is unclear whether $d_{\mathcal S}(T, T')$ is intrinsic, i.e., independent of the choice of bi-orthogonal basis used to represent the spectral decomposition when eigenvalues have multiplicity > 1.

---

> ### Author Response · Authors · 2025-11-20
>
> We thank the reviewer for the positive comments in the strengths section. We first invite the reviewer to consult the global response; in what follows, we address the specific weaknesses and questions raised.
>
> ### Additional experiments
> > - All experiments use linear kernels. There is no exploration of RBF, polynomial, or learned kernels;
>
> We have extended our experiments to include operator estimation with RBF kernels (Appendix G.3) and kernels based on learned deep features (Appendix G.4). As reported in the Global Response tables, SGOT consistently outperforms competing metrics across most datasets and estimation methods, whether linear, RBF, or deep. These additional experiments demonstrate that SGOT’s performance is robust to the choice of kernel.
>
> ### Assumption implication
> > - Assumption (A3) requires a shared RKHS  that simultaneously contains the low-rank images of all Koopman operators. It is unclear how one could construct or validate such a space in real applications. Different dynamical systems may have very different scales and smoothness properties.
> > - The theory replies on assumptions A1-A3, which is convenient to compare between the operator from different systems, but are they assured in practical cases?
>
> We thank the reviewer for these thoughtful questions concerning assumptions (A1)–(A3). We have clarified the implications of these assumptions in the revised manuscript, and we address them individually below.
>
> **A1** poses the setting: we compare stochastic dynamical systems via observations of their trajectories, which may be sampled with different time lags. This is not really an assumption but a description of our framework.
>
>
> **A2** is a mild and standard assumption in the *stochastic* setting we consider. For many stochastic dynamical systems—diffusions, Langevin dynamics, and SPDEs with non-degenerate noise—the associated transfer operators on $L^2_\pi$ are typically compact, and in many cases even Hilbert–Schmidt, because the transition kernel admits a square-integrable density. As a result, these operators possess accurate finite-rank approximations. (See Da Prato & Zabczyk, *Stochastic Equations in Infinite Dimensions*, 1992; Hairer, *Ergodic Properties of Markov Processes*, 2006; Khasminskii, *Stochastic Stability of Differential Equations*, 2012.)
>
>
> **A3** In most applications, this requirement is mild. As stated in lines 167-173 of our original submission, prototypical case is comparison of Langevin stochastic processes, under suitable assumptions on the potential, via RKHS of RBF kernel . We invite the reviewer to refer to the paragraph “2. Why a common functional space (A3) is needed” in the global response, where we discuss in detail how to select an appropriate shared functional space and how to diagnose potential violations of this assumption. In practice, any universal bounded kernel, such as the RBF, naturally induces a common functional space. Alternatively, a shared space can be obtained through representation-learning methods that produce a common embedding for all operators under comparison. Moreover, potential violations of assumption A3 can be detected by estimating the metric distortion between the chosen common subspace and the true operator domain using the estimator proposed in (Kostic et al. 2023).
>
>
> > - In Theorem 2, how would you verify the conditions $[(C^k_x)^\dagger]^{\frac{\alpha-1}{2}}T_k$ and $\lambda_i((C^k_x)^\dagger)$ by data?
>
> We address the verification of each term individually:
>
> $[(C^k_x)^\dagger]^{\frac{\alpha-1}{2}}T_k$: This condition corresponds to a source condition on the transfer operator, quntifying in terms of $\alpha$ how well the dominant part of the operator is well aligned with the chosen common functional subspace in order to analyze the rate.
>
> $\lambda_i((C^k_x)^\dagger)$: These quantities are properties of the common functional space used for operator estimation; they do not depend on the unknown transfer operators themselves. In practice, they can be estimated directly from data, both in RKHS and learned feature spaces.

---

> ### Author Response · Authors · 2025-11-20
>
> ### Comparing subspaces of eigenvalue with multiplicities
> > - When $\lambda_i$ has multiplicity , how is the space  $V_i$ uniquely defined? (4) Do we also need to show that $V_i$  is independent of basis choice? (5) Is $d_S$  independent of the choice of bi-orthogonal basis used to represent spectral decompositions? It is unclear whether $d_S$ is intrinsic, i.e., independent of the choice of bi-orthogonal basis used to represent the spectral decomposition when eigenvalues have multiplicity $>$ 1.
>
> We thank the reviewer for raising this important point, which goes to the core of SGOT’s design. SGOT is intrinsically independent of the choice of basis used in the spectral decomposition. Its cost function has two additive components: one comparing eigenvalues and the other comparing eigenspaces.
>
> The eigenspace comparison term is defined using a metric on the Grassmann manifold, which naturally compares subspaces independently of their specific representation basis. Moreover, this metric is capable of comparing subspaces of different dimensions, ensuring robustness when eigenvalues have multiplicity greater than one.
>
> As a result, SGOT is fully intrinsic: it does not depend on the choice of basis and can consistently compare any eigensubspaces, regardless of multiplicity or dimension.

---

> > ### Comment · Reviewer_AGY8 · 2025-11-25
> >
> > Thank you for addressing my concerns, I appreciate it! I will raise my score.

---

> > > ### Author Response · Authors · 2025-11-27
> > >
> > > Thank you for your positive follow-up. We appreciate your careful reading of our work and your constructive comments. If you have any further concerns or suggestions, please feel free to let us know. We would be happy to clarify anything.

---

### Official Review · Reviewer_vMWN · 2025-10-23

**Soundness:** 3
**Presentation:** 3
**Contribution:** 3
**Rating:** 6
**Confidence:** 3

**Summary:**

This paper addresses the problem of quantifying and comparing the geometry of dynamical systems from trajectory data, a central challenge in machine learning for dynamical systems. The authors build on the Koopman and transfer operator framework, which provides a linear representation of nonlinear dynamics via spectral decomposition.
They propose a new operator-based metric that represents each system as a joint distribution over operator eigenvalues and spectral projectors, and defines distances between systems using optimal transport theory. The resulting metric is invariant to the sampling frequency of trajectories, computationally efficient, and theoretically supported by finite-sample convergence guarantees. Furthermore, it allows the computation of Fréchet means, enabling smooth interpolation between dynamical systems.
Comprehensive experiments on both synthetic and real-world datasets demonstrate that the proposed method consistently outperforms standard operator-based distances in machine learning tasks such as dimensionality reduction and classification, while also yielding meaningful interpolations between dynamical systems.

**Strengths:**

This paper tackles a fundamental and timely problem in machine learning for dynamical systems — defining a theoretically sound, interpretable, and computationally tractable metric between data-driven operator representations of nonlinear and stochastic dynamics. The proposed Spectral-Grassmann Optimal Transport (SGOT) framework elegantly combines spectral theory, Grassmannian geometry, and optimal transport, yielding a true metric that is invariant to sampling frequency and robust to operator estimation errors. The method is mathematically principled, supported by finite-sample convergence guarantees, and remains computationally efficient, overcoming the typical trade-off between theoretical rigor and practicality. Extensive experiments demonstrate that SGOT outperforms existing operator-based similarities in both unsupervised and supervised learning tasks, and its ability to compute Fréchet means enables meaningful interpolation between dynamical systems — a capability with broad implications for model comparison, system identification, and generative modeling of dynamics. Overall, the paper makes a significant and original contribution to bridging operator-theoretic dynamical systems and modern machine learning.

**Weaknesses:**

While the proposed SGOT framework is theoretically elegant and empirically strong, the paper could be further strengthened by a clearer characterization of its practical limits. In particular, it remains unclear how strongly nonlinear systems the approach can accurately handle, and how robust the method is under varying levels of stochastic noise in trajectory data. Although the finite-sample convergence guarantees are valuable, the mathematical conditions quantifying robustness to nonlinearity and noise are not explicitly derived. Providing such analyses — for example, bounds on operator estimation errors or perturbation stability with respect to noise amplitude — would help clarify the method’s applicability range and further enhance its theoretical completeness.

**Questions:**

1.	To what extent do the theoretical properties of the Koopman operator, such as boundedness or continuity, play a critical role in the proposed SGOT framework?
2.	Regarding the domain and codomain of the Koopman operator, what kind of function spaces are suitable for ensuring the validity of the spectral–Grassmann representation? For example, does the analysis require specific regularity, integrability, or compactness assumptions on observables?
3.	Can the proposed approach be extended to highly complex dynamical systems, such as stochastic nonlinear partial differential equations (SPDEs), where both randomness and infinite-dimensional state spaces come into play? If so, what are the main theoretical or computational challenges expected in such settings?

---

> ### Author Response · Authors · 2025-11-20
>
> We thank the reviewer for the positive comments in the strengths section. We first invite the reviewer to consult the global response; in what follows, we address the specific weaknesses and questions raised.
>
> ### Noise sensitivity
> > - While the proposed SGOT framework is theoretically elegant and empirically strong, the paper could be further strengthened by a clearer characterization of its practical limits. In particular, it remains unclear how strongly nonlinear systems the approach can accurately handle, and how robust the method is under varying levels of stochastic noise in trajectory data.
>
> > - Although the finite-sample convergence guarantees are valuable, the mathematical conditions quantifying robustness to nonlinearity and noise are not explicitly derived. Providing such analyses — for example, bounds on operator estimation errors or perturbation stability with respect to noise amplitude — would help clarify the method’s applicability range and further enhance its theoretical completeness.
>
> We thank the reviewer for raising the question of SGOT’s sensitivity to noise. We would first like to clarify that SGOT is designed as a framework for **measuring similarity between two already estimated operators**, rather than as a method for estimating Koopman operators themselves. Consequently, the impact of noise on SGOT is indirect: noise affects the operator estimates provided to SGOT, but does not alter the SGOT formulation. In particular, SGOT can interface seamlessly with any operator estimation approach, including kernels or neural networks, with no modification to its core metric.
> The problem of handling noise during operator estimation has been studied extensively in prior work (e.g., Sinha et al., 2020; Hao et al., 2024). We made this distinction clearer in the revised manuscript.
>
> *Sinha, S., Huang, B., & Vaidya, U. (2020). On robust computation of koopman operator and prediction in random dynamical systems. Journal of Nonlinear Science, 30(5), 2057-2090.*
>
> *Hao, W., Upadhyay, D., & Mou, S. Deep Koopman Learning using Noisy Data. Transactions on Machine Learning Research.*
>
> ### Assumption implication
> > - To what extent do the theoretical properties of the Koopman operator, such as boundedness or continuity, play a critical role in the proposed SGOT framework?
>
> > - Regarding the domain and codomain of the Koopman operator, what kind of function spaces are suitable for ensuring the validity of the spectral–Grassmann representation? For example, does the analysis require specific regularity, integrability, or compactness assumptions on observables?
>
> We thank the reviewer for raising questions regarding the well-posedness and comparability of the operators. Theoretical properties, such as boundedness, continuity, or compactness, of an operator and its domain ensure that the operator is well-defined and admits a spectral decomposition that low-rank estimators can reliably approximate. These assumptions are standard in spectral operator theory and are not specific to SGOT. More importantly, independent of the chosen comparison method (SGOT or any alternative), a common functional space is required to compare transfer operators meaningfully. In the revised manuscript, we clarify this point and we invite the reviewer to refer to the paragraph “2. Why a common functional space (A3) is needed” in the global response, where we detail how such a shared space can be constructed in practice. In most applications, this requirement is mild. Any universal bounded kernel, such as the RBF kernel, naturally induces a common function space. Alternatively, a shared space can be obtained through representation-learning methods that produce a common embedding for all operators under comparison.

---

> ### Author Response · Authors · 2025-11-20
>
> ### Extension to SPDE
> > - Can the proposed approach be extended to highly complex dynamical systems, such as stochastic nonlinear partial differential equations (SPDEs), where both randomness and infinite-dimensional state spaces come into play? If so, what are the main theoretical or computational challenges expected in such settings?
>
> We thank the reviewer for raising the question of extending SGOT to SPDEs. In fact, whenever SPDE is generating a Markov semigroup $(T_t)\_{t\ge 0}$ on the infnite dimensional state space one can direcly extend our work. For instance, consider the **stochastic heat equation** on the interval $(0,1)$, see Da Prato & Zabczyk, Stochastic Equations in Infinite Dimensions (1992). The natural state space of observables is the infinite-dimensional Hilbert space $L^{2}(0,1)$. Nevertheless, the eigenvalues of the generator of the associated Markov semigroup $(T_t)_{t\ge 0}$ satisfy $\lambda_k = - (k\pi)^2$. Consequently, the spectrum of the transfer operator $T_1$ decays exponentially fast: $e^{\lambda_k} = e^{-(k\pi)^2}$. Thus $T_1$ is intrinsically low-rank in an effective sense, despite acting on an infinite-dimensional space. Furthermore, using RBF RKHS with $L^2(0,1)$ norm, one can in principle build RRR estiamtors and apply SGOT metric, with additonal complexity of approximating the kernel.

---

### Official Review · Reviewer_WDpQ · 2025-10-30

**Soundness:** 2
**Presentation:** 2
**Contribution:** 3
**Rating:** 6
**Confidence:** 4

**Summary:**

The paper introduces Spectral-Grassmann Optimal Transport (SGOT), a metric that represents each linear operator by its eigenvalue subspace atoms and compares operators by a Wasserstein distance. The authors show how to compute this metric via kernel representations, provide statistical guarantees for finite-sample estimation, and propose a parametric formulation to compute barycenters in practice. Empirical results on synthetic systems and fluid flow data illustrate that SGOT responds to changes in frequency, decay and subspace structure compared with the baselines, better performances in ML tasks, and can produce visually meaningful interpolated modes.

**Strengths:**

The paper proposes a coherent, high-level framework that combines spectral information (eigenvalues) with subspace geometry (Grassmann) under an optimal-transport view, which is an original and conceptually clean way to compare linear dynamical operators.
Methodologically, the authors introduce a concrete, computable metric (SGOT), give a parametric barycenter formulation and an optimization scheme that bridges the abstract definition and practical computation.
Experiments demonstrate the metric’s sensitivity to controlled changes, its usefulness for ML tasks (classification/embedding), and the interpolation behavior on physical flow data.

**Weaknesses:**

The paper’s choice of distances is only one possible design: Eqs. (4)  and (6) are reasonable, but the authors do not justify them or compare to clear alternatives. The empirical claims may need clear explanations. Finally, there are some poor presentation issues. Please see the questions below in detail.

**Questions:**

1. I agree that summing an eigenvalue distance and a subspace distance in Eq. (4) is reasonable, but it’s just one possible choice. For example, distances based on Koopman trace/determinant kernels (Fujii et al., 2017) could also be defined. Please explain why you chose the current form and mention other viable alternatives.

2. Similarly, the second term in Eq. (6) is only one concrete realization of a subspace distance. Eq. (6) evaluates subspace differences as a quadratic form with kernel matrices and effectively uses a trace kernel-like representation. Why did you select this particular form? Please discuss alternatives (e.g. principal angles) and comment on related prior work (e.g. Kawahara et al., 2016 [a]), which is not cited but is relevant to distances in Koopman spectral decomposition.

3. The results in Fig. 1 can be interpreted, but the current explanation lacks enough information for a reader to judge whether SGOT is truly better. In particular, please justify the claim that “grows linearly with the shifts almost everywhere” is a desirable property and explain why this behavior indicates superiority.

4. Figure 5 may visualize the intermediate modes computed by SGOT, but a barycenter does not necessarily correspond to a physically realizable intermediate system. To make Fig. 5 more convincing, please evaluate the barycenter against concrete metrics that relate to physical realism. For example, mode frequencies, decay rates, and reconstruction/forecast RMSE.

5. The experiments report that Koopman operators are estimated with a linear kernel (including Appendix H). Please state why you chose a linear kernel (interpretability, computational reasons, or data suitability) and discuss the relationship with non-linear kernels such as RBF.

Minor points
6. From Section 2 onward you use \cite where \citep would be appropriate.

7. Line 308 contains two periods in a row.

8. Figure 2 appears after Fig. 5 and a label seems to be “ilbert”.

---

> ### Author Response · Authors · 2025-11-20
>
> We thank the reviewer for the positive comments in the strengths section. We first invite the reviewer to consult the global response; in what follows, we address the specific weaknesses and questions raised.
>
> ### Metric design choices 1/2
> > - The paper’s choice of distances is only one possible design: Eqs. (4) and (6) are reasonable, but the authors do not justify them or compare to clear alternatives. The empirical claims may need clear explanations. Finally, there are some poor presentation issues. Please see the questions below in detail.
>
> > - I agree that summing an eigenvalue distance and a subspace distance in Eq. (4) is reasonable, but it’s just one possible choice. For example, distances based on Koopman trace/determinant kernels (Fujii et al., 2017) could also be defined. Please explain why you chose the current form and mention other viable alternatives.
>
> We thank the reviewer for raising these important points regarding the design choices behind our metric.
>
> SGOT is built around a cost function that compares pairs of eigenvalues and eigensubspaces. This choice aims to covering a wide variety of dynamical systems while preserving interpretability: the final distance decomposes transparently into eigenvalue and eigensubspace components. In contrast to prior OT-based distances, either comparing only eigenvalues (Redman et al., 2024) or requiring self-adjoint operators (Antonini et al., 2021), our formulation enables an exact and well-defined comparison for the broader class of non-defective operators, with possible extension to defective ones as discussed in lines 197-199 of our original submission.
>
> As the reviewer mentioned, another line of work proposes distances based on trace/determinant kernels (Chaudhry & Vidal, 2013; Fujii et al., 2017; Ishikawa et al., 2018), which we discussed in lines 56-57 of our submission. These approaches compare dynamical systems through observability subspaces, producing kernel evaluations that mix eigenvalue and eigensubspace information in a single quantity through a geometric sum. This blending limits the interpretability of the contributions of each spectral component in comparison to the explicit decomposition offered by SGOT. Moreover, these kernels typically require careful handling of trajectory initial conditions, whereas SGOT does not rely on or assume any particular initialization, simplifying its use in practice.
>
> *William Redman, Juan Bello-Rivas, Maria Fonoberova, Ryan Mohr, Yannis Kevrekidis, and Igor Mezic. Identifying equivalent training dynamics. Advances in Neural Information Processing Systems, 37:23603–23629, 2024.*
>
> *Paolo Antonini and Fabio Cavalletti. Geometry of grassmannians and optimal transport of quantum states. arXiv preprint arXiv:2104.02616, 2021.*
>
> *Keisuke Fujii, Yuki Inaba, and Yoshinobu Kawahara. Koopman spectral kernels for comparing complex dynamics: Application to multiagent sport plays. In Joint European Conference on Machine Learning and Knowledge Discovery in Databases, pp. 127–139. Springer, 2017.*
>
> *Isao Ishikawa, Keisuke Fujii, Masahiro Ikeda, Yuka Hashimoto, and Yoshinobu Kawahara. Metric on non-linear dynamical systems with perron-frobenius operators. Advances in Neural Information Processing Systems, 31, 2018.*
>
> *Rizwan Chaudhry and Rene Vidal. Initial-state invariant binet-cauchy kernels for the comparison of linear dynamical systems. In 52nd IEEE Conference on Decision and Control, pp. 5377–5384. IEEE, 2013.*

---

> ### Author Response · Authors · 2025-11-20
>
> ### Metric design choices 2/2
> > - Similarly, the second term in Eq. (6) is only one concrete realization of a subspace distance. Eq. (6) evaluates subspace differences as a quadratic form with kernel matrices and effectively uses a trace kernel-like representation. Why did you select this particular form? Please discuss alternatives (e.g. principal angles) and comment on related prior work (e.g. Kawahara et al., 2016 [a]), which is not cited but is relevant to distances in Koopman spectral decomposition.
>
> We thank the reviewer for pointing out the relevant work by (Kawahara et al. 2016), which we will add to the related works, is focused on developing an estimator of Koopman spectral decompositions for deterministic systems, and provides and experiment where the principal angles are used to classify human motion dataset.
>
> Regarding the choice of subspace metric in Eq. (6), we use the chordal metric, defined through the Hilbert–Schmidt norm of the difference between the orthogonal projectors onto the two subspaces. We selected this metric because it naturally accommodates subspaces of different dimensions, which is essential in our setting: when spectral decompositions are matched via optimal transport, eigenspaces may have unequal multiplicities. In contrast, classical metrics based on principal angles are inherently defined only for subspaces of the same dimension and require additional conventions or padding procedures when dimensions differ, making them less suitable here. For completeness, when subspaces have the same dimension k, the chordal metric is linked to principal angles $\theta_i$ via $\|P_V-P_U\|_F = \sqrt{2 \sum_{i \in [k]} \sin^2(\theta_i)}$.
>
> Nevertheless, SGOT is not tied to this specific choice. The cost function can incorporate other Grassmannian metrics, depending on the prior geometry one wishes to emphasize. For comparisons involving subspaces of different dimensions, general constructions such as those proposed by Ye & Lim (2016) can be used. We have added a discussion of these alternatives in the main text and included a summary of commonly used Grassmannian metrics in Appendix C.3.
>
> *Ke Ye and Lek-Heng Lim. Schubert varieties and distances between subspaces of different dimensions.SIAM Journal on Matrix Analysis and Applications, 37(3):1176–1197, 2016.*

---

> ### Author Response · Authors · 2025-11-20
>
> ### Metric behavior interpretation
> > - The results in Fig. 1 can be interpreted, but the current explanation lacks enough information for a reader to judge whether SGOT is truly better. In particular, please justify the claim that “grows linearly with the shifts almost everywhere” is a desirable property and explain why this behavior indicates superiority.
>
> While the “linear growth” of SGOT with respect to simple shifts is not a critical property, it does illustrate that, for many well-understood perturbations, the divergence increases with the magnitude of the shift, a behavior that is not observed in several competing metrics. More importantly, SGOT exhibits continuity and smoothness with respect to these shifts. This smoothness provides a more well-behaved geometric structure, which is beneficial for optimization-based applications, such as barycenter computation (as demonstrated in the paper). It also opens the door to potential future uses, for example as a regularization term in learning tasks. We will clarified this and its practical implications in the revised manuscript.
>
>
> ### Additional experiments
> > - Figure 5 may visualize the intermediate modes computed by SGOT, but a barycenter does not necessarily correspond to a physically realizable intermediate system. To make Fig. 5 more convincing, please evaluate the barycenter against concrete metrics that relate to physical realism. For example, mode frequencies, decay rates, and reconstruction/forecast RMSE.
>
> We thank the reviewer for this insightful comment. We agree that evaluating the physical realism of barycenters is an important and challenging research direction that we plan to pursue in future work. For the fluid dynamics experiment, we do not have access to the corresponding simulator, which prevents a quantitative comparison with interpolated shapes. We clarified this limitation in the revised manuscript and highlighted it as a promising direction for future investigation.
>
> > - The experiments report that Koopman operators are estimated with a linear kernel (including Appendix H). Please state why you chose a linear kernel (interpretability, computational reasons, or data suitability) and discuss the relationship with non-linear kernels such as RBF.
>
> As the reviewer mentioned, in the initial submission, Koopman operators were estimated using linear kernels to facilitate interpretability, leverage computational efficiency in the classification tasks, and connect barycenter results with the classical intuition of DMD in fluid dynamics. Since then, we have extended our experiments to include operator estimation with RBF kernels (Appendix G.3) and kernels based on learned deep features (Appendix G.4). As reported in the Global Response tables, SGOT consistently outperforms competing metrics across most datasets and estimation methods, whether linear, RBF, or deep. These additional experiments demonstrate that SGOT’s performance is robust to the choice of kernel.
>
>
> > - Minor points 6. From Section 2 onward you use `cite` where `citep` would be appropriate. Line 308 contains two periods in a row. Figure 2 appears after Fig. 5 and a label seems to be “ilbert”.
>
> We thank the reviewer for pointing out these minor formatting issues. We have corrected them in the revised manuscript.

---

### Official Review · Reviewer_Tstr · 2025-10-31

**Soundness:** 3
**Presentation:** 3
**Contribution:** 3
**Rating:** 4
**Confidence:** 3

**Summary:**

This paper introduces a novel metric, named the Spectral-Grassmann Optimal Transport (SGOT) metric, for comparing dynamical systems represented by Koopman or transfer operators. The core idea is to represent each dynamical system as a probability distribution over its joint spectral data, specifically its eigenvalues and corresponding spectral projectors (viewed as points on a Grassmann manifold). The distance between two systems is then defined as the Wasserstein distance between these two distributions. The ground metric for the optimal transport problem is a weighted combination of the distance between eigenvalues and a Grassmannian metric between the projector subspaces. The authors provide theoretical guarantees for the metric, including finite-sample convergence rates for its estimation. They also propose an algorithm to compute the Fréchet mean (barycenter) of a set of dynamical systems under this metric, enabling interpolation. The method's effectiveness is demonstrated through experiments on simulated and real-world datasets, where it is shown to outperform other operator-based metrics in tasks like dimensionality reduction and classification, and provides more meaningful interpolations.

**Strengths:**

1.  **Principled Metric Definition:** The SGOT metric is well-founded in theory, combining optimal transport with the geometry of Grassmann manifolds. It offers a holistic comparison of operators by considering both their spectral values (eigenvalues) and their geometric structure (eigenspaces).
2.  **Theoretical Guarantees:** The paper provides finite-sample convergence guarantees (Theorem 2), which is a significant theoretical result that adds rigor to the proposed data-driven metric.
3.  **Key Invariances:** The metric is designed to be invariant to the sampling frequency of trajectories and permutations of the spectral decomposition, which are desirable properties for a robust similarity measure.
4.  **Enables Geometric Operations:** The framework naturally allows for the computation of barycenters, providing a principled way to average and interpolate between dynamical systems. The experimental results show this leads to more meaningful interpolations than simpler methods.

**Weaknesses:**

1.  **Strong and Potentially Impractical Assumptions:** The requirement of a "Common functional space" (A3) is a major limitation. It is not clear how one would verify this assumption in practice or how to choose a suitable RKHS when comparing a diverse set of dynamical systems. The paper does not address the sensitivity of the method to violations of this assumption.
2.  **Limited Generality of Guarantees:** The statistical guarantees are tied to a specific operator estimation method (RRR) and kernel type. It is unclear if these guarantees would extend to other popular and powerful estimation methods, such as those based on neural networks.
3.  **Scalability Concerns:** The metric computation involves calculating a cost matrix and solving an OT problem. While the paper claims it is computationally efficient for small ranks (`r`), the complexity `O(n^2 * r^2)` can be prohibitive for high-resolution data (large `n`) or when a large number of modes are needed to accurately represent the system (large `r`). The barycenter computation adds another layer of iterative optimization.
4.  **Limited Experimental Scope for Barycenters:** The interpolation experiments are primarily demonstrated on simple 1D linear systems. The single fluid dynamics example is more compelling, but a more extensive evaluation on a wider range of complex, nonlinear systems is needed to fully validate the claimed superiority of SGOT for interpolation and averaging tasks.

**Questions:**

1.  How can a practitioner verify or enforce the "Common functional space" assumption (A3) when given a collection of time-series datasets from potentially very different underlying systems? What are the consequences for the metric if this assumption is violated (i.e., the relevant eigenspaces do not lie in the chosen RKHS)?
2.  Your statistical guarantees are derived for the RRR estimator. Could you comment on the challenges of extending these guarantees to Koopman operator estimators based on deep neural networks, which are widely used for complex systems? Does the metric `ds` remain well-behaved if the estimated operators `T_k` come from different model classes?
3.  Regarding scalability: What is the practical limit on the number of samples (`n`) and the rank (`r`) for which the SGOT metric and its barycenter can be computed in a reasonable amount of time? How does this compare to the scale of problems typically found in fields like fluid dynamics or climate science?
4.  In Figure 1, the SGOT metric shows a desirable linear-like response. However, this is dependent on the hyperparameter `η`. The paper mentions a sensitivity analysis in the appendix but could you elaborate on how `η` should be chosen in practice? Does it require extensive cross-validation, and does the optimal `η` vary significantly across different tasks and datasets?

---

> ### Author Response · Authors · 2025-11-20
>
> We thank the reviewer for the positive comments in the strengths section. We first invite the reviewer to consult the global response; in what follows, we address the specific weaknesses and questions raised.
>
> ### Implication of the common functional space (A3)
> > - Strong and Potentially Impractical Assumptions: The requirement of a "Common functional space" (A3) is a major limitation. It is not clear how one would verify this assumption in practice or how to choose a suitable RKHS when comparing a diverse set of dynamical systems. The paper does not address the sensitivity of the method to violations of this assumption.
>
> > - How can a practitioner verify or enforce the "Common functional space" assumption (A3) when given a collection of time-series datasets from potentially very different underlying systems? What are the consequences for the metric if this assumption is violated (i.e., the relevant eigenspaces do not lie in the chosen RKHS)?
>
> We invite the reviewer to refer to paragraph “2. Why a common functional space (A3) is needed” in the global response, where we discuss in detail how to choose a common functional space and how to diagnose violations of the commonality assumption.
>
> In essence, any universal bounded kernel, like the RBF, naturally defines a common functional space, since its associated RKHS lies within the domain of the transfer operators of all ergodic dynamical systems. Moreover, potential violations of assumption A3 can be detected by estimating the metric distortion between the chosen common subspace and the true operator domain. An efficient estimator for this distortion has been proposed by Kostic et al. (2023), providing a practical diagnostic tool for practitioners.
>
> ### Generalization of the statistical guarantees
>
> > - Limited Generality of Guarantees: The statistical guarantees are tied to a specific operator estimation method (RRR) and kernel type. It is unclear if these guarantees would extend to other popular and powerful estimation methods, such as those based on neural networks.
>
> > - Your statistical guarantees are derived for the RRR estimator. Could you comment on the challenges of extending these guarantees to Koopman operator estimators based on deep neural networks, which are widely used for complex systems? Does the metric ds remain well-behaved if the estimated operators $T_k$ come from different model classes?
>
> The proof of Theorem 2, which establishes statistical guarantees for the convergence of the SGOT metric, essentially relies on properties of discrete optimal transport. As a consequence, the proof methodology can be extended to any estimation method (including neural networks) that provides a learning guarantee for the spectral decomposition of transfer operators,  immediately yielding a corresponding guarantee for the estimation of SGOT. In this paper, we focus on the RKHS + RRR setting because it offers sharp and well-understood spectral guarantees, not because SGOT depends on RKHS methodology. For instance, learning guarantees such as those provided by Kostic et al. (2023) for Principal Component Regression can be directly combined with our analysis.
>
> Finally, our empirical results (Appendix G.2–G.4 and paragraph 4 of the global response) confirm that SGOT behaves consistently across various operator estimators, including linear kernels, RBF, and deep neural networks.
>
> Kostic et al. (2023). Sharp spectral rates for Koopman operator learning, NeurIPS (2023).

---

> ### Author Response · Authors · 2025-11-20
>
> ### Metric scalability
> > -  The metric computation involves calculating a cost matrix and solving an OT problem. While the paper claims it is computationally efficient for small ranks r, the complexity $O(n^2 * r^2)$ can be prohibitive for high-resolution data (large n) or when a large number of modes are needed to accurately represent the system (large r). The barycenter computation adds another layer of iterative optimization.
>
> >-  What is the practical limit on the number of samples (n) and the rank (r) for which the SGOT metric and its barycenter can be computed in a reasonable amount of time? How does this compare to the scale of problems typically found in fields like fluid dynamics or climate science?
>
> We first emphasize that the overall computational cost of SGOT is dominated by the construction of the OT cost matrix, whose complexity depends on the chosen operator estimation method. Once the cost matrix is available, computing SGOT itself is extremely fast: **less than 100ms** for 1000×1000 matrices using exact OT solvers. Depending on the functional space dimension (d) and the number of samples (n), different strategies can be adopted:
> - If d<n, estimation with the function basis yields a complexity of $O(d^2r^2)$ (Appendix D.3). In this setting, SGOT is substantially cheaper than competing metrics; for instance, it is **≈40× faster** than Hilbert–Schmidt and **≈100× faster** than the operator norm (Appendix C.2).
> - If n<d, estimation in the dual space (RKHS) is more efficient, leading to the computational complexity $O(n^2r^2)$ as discussed in the main text (RRR with kernel). When n becomes large, scalable kernel approximations, such as Nyström or Random Fourier Features, can be used to further reduce the cost without loosing statistical guarantees,
>    - Meanti, G., Chatalic, A., Kostic, V. R. , Novelli, P., Pontil, M., Rosasco, L., (2023) *Estimating Koopman operators with sketching to provably learn large scale dynamical systems*, Advances in Neural Information Processing Systems 36, 77242-77276
>
> ### Additional experiments
> > - Limited Experimental Scope for Barycenters: The interpolation experiments are primarily demonstrated on simple 1D linear systems. The single fluid dynamics example is more compelling, but a more extensive evaluation on a wider range of complex, nonlinear systems is needed to fully validate the claimed superiority of SGOT for interpolation and averaging tasks.
>
> We thank the reviewer for highlighting the importance of interpolation and averaging between dynamical systems. To our knowledge, our work is among the first to address this problem in a practical and computationally tractable way. In this paper, we focus on linear systems and fluid-dynamics examples to illustrate that SGOT yields interpolations that align with physical intuition. Interpolation between dynamical systems is an important research direction, with key outcomes for applications, that we plan to address specifically in future work.
>
> Regarding the broader applicability of SGOT, our classification experiments already cover a diverse range of domains, including biomedical signals (ECG, EEG), audio, and motion, which inherently involve nonlinear and stochastic dynamics. Additionally, as detailed in the global response “4. Additional Experiments”, we incorporated two further operator-estimation approaches: RBFs and kernels based on learned deep features (see Appendix G.3–G.4). Across all datasets and all operator-estimation methods, SGOT consistently outperforms other metrics (see table in the global response “4. Additional Experiments”). This expanded evaluation demonstrates SGOT’s robustness and effectiveness across diverse systems and estimation settings.
>
> > - In Figure 1, the SGOT metric shows a desirable linear-like response. However, this is dependent on the hyperparameter $\eta$. The paper mentions a sensitivity analysis in the appendix but could you elaborate on how $\eta$ should be chosen in practice? Does it require extensive cross-validation, and does the optimal $\eta$ vary significantly across different tasks and datasets?
>
> As suggested by the reviewer, we included an additional experiment (Appendix G.5) that analyzes SGOT's performance sensitivity to the parameter $\eta$, which controls the trade-off between the eigenvalue and eigensubspace costs. Results show that SGOT’s performance varies smoothly with $\eta$, and the optimal values are typically in favor of the eigensubspace cost. We also introduced a heuristic based on the Nyquist frequency: $\tilde{\eta} = (1+ \frac{f_{samp}}{2\sqrt{2}})^{-1}$ which consistently lies near regions of high accuracy. Across datasets, optimal values are typically located within the interval $(0, \tilde{\eta})$, making $\tilde{\eta}$ both a strong default choice and an effective upper bound for limiting the grid-search range. Overall, selecting $\eta$ does not require extensive cross-validation, and the smooth response enables efficient hyperparameter tuning.

---

### Author Response · Authors · 2025-11-20
**Global response**

We thank all reviewers for their constructive feedback. Several comments touched on aspects: (i) the generality of our framework, and its dependence on operator estimators, (ii) the role of the common functional space assumption, (iii) computational scalability and (iv) experimental validation of the method. We address these key points here.

### **1. SGOT is a *comparison* framework, not an estimation method**
**SGOT is an estimator-agnostic, principled, and computationally efficient metric for comparing (stochastic) dynamical systems.** Importantly, SGOT is **not** a method for estimating transfer or Koopman operators.

Practitioners may use any relevant kernel method, EDMD variant, dictionary learning, or neural architectures to learn a **finite-rank spectral representations** of the underlying transfer operators and then use our SGOT metric to compare them.

**Once operators are estimated in a shared functional space, SGOT offers a fast and well-behaved metric to compare them.**

Our theoretical analysis is developed in the RKHS + RRR setting for comparing systems at equilibrium because this framework offers sharp well-understood spectral guarantees for estimation of the operators, **not** because SGOT depends on RKHS methodology nor because it is limited to systems at equilibrium.

### 2. Why a *common functional space* (A3) is needed
SGOT compares **spectral subspaces** via Grassmannian geometry, which is only meaningful if all subspaces lie in the *same* ambient space. This is why Assumption (A3) is required: it ensures that operators are compared on the part of their action that is consistently represented across systems.

This corresponds to projecting operators onto a **shared functional space**. If this projection is accurate, so that the missing components are small, then SGOT provides an accurate comparison. Violations of A3 can be diagnosed through the metric distortion between the chosen common subspace and the true operator domain, a phenomenon quantified in Kostic et al. (2023), where an efficient estimator of this distortion is provided.

Finally, we would like to draw attention of the reviewers to lines 167-173 of our original subission, and stress out that our A3 is significanly weaker compared to the regularity assumption under which SOTA guarantees from the above cited work were obtained.

* Kostic et al. (2023). Sharp spectral rates for Koopman operator learning, NeurIPS (2023).

**Practical construction of the common space can be:**

* **Any RKHS with a universal bounded kernel (e.g., RBF)**, whose closure is well known to be contained in every $L^2_\pi(\mathcal{X})$ associated with ergodic dynamical systems, or
* **A learned shared embedding** produced by representation-learning methods, designed so that all operators are represented in a common finite-dimensional space that minimizes projection biases.

Many existing approaches correspond to the cases above. We will make this clearer in the revision.

### 3. SGOT is computationally efficient
SGOT is very fast once spectral decompositions are available. In particular:

* rank-1000 operators with pre computed cost matrices can be compared in **less than 100 ms** with exact OT solvers,
* SGOT is **≈40× faster** than Hilbert–Schmidt and **≈100× faster** than the operator norm (Appendix C.2).

If RKHS methods are used, then approximation techniques (Nyström, RFF, low-dimensional embeddings) provide further scalability when needed.

### 4. Additional Experiments
We added three additional experiments to the paper:
1. **Classification using operators estimated with RBF kernels (App. G.3)**
2. **Classification using operators estimated with kernel based on learned deep features (App. G.4)**
3. **Sensitivity analysis of the parameter controlling the trade-off between eigenvalue and eigensubspace costs (App. G.5)**

**Classification results:** Across all classification settings, the **SGOT metric consistently outperforms all alternative metrics** on most datasets, as shown in the average-rank table:

**Average rank (lower is better)** *(OP-RBF not computed)*

||HS|OP|Martin|SOT|GOT|SGOT|
|-|-|-|-|-|-|-|
|Linear| 3.29 | 3.92 | 5.3 | 4.49 | 2.66 | **1.34** |
|RBF| 3.74 | NA | 4.02 | 3.28 | 2.48 | **1.48**  |
|Deep| 3.33 | 4.14 | 5.06 | 3.84 | 2.94 | **1.71**|

Relative ranking of all metrics remains stable across all estimation methods. These results show that SGOT is a robust and well-behaved metric for comparing operators, regardless of the estimation method. Detailed results are provided in the paper's appendices G.3-4 and in the next paragraph: *5. Detailed Experimental Results*.

**Sensitivity results:** SGOT’s performance varies smoothly with respect to the trade-off parameter $\eta$, and a simple $\eta$-heuristic based on the Nyquist frequency achieves performance close to the optimum. Combined with SGOT’s low computational cost, this smooth behavior and heuristic enable efficient hyperparameter tuning via fast grid-search strategies.

---

> ### Author Response · Authors · 2025-11-20
> **5. Detailed Experimental Results**
>
> We list classification accuracies (mean ± std over 5 runs) of K-NN defined with different metrics (Hilbert-Schmidt, Operator, Martin, SOT, GOT, SGOT (ours)) and for different operator estimation methods (RRR with RBF kernel, or kernel based on learned deep features). Best results are highlighted in bold.
>
> **Results with RBF kernel (Appendix G.3):**
> |  | Hilbert-Schmidt    | Martin          | SOT                         | GOT    | SGOT (ours)              |
> |-----------------------------|-----------------------------|-----------------|-----------------------------|-----------------------------|--------------------------|
> | BasicMotions                | 0.26 $\pm$ 0.17             | 0.77 $\pm$ 0.06 | 0.87 $\pm$ 0.05             | 0.69 $\pm$ 0.14             | **0.95 $\pm$ 0.02**      |
> | ERing                       | 0.74 $\pm$ 0.07             | 0.22 $\pm$ 0.05 | 0.38 $\pm$ 0.05             | 0.96 $\pm$ 0.01             | **0.98 $\pm$ 0.02**      |
> | Epilepsy                    | 0.31 $\pm$ 0.02             | 0.8 $\pm$ 0.01  | 0.77 $\pm$ 0.02             | 0.93 $\pm$ 0.02             | **0.95 $\pm$ 0.02**      |
> | FingerMovements             | 0.53 $\pm$ 0.06             | 0.5 $\pm$ 0.03  | **0.53 $\pm$ 0.05**.        | 0.5 $\pm$ 0.06              | 0.53 $\pm$ 0.01          |
> | NATOPS                      | 0.59 $\pm$ 0.06             | 0.25 $\pm$ 0.02 | 0.35 $\pm$ 0.02             | 0.78 $\pm$ 0.03             | **0.8 $\pm$ 0.05**       |
> | avg. rank (lower is better) | 3.74 $\pm$ 1.27             | 4.02 $\pm$ 0.98 | 3.28 $\pm$ 1.15             | 2.48 $\pm$ 1.19             | **1.48 $\pm$ 0.7**       |
>
>
> **Results with kernels based on learned deep features (Appendix G.4):**
> || Hilbert-Schmidt | Operator  | Martin  | SOT   | GOT   | SGOT (ours)   |
> |-|-|-|-|-|-|-|
> | AtrialFibrillation    | 0.22 $\pm$ 0.21    | 0.18 $\pm$ 0.1              | 0.24 $\pm$ 0.14             | 0.24 $\pm$ 0.09             | 0.2 $\pm$ 0.21              | **0.36 $\pm$ 0.14**         |
> | BasicMotions                | 0.28 $\pm$ 0.05             | 0.34 $\pm$ 0.07             | 0.81 $\pm$ 0.06             | 0.68 $\pm$ 0.11             | **0.92 $\pm$ 0.06**         | 0.9 $\pm$ 0.08              |
> | Cricket                     | **0.82 $\pm$ 0.05**         | 0.73 $\pm$ 0.04             | $\emptyset$                 | 0.4 $\pm$ 0.1               | 0.72 $\pm$ 0.07             | 0.72 $\pm$ 0.07             |
> | ERing                       | 0.31 $\pm$ 0.07             | 0.31 $\pm$ 0.04             | $\emptyset$                 | 0.39 $\pm$ 0.04             | 0.55 $\pm$ 0.06             | **0.57 $\pm$ 0.04**         |
> | EigenWorms                  | 0.78 $\pm$ 0.03             | 0.74 $\pm$ 0.08             | $\emptyset$                 | 0.72 $\pm$ 0.06             | 0.80 $\pm$ 0.06             | **0.82 $\pm$ 0.05**         |
> | Epilepsy                    | 0.7 $\pm$ 0.06              | 0.59 $\pm$ 0.05             | $\emptyset$                 | 0.91 $\pm$ 0.06             | 0.88 $\pm$ 0.04             | **0.92 $\pm$ 0.05**         |
> | FingerMovements             | 0.52 $\pm$ 0.04             | 0.52 $\pm$ 0.05             | 0.48 $\pm$ 0.05             | 0.51 $\pm$ 0.04             | 0.51 $\pm$ 0.03             | **0.56 $\pm$ 0.04**         |
> | HandMovementDirection       | 0.25 $\pm$ 0.02             | 0.27 $\pm$ 0.05             | 0.23 $\pm$ 0.05             | 0.28 $\pm$ 0.03             | 0.26 $\pm$ 0.04             | **0.32 $\pm$ 0.04**         |
> | Handwriting                 | 0.07 $\pm$ 0.02             | 0.05 $\pm$ 0.01             | $\emptyset$                 | 0.07 $\pm$ 0.01             | 0.08 $\pm$ 0.02             | **0.10 $\pm$ 0.01**         |
> | Heartbeat                   | 0.7 $\pm$ 0.02              | 0.69 $\pm$ 0.04             | 0.72 $\pm$ 0.04             | 0.71 $\pm$ 0.04             | 0.71 $\pm$ 0.03             | **0.73 $\pm$ 0.04**         |
> | NATOPS                      | 0.41 $\pm$ 0.08             | 0.27 $\pm$ 0.06             | 0.3 $\pm$ 0.06              | 0.33 $\pm$ 0.07             | 0.45 $\pm$ 0.05             | **0.47 $\pm$ 0.03**         |
> | SelfRegulationSCP1          | 0.77 $\pm$ 0.04             | 0.77 $\pm$ 0.03             | 0.51 $\pm$ 0.03             | 0.52 $\pm$ 0.04             | 0.77 $\pm$ 0.03             | **0.78 $\pm$ 0.03**         |
> | StandWalkJump               | 0.16 $\pm$ 0.13             | 0.27 $\pm$ 0.13             | $\emptyset$                 | 0.27 $\pm$ 0.1              | 0.22 $\pm$ 0.08             | **0.36 $\pm$ 0.09**         |
> | UWaveGestureLibrary         | **0.74 $\pm$ 0.05**         | 0.59 $\pm$ 0.02             | $\emptyset$                 | 0.29 $\pm$ 0.04             | 0.66 $\pm$ 0.05             | 0.66 $\pm$ 0.05             |
> | avg. rank (lower is better) | 3.33 $\pm$ 1.56             | 4.14 $\pm$ 1.27             | 5.06 $\pm$ 1.48             | 3.84 $\pm$ 1.34             | 2.94 $\pm$ 1.33             | **1.71 $\pm$ 0.77**         |

---

### Meta-Review · Area_Chair_Vcef · 2026-01-07

**Summary:**

The paper propose a novel approach for representing the transfer operator of a dynamical system as a distribution of its joint operator eigenvalues and spectral projectors, leading to a metric between systems via optimal transport. Numerical experiments are provided to illustrate the proposed metric.

Reviewers generally appreciate the proposed mathematical framework and the accompanying numerical experiments, which consistently outperform previous methods.

Note that one limitation of the proposed framework is that it is a metric on the set of all non-defective operators of rank at most $r$, that is it is not a metric on the set of all non-defective operators. This limitation should be explicitly pointed out.

**Reviewer Concerns:**

Some concerns from reviewers include

- The requirement of a "Common functional space" (A3): Reviewer Tstr. In their response,  the authors state that any universal bounded kernel, like the RBF, would provide such a common function space.

- Scalability Concerns: Reviewer Tstr. In their response,  the authors state that the proposed method is much faster compared with some previous methods,

- Experiments with linear kernels: Reviewer WDpQ,  Reviewer AGY8. In their rebuttal, the authors provided further experiments with other kernels, including RBF kernel, kernel based on learned deep features, showing that their method consistently outperforms previous work.

**Reviewer Scores:**

The scores are 4,6,6,6

Reviewer Tstr (4), who would not object to the paper being accepted, would likely have raised their score, given that some of their main concerns have been addressed.

---

### Decision · Program_Chairs · 2026-01-26

Accept (Poster)